# HyperMLP: An Integrated Perspective for Sequence Modeling

**Jiecheng Lu**[1]   **Shihao Yang**[1]

## Abstract

Self-attention is often viewed as probabilistic query-key lookup, motivating designs that preserve normalized attention scores and fixed positional semantics. We advocate a simpler and more unified perspective: an autoregressive attention head can be viewed as a dynamic two-layer MLP whose weights are instantiated from the context history. From this view, attention scores form an ever-growing hidden representation, and standard MLP activations such as ReLU or GLU naturally implement input-conditioned selection over a context-dependent memory pool rather than a probability distribution. Based on this formulation, we introduce **HyperMLP** and **HyperGLU**, which learn dynamic mixing in both feature space and sequence space, using a reverse-offset (lag) layout to align temporal mixing with autoregressive semantics. We provide theoretical characterizations of the expressivity and implications of this structure, and empirically show that HyperMLP/HyperGLU consistently outperform strong softmax-attention baselines under matched parameter budgets. Code is available at this link.

## 1. Introduction

Transformers, built on self-attention, dominate sequence modeling and serve as the backbone of foundation models across language, vision, speech, and decision making (Vaswani et al., 2017; Devlin et al., 2019; Brown et al., 2020; Dosovitskiy et al., 2020; Chen et al., 2021; Touvron et al., 2023). In autoregressive (AR) generation, attention enables parallel training and efficient inference: each new token attends to the prefix via an incrementally maintained KV cache. Stacked with residual connections, this prefix access supports long-range retrieval and is closely tied to

in-context learning behaviors (Olsson et al., 2022; Elhage et al., 2021).

The empirical "scaling laws" have further pushed models toward larger parameter size training (Kaplan et al., 2020; Hoffmann et al., 2022). However, recent industrial practice suggests that the marginal returns of continued scaling are increasingly difficult to sustain (Hu & Tong, 2024; Gundlach et al., 2025), bringing the question of how to improve capability per unit resource back to the research community. A common response is the efficiency direction: subquadratic attention or alternative sequence models to reduce per-token cost while preserving capability (Gu & Dao, 2023; Poli et al., 2023). Yet such changes often restrict the attention block's function class, leading to saturation and a persistent gap to full quadratic attention. This motivates a complementary expressivity direction: **revisiting what the attention block parameterizes** to improve expressive ability under the same parameter/compute budget.

Classically, attention is often introduced as a mechanism that looks fundamentally different from standard MLP/CNN/RNN: a content-addressable lookup in which a query matches against keys, softmax converts the resulting scores into a distribution over positions, and the output is an expectation-style read of the corresponding values (Vaswani et al., 2017; Graves et al., 2014; Miller, 2016). This probabilistic view shapes both mechanistic interpretation (e.g., attention maps as alignments or importance signals, with debated faithfulness) (Jain & Wallace, 2019; Abnar & Zuidema, 2020) and engineering practice, where many efficient variants aim to approximate the same probability-over-positions behavior (Child, 2019; Beltagy et al., 2020; Zaheer et al., 2020). However, recent results suggest the probability-simplex constraint may not be essential and can be restrictive: alternative normalizations and signed/affine weightings can improve general ability (Richter & Wattenhofer, 2020; Ye et al., 2025; Lv et al., 2025). These results motivate our stance: rather than treating attention scores as probabilities that must be preserved, we view an attention head as the simplest dynamic two-layer MLP, where the "scores" are an ever-growing hidden representation that can adopt whatever mixing geometry the data demands.

**The bitter lesson.** Following Sutton's observation that scalable, data-driven methods tend to outperform hand-designed

[1]Georgia Institute of Technology, Atlanta, US. Correspondence to: Jiecheng Lu <jlu414@gatech.edu>, Shihao Yang < shihao.yang@isye.gatech.edu>.

*Proceedings of the 43rd International Conference on Machine Learning*, Seoul, South Korea. PMLR 306, 2026. Copyright 2026 by the author(s).

feature engineering (Sutton, 2019), we advocate an integrated perspective: an autoregressive attention head can be simply viewed as a two-layer MLP whose weights are dynamically instantiated from the context $X_{1:t}$, and the "attention scores" form an ever-growing width-$t$ hidden representation. Recent evidence that lightweight sequence-axis operators (e.g., convolutions) improve attention when inserted around projections suggests that the sequence dimension itself can be an effective learnable mixing space (So et al., 2021). We therefore treat the score space like a standard MLP hidden space: we learn explicit sequence mixing to relax fixed positional coordinates, and we use familiar MLP activations (ReLU/GLU) with L2 normalization (similar to RMSNorm) instead of probability normalization (Shazeer, 2020; Zhang & Sennrich, 2019; Richter & Wattenhofer, 2020). This lens also directly explains several empirical design choices, such as which projections are most effective to compress/adapt in low-rank fine-tuning (Hu et al., 2021) and where gating is most beneficial (Hua et al., 2022; Qiu et al., 2025). It also distinguishes our causal temporal mixing from MLP-Mixer/gMLP-style token mixers and classic fast-weight views (Tolstikhin et al., 2021; Liu et al., 2021; Schmidhuber, 1992; Schlag et al., 2021); see Appendix L.8 for details.

Building on this perspective, we propose **HyperMLP**, which learns input-conditioned mixing in both feature space and sequence space to instantiate flexible 2-layer MLP weights from the context. HyperMLP performs sequence mixing on the reverse-offset (lag) history $X_{t:1}$, aligning semantics across autoregressive steps. Using conditioned low-rank parameterizations for feature and sequence mixing and ReLU (or a GLU variant) for activation, we show theoretically and empirically that HyperMLP form an expressive superset of baseline ReLU attention and consistently outperform strong softmax attention baselines under matched parameter budgets.

Our contributions are: (i) reframing autoregressive attention as a dynamic two-layer MLP with width-$t$ hidden scores and ReLU/GLU-based input-conditioned selection; (ii) introducing HyperMLP/HyperGLU with learned feature/sequence mixing and a lag layout; (iii) providing theoretical characterizations that explain several prior attention design principles; and (iv) demonstrating consistent empirical gains over strong attention baselines at matched parameter budgets. The scaling-laws discussion above is intended as motivation rather than a claim; Appendix L.7 states the precise scope.

## 2. Redesigning Attention from First Principles

### 2.1. ReLU MLPs select input-conditioned sub-networks

We start from a depth-two ReLU MLP and omit biases as in common LLM practice:

$$o = \text{ReLU}\big(xW_{\text{MLP}}^{(1)}\big) W_{\text{MLP}}^{(2)}, \qquad x \in \mathbb{R}^{1 \times d}. \quad (1)$$

As is well known, the ReLU non-linearity implements an input-conditioned gate over hidden units, so that each input effectively selects a linear sub-network of this two-layer map (see Fig. 1(c)); we give a derivation in Appendix C.1.

### 2.2. Attention as a dynamic two-layer MLP

We apply the same template to an autoregressive attention head. Let $x_t \in \mathbb{R}^{1 \times d}$ be the current token/state and $X := X_{1:t} = [x_1; x_2; \ldots; x_t] \in \mathbb{R}^{t \times d}$ be the prefix. For one head,

$$q_t := x_t W_q, \qquad K := X W_k, \qquad V := X W_v,$$

where $W_q, W_k \in \mathbb{R}^{d \times d_{qk}}$ and $W_v, W_o \in \mathbb{R}^{d \times d_{vo}}$, write

$$o_t = \sigma(q_t K^\top) V W_o^\top = \sigma\big(x_t W_q W_k^\top X^\top\big) X W_v W_o^\top \quad (2)$$

with $\sigma(\cdot)$ typically $\text{softmax}(\cdot)$. We refer to the increasingly studied alternative that replaces softmax with a normalized ReLU-style map as *ReLU attention* (Richter & Wattenhofer, 2020; Zhang et al., 2021; Wortsman et al., 2023; Shen et al., 2023; Bai et al., 2023). This choice determines whether the hidden vector is interpreted as probabilities (softmax) or as a gate for input-conditioned selection (ReLU-style). Since ReLU attention often matches softmax empirically while being algebraically simpler, we use the ReLU-style activation as default in our analysis; see Section 3.

The key observation is that (2) already has the similar form of a depth-two MLP, except that its weights are instantiated from the context $X$ (see Fig. 1(a)). Define the effective (context-instantiated) matrices

$$W_{\text{MLP}}^{(1)}(X) := W_q W_k^\top X^\top \in \mathbb{R}^{d \times t}, \qquad (3)$$

$$W_{\text{MLP}}^{(2)}(X) := X W_v W_o^\top \in \mathbb{R}^{t \times d}. \qquad (4)$$

$$o_t = \sigma\big(x_t W_{\text{MLP}}^{(1)}(X)\big) W_{\text{MLP}}^{(2)}(X), \qquad (5)$$

$$h_t := x_t W_{\text{MLP}}^{(1)}(X) = q_t K^\top \in \mathbb{R}^{1 \times t}. \quad (6)$$

Thus the "attention scores" $h_t$ are simply a width-$t$ hidden pre-activation, and the head is a two-layer *dynamic* MLP whose hidden width grows with context length.

This viewpoint unifies many variants as edits to the same dynamic-MLP backbone (Fig. 2; Fig. 1(a)). With RoPE (Su et al., 2024), the parameterization of $W_{\text{MLP}}^{(1)}(X)$ is altered by inserting a block-diagonal (per sequence channel of $X$) rotation, $q_t k_i^\top \rightsquigarrow x_t W_q \mathcal{R}_{t,i} W_k^\top x_i^\top$, while output gating

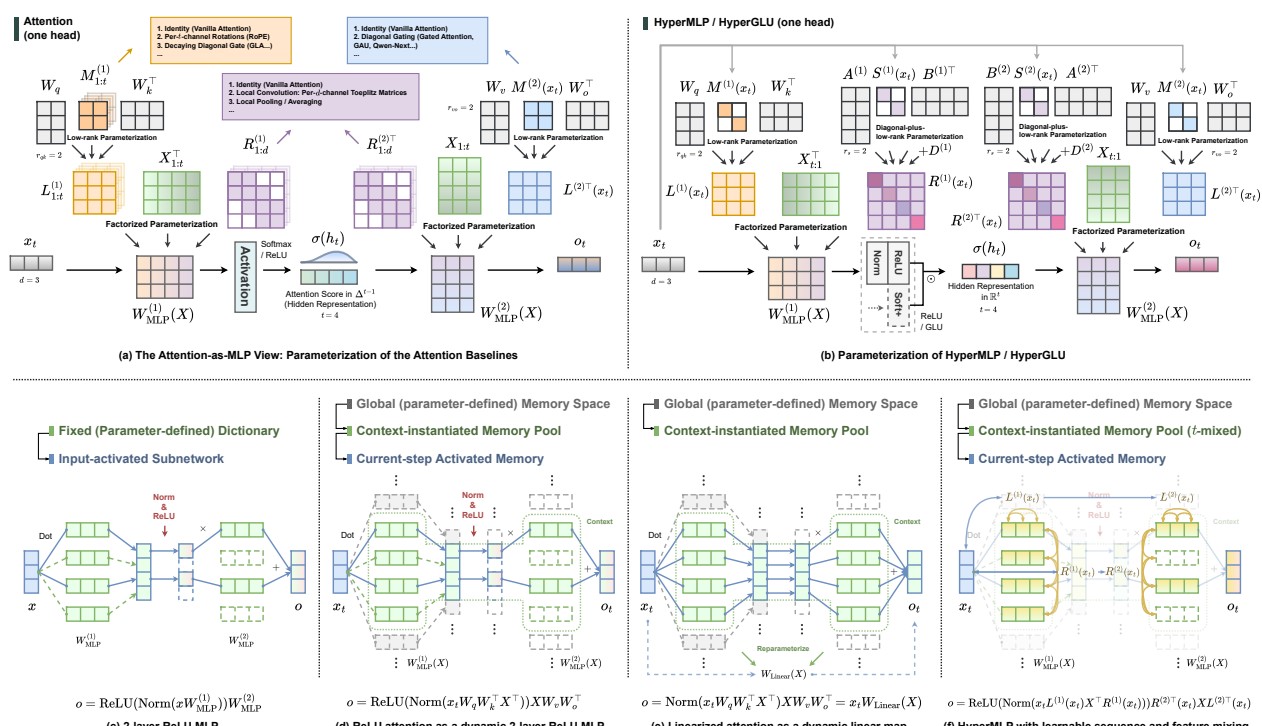

*Figure 1.* The integrated attention-as-MLP view: dynamic two-layer MLPs, memory instantiation, and HyperMLP.

(Qiu et al., 2025) inserts a diagonal gate on the readout side, $XW_vW_o^\top \rightsquigarrow XW_v G(x_t) W_o^\top$, $G(x_t)$ diagonal. Under this view, multi-head attention is simply the sum of $n_{\text{head}}$ parallel dynamic MLPs: concatenating weighted values followed by a shared output projection $W_o$ is equivalent to absorbing $W_o$ into each head's parameterization (Fig. 3(b)).

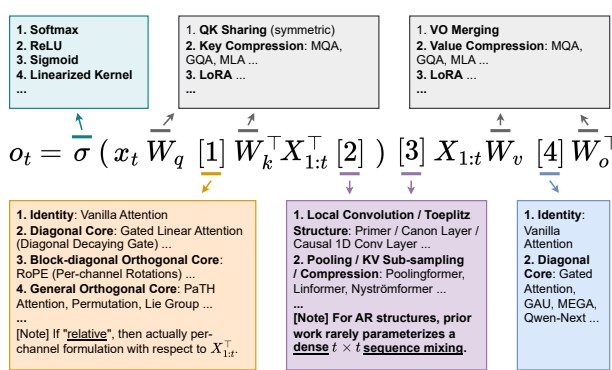

*Figure 2.* Attention as a dynamic two-layer MLP: many attention variants can be viewed as edits in the same backbone: $\sigma$ (normalization/gating), QK feature mixing (sharing/compression/structured cores such as RoPE), sequence-axis mixing on $X_{1:t}$ (e.g., convolution/pooling/low-rank mixing), and VO readout (merging/compression/gating).

### 2.3. Limitation of the classical form: a fixed positional basis in the hidden space

**Reading guide.** An intuition-first walkthrough of this and the next subsection, with the denser symbolic derivations broken out separately, is provided in Appendix L.9.

Writing attention in the dynamic two-layer form (6) exposes a key limitation: although the weights are context-instantiated ($W_{\text{MLP}}^{(1)}(X) = W_q W_k^\top X^\top$ and $W_{\text{MLP}}^{(2)}(X) = XW_vW_o^\top$), the hidden representation $h_t \in \mathbb{R}^{1\times t}$ remains hard-tied to positions: coordinate $i$ corresponds to the interaction between $x_t$ and the $i$-th row of $X$. Since both factors pass through $X$ with no trainable mixing acting on $\mathbb{R}^t$, a single head can only gate and aggregate in this fixed positional basis, rather than learning a task-adapted basis in the hidden dimension as a standard MLP can. Recent results show that even lightweight local sequence mixing can yield substantial gains (Allen-Zhu, 2025), motivating explicit dense mixing along the hidden (sequence) dimension.

### 2.4. HyperMLP: learning sequence mixing effectively

We implement each head as a dynamic two-layer MLP whose weights are instantiated from the **lag-ordered** history $X_{t:1} = [x_t; x_{t-1}; \ldots; x_1] \in \mathbb{R}^{t\times d}$ (see Theorem 2.3, C.6 and F.1 for why a reversed lag order (offset layout) better aligns the canonical length extension of sequence mixing with autoregressive truncation semantics). Let $d_{qk}, d_{vo} \ll d$ denote the per-head left side ranks of dynamic MLP weights (typically $d_{qk} \approx d_{vo} \approx d/n_{\text{head}}$). For one head, we write

$$h_t := x_t W_{\text{MLP}}^{(1)}(X_{t:1}) \in \mathbb{R}^{1\times t}, \tag{7}$$

$$o_t := \sigma(h_t) W_{\text{MLP}}^{(2)}(X_{t:1}) \in \mathbb{R}^{1\times d}, \tag{8}$$

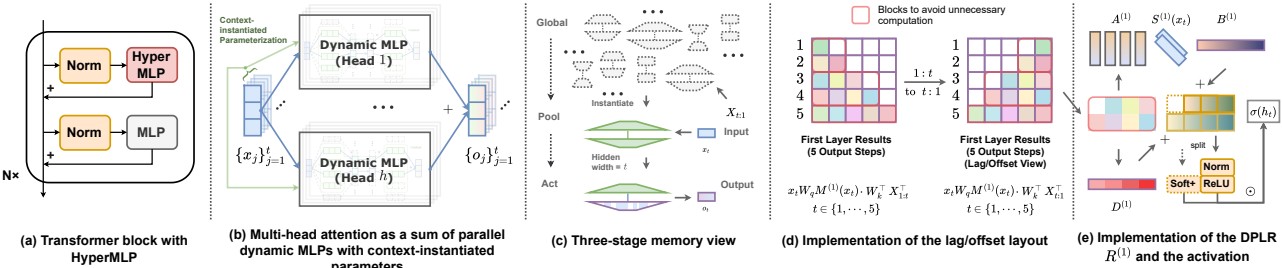

*Figure 3.* The integrated attention-as-MLP view: Dynamic MLP Heads, 3-stage memory, Lag Layout, and DPLR sequence Mixing.

where the activation[1] is $\sigma(z) = \text{ReLU}\big(\text{L2Norm}_t(z)\big)$. The dynamic weights are instantiated by a factorized (core-on-context) parameterization:

$$W_{\text{MLP}}^{(1)}(X_{t:1}) := L^{(1)}(x_t) \, X_{t:1}^\top \, R^{(1)}(x_t) \in \mathbb{R}^{d \times t} \qquad (9)$$

$$W_{\text{MLP}}^{(2)}(X_{t:1}) := R^{(2)\top}(x_t) \, X_{t:1} \, L^{(2)\top}(x_t) \in \mathbb{R}^{t \times d} \quad (10)$$

We write $W^{(\cdot)}(X_{t:1})$ as a slight abuse of notation $W^{(\cdot)}(X_{t:1}, x_t)$ since $x_t$ is the most recent row of $X_{t:1}$. Here $L^{(1)}(\cdot), L^{(2)}(\cdot) \in \mathbb{R}^{d \times d}$ mix features (left space), and $R^{(1)}(\cdot), R^{(2)}(\cdot) \in \mathbb{R}^{t \times t}$ mix sequence slots (right space). Both are input-conditioned with low-rank or diagonal-plus-low-rank (DPLR) forms:

$$L^{(1)}(x_t) := W_q \, M^{(1)}(x_t) \, W_k^\top, \quad W_q, W_k \in \mathbb{R}^{d \times d_{qk}},$$

$$L^{(2)\top}(x_t) := W_v \, M^{(2)}(x_t) \, W_o^\top, \quad W_v, W_o \in \mathbb{R}^{d \times d_{vo}},$$

$$R^{(j)}(x_t) := D^{(j)} + A^{(j)} S^{(j)}(x_t) B^{(j)\top}, \quad j \in \{1, 2\},$$

with

$$M^{(1)}(x_t) := \text{Diag}\big(\phi(x_t W_M^{(1)})\big) \in \mathbb{R}^{d_{qk} \times d_{qk}},$$

$$M^{(2)}(x_t) := \text{Diag}\big(\phi(x_t W_M^{(2)})\big) \in \mathbb{R}^{d_{vo} \times d_{vo}},$$

$$S^{(j)}(x_t) := \text{Diag}\big(\phi(x_t W_S^{(j)})\big) \in \mathbb{R}^{r_s \times r_s},$$

$$D^{(j)} := I + \text{Diag}(p^{(j)}) \in \mathbb{R}^{t \times t}, \quad A^{(j)}, B^{(j)} \in \mathbb{R}^{t \times r_s},$$

where $\phi(\cdot) = \text{Sigmoid}(\cdot)$. Intuitively, $M^{(1)}(x_t)$ and $M^{(2)}(x_t)$ provide input-conditioned per-channel scaling inside low-rank feature mixing, while $R^{(j)}(x_t)$ learns input-conditioned temporal mixing through a rank-$r_s$ update on top of a learned diagonal baseline.

**Notation and relation to prior variants.** We keep the symbols $W_q, W_k, W_v, W_o$ only to match attention conventions, but we emphasize a different interpretation: they are simply trainable factors in a per-head low-rank parameterization of the feature-space mixing operators $L^{(1)}$ and $L^{(2)}$; we do not ascribe special meaning to explicit "queries/keys/values" beyond intermediate computations (Elhage et al., 2021). Viewing (9), many familiar designs become special cases:

vanilla attention corresponds to $R^{(1)} = R^{(2)} = I$ with $\sigma = softmax$ (or a ReLU-based $\sigma$); RoPE and output gating can be seen as inserting structured or diagonal cores into the $L$ feature mixings; convolution variants correspond to structured choices of $R$, see Fig. 1(a-b).

**HyperGLU.** HyperGLU replaces the ReLU hidden activation by a GLU-style modulation. Concretely, we split the first-layer instantiation along the latent core rank (equivalently, allocate two $d_{qk}/2$-dimensional cores, doubling the $n_{heads}$) to produce two $h_t^{\text{scale}}, h_t^{\text{gate}} \in \mathbb{R}^{1 \times t}$, and use

$$a_t := \text{Softplus}\big(h_t^{\text{scale}}\big) \odot \text{ReLU}\big(\text{L2Norm}_t(h_t^{\text{gate}})\big) \quad (11)$$

with $o_t := a_t \, W_{\text{MLP}}^{(2)}(X_{t:1})$ so that selection and magnitude modulation can be decoupled (see Prop. 2.4). The parameterization of HyperMLP/GLU is illustrated in Fig. 1(b).

### 2.5. The Attention-as-MLP View: Pool instantiation, routing geometry, and autoregressive consistency

This subsection makes precise what a HyperMLP / attention as a dynamic 2-layer MLP head is doing at runtime. The central object is the width-$t$ vector $h_t$: its sign pattern determines which context-dependent coordinates are selected (routing), and the selected coordinates are then read out through a second dynamic matrix. We use L2-based normalization only as a stabilizer; under a scalar normalization it does not change the selected set.

Throughout we adopt the regime where $\text{L2Norm}_t$ is a positive scalar rescaling across the length-$t$ vector: there exists $\rho_t : \mathbb{R}^{1 \times t} \to (0, \infty)$ such that $\text{L2Norm}_t(z) = z/\rho_t(z)$.

$$\sigma(z) = \text{ReLU}(\text{L2Norm}_t(z)) = \frac{1}{\rho_t(z)} \text{ReLU}(z), \quad (12)$$

and $\mathbf{1}\{\sigma(z) > 0\} = \mathbf{1}\{z > 0\}$: routing is unchanged; only magnitudes are rescaled (Appendix Lemma B.2).

**Three-stage memory view**[2]**.** As shown in Fig. 1(c-f) and Fig. 3(c), in the attention-as-MLP view, one head passes

---

[1]Throughout this paper, the used $\text{L2Norm}_t(x) := x/\sqrt{\|x\|_2^2 + \varepsilon}$ is affine-free and rescales the entire length-$t$ vector; there is no per-coordinate gain or bias.

[2]Autoregressive convention: at step $t$ we use the lag-ordered prefix $X_{t:1} = [x_t; \ldots; x_1]$ ($x_t$ is the newest row of $X_{t:1}$). We sometimes write $(X, x)$ only to expose the factorization of dynamic weights; here $X = X_{t:1}$ and $x = x_t$ are not independent.

through three memory stages: Global (parameter-defined) Memory Space $\to$ Context-instantiated Memory Pool $\to$ Current-step Activated Memory (Theorem C.5). We denote and summarize the forward as $\Omega \to \mu_{X,x}^{\text{pool}} \to \mu_{X,x}^{\text{act}}$.

Let $\Omega := \mathbb{R}^d \times \mathbb{R}^d$ with atoms $\omega = (u, v)$ and define

$$\alpha_\theta(x; u) := x\, L_\theta^{(1)}(x)\, u^\top \in \mathbb{R}, \qquad (13)$$

$$\beta_\theta(x; v) := v\, L_\theta^{(2)\top}(x) \in \mathbb{R}^{1 \times d}. \qquad (14)$$

Given $X := X_{t:1}$ and $x := x_t$, define mixed contexts $U := R_\theta^{(1)\top}(x)X$, $V := R_\theta^{(2)\top}(x)X$, and slots $(u_i, v_i) = (U[i,:], V[i,:])$ for $i \in [t]$. The pool measure is

$$\mu_{X,x}^{\text{pool}} := \sum_{i=1}^t \delta_{(u_i, v_i)},$$

and the activated measure is the restriction

$$\mu_{X,x}^{\text{act}} := g_x\, \mu_{X,x}^{\text{pool}}, \qquad g_x(u, v) := \mathbf{1}\{\alpha_\theta(x; u) > 0\}.$$

With this notation, the forward pass is a gated readout from the pool instantiated by the current prefix.

**Theorem 2.1** (Dynamic-head decomposition: MLP form and context-wide slots). *Assume* $\text{L2Norm}_t(z) = z/\rho_t(z)$ *where* $\rho_t(z) > 0$ *is a scalar. The head in* (7)–(10) *satisfies:*

*(i) Dynamic two-layer MLP.*

$$o_t = \sigma\big(x_t W_{\text{MLP}}^{(1)}(X_{t:1})\big) W_{\text{MLP}}^{(2)}(X_{t:1}), \qquad (15)$$

$$h_t \in \mathbb{R}^{1 \times t} \text{ is the width-}t \text{ hidden pre-activation.} \qquad (16)$$

*(ii) Pool $\to$ activated readout.* With $(u_i, v_i)$ and $(\mu_{X,x_t}^{\text{pool}}, \mu_{X,x_t}^{\text{act}})$ defined above,

$$o_t = \frac{1}{\rho_t(h_t)} \sum_{i=1}^t \alpha_\theta(x_t; u_i)_+\, \beta_\theta(x_t; v_i) \qquad (17)$$

$$= \frac{1}{\rho_t(h_t)} \int_\Omega \alpha_\theta(x_t; u)_+\, \beta_\theta(x_t; v)\, d\mu_{X,x_t}^{\text{pool}} \qquad (18)$$

$$= \frac{1}{\rho_t(h_t)} \int_\Omega \alpha_\theta(x_t; u)\, \beta_\theta(x_t; v)\, d\mu_{X,x_t}^{\text{act}}. \qquad (19)$$

*See integral notations in* (27). *The derivation is in* C.5.

*(iii) Sequence mixing builds context-wide slots.* For each $i \in [t]$, $u_i = \sum_{j=1}^t R_{j,i}^{(1)}(x_t)\, X_{t:1}[j,:]$, $v_i = \sum_{j=1}^t R_{j,i}^{(2)}(x_t)\, X_{t:1}[j,:]$, *and the token-wise basis is the special case* $R^{(1)}(x_t) = R^{(2)}(x_t) = I_t$.

*Proof sketch.* (i) is immediate from (7)–(10). (ii) uses $\text{ReLU}(\text{L2Norm}_t(z)) = \rho_t(z)^{-1}\text{ReLU}(z)$ and the identities $X_{t:1}^\top R^{(1)}(x_t) = U^\top$, $R^{(2)\top}(x_t)X_{t:1} = V$; see C.5 and Prop. E.1. (iii) is the row expansion of $U = R^{(1)\top}(x_t)X_{t:1}$ and $V = R^{(2)\top}(x_t)X_{t:1}$; see Prop. H.3. $\square$

Routing is the active-set partition induced by the sign pattern of $h_t$. When the mixing is static in $x_t$, this partition is the usual hyperplane arrangement of a two-layer ReLU map; when the mixing depends on $x_t$, the boundaries become curved level sets.

**Corollary 2.2** (Warped routing strictly generalizes polyhedral routing). *Fix a context* $X_{t:1}$. *If* $R^{(1)}(\cdot)$ *and* $L^{(1)}(\cdot)$ *are constant in* $x_t$, *then each gating boundary* $\{x_t : h_{t,i} = 0\}$ *is a hyperplane, hence the activated-set partition of* $\mathbb{R}^d$ *is polyhedral. If* $R^{(1)}(x_t)$ *(or* $L^{(1)}(x_t)$) *depends smoothly and nontrivially on* $x_t$, *then the boundaries are generically curved level sets, yielding a richer routing geometry.*

*Proof sketch.* Static case: Theorem C.10 (and Corollary C.12 for diagonal $R^{(1)}$). Dynamic case: Theorem C.13, with deformation term in Proposition C.14. $\square$

**Lag Order.** In autoregressive generation the available prefix length varies, so we want the head to be consistent when extending the history by adding far-past tokens. Under the lag order, the canonical *prefix* extension of sequence operators matches the autoregressive truncation window.

**Theorem 2.3** (Lag layout: extension consistency implies AR truncation invariance). *Fix integers* $1 \leq t < T$ *and let* $P_{t \to T} = \begin{bmatrix} I_t \\ 0 \end{bmatrix}$. *Let* $\widetilde{X}_{T:1} \in \mathbb{R}^{T \times d}$ *be lag-ordered (newest$\to$oldest) and set* $x := \widetilde{X}_{T:1}[1,:] \in \mathbb{R}^{1 \times d}$. *Let the lag-prefix truncation (most recent $t$ rows) be* $X_{t:1} := P_{t \to T}^\top \widetilde{X}_{T:1} = \widetilde{X}_{T:1}[1:t,:] \in \mathbb{R}^{t \times d}$, *so* $x = X_{t:1}[1,:]$.

*Assume extension consistency for* $m \in \{1, 2\}$:

$$R_T^{(m)}(x) = P_{t \to T}\, R_t^{(m)}(x)\, P_{t \to T}^\top, \qquad (20)$$

*and padding invariance* $\rho_T([z, 0]) = \rho_t(z)$. *Then appending older history has no effect:* $o_T(x; \widetilde{X}_{T:1}) = o_t(x; X_{t:1})$.

*Proof sketch.* This is Appendix Theorem F.1 (see also Appendix Corollary C.7). For HyperMLP, the DPLR $R^{(m)}(\cdot)$ admits a prefix-extension-consistent construction satisfying (20); see Lemma C.6. Padding invariance holds for $\rho_\ell(z) = \sqrt{\|z\|_2^2 + \varepsilon}$ (Equation (49)).[3] $\square$

**Implications of the attention-as-MLP view.** First, two design choices we used can be stated cleanly in this attention-as-MLP language: i) we use GLU activation since it separates selection from slot strength, and ii) under a fixed parameter budget it is better to preserve the second-layer/readout (VO) rank while shrinking the first-layer/routing (QK) rank.

---

[3]In practice, we parameterize DPLR factors at a fixed maximum length and slice the length-$t$ prefix at runtime, realizing the canonical prefix embedding $R_T = P_{t \to T} R_t P_{t \to T}^\top$ used in the extension-consistency analysis.

**Proposition 2.4** (HyperGLU decouples routing and magnitude)**.** *Let* $h_t^{\mathrm{gate}}, h_t^{\mathrm{scale}} \in \mathbb{R}^{1 \times t}$ *be two first-layer score vectors produced by the same instantiation form as* $h_t$ *(with split cores). With the notations in* (11)*, the activated set depends only on the sign pattern of* $h_t^{\mathrm{gate}}$ *(routing), while* $\mathrm{Softplus}(h_t^{\mathrm{scale}}) > 0$ *independently modulates magnitudes.*

*Proof sketch.* By (12), $\mathrm{L2Norm}_t$ preserves $\mathbf{1}\{h_t^{\mathrm{gate}} > 0\}$, so routing is set by the gate branch. The scale branch is strictly positive elementwise and therefore only modulates magnitudes. See § F.3, especially Equations (60) and (61). □

**Theorem 2.5** (Budget asymmetry in residual two-layer blocks)**.** *Consider* $f(x) = x + \sigma(xW_1)W_2$ *with* $x \in \mathbb{R}^{1 \times d}$. *(i) If* $\mathrm{rank}(W_2) \leq r$, *then* $f(x) - x$ *lies in a fixed subspace of dimension at most* $r$ *for all* $x$. *(ii) If* $\mathrm{rank}(W_1) \leq r$, *then there exist* $L \in \mathbb{R}^{d \times r}$ *and* $\psi$ *such that* $f(x) = x + \psi(xL)$. *Hence, under a fixed parameter budget, shrinking the second-layer (VO) core directly constrains the update subspace, while shrinking the first-layer (QK) core primarily constrains conditioning/routing.*

*Proof sketch.* (1) is Appendix Theorem F.3 and (2) is Appendix Theorem F.5. See § F.2.1 and F.2.2. □

Here, we present a set of theoretical results that follow naturally from the attention-as-MLP perspective. These results provide direct explanations for several empirical findings that previously required extensive experimentation to uncover. Further discussion and connections to the corresponding prior literature are provided in the appendix.

**Proposition 2.6** (Dynamic-MLP view explains several empirical design principles)**.** *Write one head as a residual dynamic 2-layer map* $f(x; X) = x + \sigma(xW^{(1)}(X, x))W^{(2)}(X, x)$. *Then:*

*(i) Prefer* $QK$ *compression over* $VO$ *compression (budget allocation)* *(Mi et al., 2025; Bai et al., 2021). Shrinking/adapting the* readout/action side *(*$W^{(2)}$*, i.e.* $VO$*) directly restricts the update subspace, whereas shrinking* $W^{(1)}$ *(i.e.* $QK$*) primarily restricts routing/conditioning; see Theorem 2.5 (and Section F.2.1).*

*(ii) LoRA and gates are most parameter-efficient on the* readout side *(*$V/O$*) (Radiya-Dixit & Wang, 2020). A rank-*$r$ *adapter on* $W^{(2)}$ *yields*

$$\Delta o = \sigma(xW^{(1)}) \Delta W^{(2)} \in \mathrm{Row}(\Delta W^{(2)}), \quad \dim \leq r, \tag{21}$$

*so it adds up to* $r$ *new update directions. A* $V \rightarrow O$*-side gate changes only the readout map* $\beta(x; v)$ *while leaving the address/routing map* $\alpha(x; u)$ *(and thus the active-set geometry) unchanged; see Appendix Subsections H.6 and H.7 (and margin stability in Appendix Lemma D.11).*

*(iii) Linear attention collapses routing/selection (See Fig. 1(e))* *(Katharopoulos et al., 2020). Replacing activation by ungated normalization* $\sigma_{\mathrm{lin}}(z) = \mathrm{L2Norm}_t(z)$ *removes the pool→activated restriction: the output becomes a signed full-pool readout and the active set is always the full pool; see Appendix Proposition H.10.*

*(iv) Learnable registers enlarge the hidden pool* *(Darcet et al., 2024). Appending* $k$ *register rows* $R \in \mathbb{R}^{k \times d}$ *increases the hidden width from* $t$ *to* $t + k$ *and (without sequence mixing) adds exactly* $k$ *pool atoms:* $\mu_{X \| R}^{\mathrm{pool}} = \mu_X^{\mathrm{pool}} + \sum_{i=1}^k \delta_{(R[i,:], R[i,:])}$*; see Prop. H.5 (and Prop. C.8).*

*Proof sketch.* (i) is Theorem 2.5. (ii) uses the row-space restriction of multiplying by a rank-$r$ matrix and the measure form where $V/O$-side gating modifies only $\beta$ (§ H.6, H.7). (iii) is Prop. H.10. (iv) is Prop. H.5 plus Prop. C.8. □

### 2.6. Parameter budgeting and efficient implementation

**Parameter Allocation.** The analysis above clarifies the relative importance of parameters across different components. In practice, we still recommend sufficient-rank factorizations such as $d_{qk} \approx d_{vo} \approx d/n_{\mathrm{head}}$, which implies that HyperMLP will allocate a larger parameter share than vanilla attention in Transformer blocks due to the additional low-rank sequence-mixing operators $R^{(j)}$. For fair comparison in experiments, however, we match the parameter budget of vanilla attention. Following the attention-as-MLP view, we shrink the first-layer (QK) rank to "pay" for temporal mixing: from a total budget of $4d^2$, we allocate $3d^2$ to the second layer (VO) since the $M^{(2)}(x_t)$ gate adds an extra $d^2$ relative to standard VO, and we use the remaining $d^2$ for the first layer (QK) together with the low-rank sequence mixing $R$. Concretely, we set the rank of $L^{(1)}$ to $d/(4n_{\mathrm{head}})$ or $d/(8n_{\mathrm{head}})$ and use $r_s = 16$ to control the overhead.

**Complexity.** HyperMLP augments each head with two input-conditioned sequence-slot mixing operators $R_t^{(1)}(x_t)$ and $R_t^{(2)}(x_t)$ in diagonal-plus-low-rank (DPLR) form with rank $r_s$. These mixers can be applied without materializing any $t \times t$ matrices, incurring an additional $O(tr_s)$ time and $O(r_s)$ extra per-step state per head on top of the standard projection and aggregation costs. Over length-$T$ teacher forcing, this yields a total $O(T^2 r_s)$ overhead, so when $r_s \ll d_{qk}, d_{vo}$ the asymptotic training and autoregressive inference scaling matches quadratic attention up to lower-order terms. Since the additional complexity is per head, we recommend—and use in our experiments—fixing $n_{\mathrm{head}} = 2$. A formal proof is provided in Appendix G.2.

Introducing sequence mixing fundamentally changes the attention operator, so HyperMLP cannot directly reuse existing efficient backends such as FlashAttention (Dao et al.,

*Table 1.* Empirical results on the MAD benchmark (left) and NanoGPT OpenWebText2 training (right). The hyphenated `Label` encodes the model family and enabled mechanisms. Each row corresponds to one model variant, specified by `Label`: S/R/G denote Softmax, ReLU∘L2Norm, and GLU, respectively. Optional suffixes `p`, `c`, `g` denote RoPE, KV depthwise convolution, and QK/VO gating ($M^{(j)}(x_t)$), respectively. `q`/`v` indicate matched-budget compression on QK / VO. `1`, `2`, `12` indicate temporal mixing in $R^{(1)}$ / $R^{(2)}$ / both. `o` indicates the reverse-offset (lag) layout. `!` indicates over-parameterization (no compression). We fix the $n_{\text{head}}$ to 2 for fair comparison.

| Label | Compress | Fuzzy Recall | In-ctx Recall | Memorize Train Set | Noisy Recall | Selective Copy | MAD Avg | NanoGPT Loss (L5) | Steps loss<3.3 | Steps loss<3.2 | Steps loss<3.1 | Steps loss<3.0 |
|---|---|---|---|---|---|---|---|---|---|---|---|---|
| S (SoftmaxAttn) | 35.19 | 8.92 | 83.68 | 29.24 | 85.03 | 62.08 | 50.69 | 3.9487 | 18 | — | — | — |
| S-p | 44.69 | 34.74 | 93.43 | 78.65 | 86.32 | 96.42 | 72.38 | 3.0669 | 11 | 20 | 47 | — |
| S-c | 41.31 | 62.44 | 99.99 | 78.31 | 99.99 | 51.05 | 72.18 | 3.0897 | 15 | 27 | 64 | — |
| S-pc | 51.62 | 51.10 | 99.99 | 86.81 | 99.99 | 91.39 | 80.15 | 3.0466 | 11 | 19 | 39 | — |
| R (ReLUAttn) | 34.87 | 9.70 | 74.19 | 15.64 | 77.53 | 53.91 | 44.31 | 3.2573 | 50 | — | — | — |
| R-p | 43.35 | 31.38 | 93.65 | 68.83 | 92.81 | 97.34 | 71.23 | 3.0785 | 13 | 22 | 54 | — |
| R-c | 39.91 | 43.85 | 99.99 | 57.70 | 99.99 | 72.85 | 69.05 | 3.1070 | 17 | 31 | 80 | — |
| R-pc | 44.41 | 61.26 | 100.00 | 79.20 | 99.99 | 98.01 | 80.50 | 3.0481 | 11 | 19 | 37 | — |
| S-g-q | 39.08 | 51.52 | 99.99 | 78.51 | 99.99 | 54.88 | 70.66 | 3.0428 | 13 | 20 | 41 | — |
| R-cg-q | 36.49 | 32.57 | 100.0 | 57.70 | 99.99 | 73.90 | 66.78 | 3.0828 | 16 | 29 | 76 | 80 |
| R-c-12o! | 49.23 | 65.12 | 100.0 | 71.24 | 99.99 | 99.31 | 80.81 | 3.0590 | 10 | 17 | 36 | — |
| R-cg-q-12o (HyperMLP) | 49.20 | 62.76 | 99.99 | 74.97 | 99.99 | 99.14 | 81.01 | 2.9956 | 9 | 14 | 26 | 72 |
| R-pcg-q-12o | 47.86 | 36.22 | 99.99 | 66.06 | 99.99 | 99.29 | 74.90 | 3.0212 | 10 | 15 | 28 | 75 |
| S-c-q | 40.42 | 60.37 | 99.99 | 77.53 | 99.99 | 44.30 | 70.43 | 3.1042 | 16 | 29 | 78 | — |
| S-c-v | 33.57 | 57.87 | 99.99 | 77.23 | 99.99 | 52.89 | 70.26 | 3.1384 | 18 | 33 | — | — |
| R-c-q | 37.28 | 36.27 | 99.99 | 55.98 | 100.00 | 74.33 | 67.31 | 3.1762 | 22 | 55 | — | — |
| R-c-v | 31.84 | 43.47 | 100.00 | 65.58 | 99.99 | 70.36 | 68.54 | 3.1323 | 20 | 44 | 80 | — |
| G-cg-q | 38.56 | 35.84 | 99.99 | 66.73 | 99.99 | 75.23 | 69.39 | 3.0530 | 12 | 21 | 42 | — |
| G-c-12o! | 49.81 | 65.77 | 100.00 | 72.98 | 100.00 | 99.67 | 81.37 | 3.0159 | 9 | 15 | 28 | 80 |
| G-cg-q-12o (HyperGLU) | 49.60 | 61.50 | 99.99 | 76.07 | 99.99 | 99.58 | 81.12 | 2.9865 | 8 | 13 | 22 | 58 |
| G-pcg-q-12o | 49.90 | 29.12 | 99.98 | 70.35 | 99.99 | 99.28 | 74.77 | 2.9974 | 9 | 13 | 25 | 77 |
| G-cg-q-1o | 47.35 | 49.90 | 99.99 | 69.66 | 99.99 | 99.75 | 77.77 | 2.9907 | 9 | 13 | 24 | 68 |
| R-cg-q-1o | 47.34 | 57.69 | 99.99 | 70.62 | 99.99 | 98.97 | 79.10 | 3.0001 | 9 | 14 | 24 | 75 |
| G-g-q-12o | 50.49 | 58.88 | 99.99 | 76.95 | 99.97 | 99.25 | 80.92 | 2.9980 | 9 | 14 | 24 | 74 |
| R-g-q-12o | 49.14 | 61.98 | 99.99 | 73.55 | 99.97 | 99.06 | 80.61 | 3.0130 | 10 | 16 | 30 | — |
| G-12o! | 49.65 | 49.56 | 99.97 | 72.38 | 99.85 | 99.72 | 78.52 | 3.0386 | 12 | 17 | 33 | — |
| G-1o! | 47.15 | 32.11 | 99.96 | 71.72 | 99.96 | 99.67 | 75.09 | 3.0556 | 12 | 20 | 39 | — |
| G-2o! | 48.07 | 68.35 | 99.95 | 52.55 | 99.97 | 96.52 | 77.57 | 3.0631 | 16 | 27 | 55 | — |
| R-12o! | 49.36 | 61.37 | 99.95 | 82.01 | 99.97 | 98.28 | 81.82 | 3.0497 | 11 | 17 | 36 | — |
| R-1o! | 44.70 | 60.51 | 99.94 | 77.17 | 99.67 | 99.43 | 80.24 | 3.0699 | 12 | 20 | 47 | — |
| R-2! | 45.28 | 64.03 | 99.97 | 75.75 | 99.85 | 96.05 | 80.16 | 3.1022 | 17 | 30 | 72 | — |
| G-12! | 47.44 | 6.81 | 55.75 | 16.45 | 54.87 | 95.74 | 46.18 | 4.3662 | — | — | — | — |
| R-12! | 47.71 | 8.02 | 55.60 | 16.28 | 55.27 | 94.78 | 46.27 | 4.3567 | — | — | — | — |

2022). Achieving FlashAttention-level performance would require extensive operator fusion and hardware-aware kernel design, which is beyond the scope of this work. Accordingly, we focus on algorithmic expressivity rather than engineering optimization. Nevertheless, we provide a minimal, efficient block-wise implementation of HyperMLP (Fig. 3(d–e), § J, and the code repository) that computes only the necessary blocks and leverages efficient GEMM operations after compilation. On modern hardware, this implementation achieves performance comparable in order of magnitude to many recent expressive attention variants. See Table 5 and §K for the detailed comparison of computational costs.

## 3. Empirical Evaluation

**3.a. Controlled Design Study** Before comparing Hyper-MLP/HyperGLU with strong external baselines, we conduct a Controlled Design Study to isolate the effect of individual design choices under fixed budgets. We report results on two complementary sources: (i) MAD (Poli et al., 2024), a set of mechanistic diagnostics, and (ii) GPT-2 structure NanoGPT (Karpathy, 2022) language modeling on OpenWebText2, reported by final loss (last-5 average) and simple training-progress statistics. For NanoGPT, we use the default settings and evaluate every 500 training steps; all reported "steps" refer to these evaluations, with loss measured on the sam-

pled training subset. We train all of the settings for 80 steps. **See Fig. 5 for detailed loss curves**. See §M-N for details of implementations and datasets.

Each configuration is denoted by a compact label `Base-Feat-Rank-TmixLayout`, which explicitly encodes the active mechanisms. `Base` ∈ {S, R, G} specifies softmax attention, ReLU attention (ReLU∘L2Norm), or GLU-style gating. `Feat` optionally includes RoPE (`p`), KV depthwise convolution (`c`), and QK/VO-side gating (`g`). `Rank` appends `q` or `v` to indicate matched-budget compression on the QK or VO side. `TmixLayout` uses `1`, `2`, or `12` to denote temporal mixing in $R^{(1)}$, $R^{(2)}$, or both (DPLR), with an optional `o` indicating the reverse-offset (lag) layout.

Unless marked with `!`, all runs use the same training setup and strictly matched parameter budgets, so differences reflect operator structure rather than scale. Runs with `!` are over-parameterized upper bounds that relax the budget constraint and are used only to estimate headroom. The label system enables controlled, local comparisons—varying one factor at a time (e.g., feature mods, QK vs. VO compression, or temporal mixing layout)—providing a clean attribution of empirical effects in the following section.

Please note that we still use depthwise convolution in the default HyperMLP/HyperGLU, since it introduces very little additional parameter and computational cost, and when

*Table 2.* Unified language modeling evaluation results across model families and scales. Abbr: Acc_n=normalized accuracy; EM=exact match; IFE-I/P = IFEval (Inst/Prompt, strict only). Shots: MMLU-P=5, GPQA=0, BBH=3, MATH=4, MuSR=0; others 0-shot. **Bold** and underline indicate group-wise best and second-best results, respectively. See Appendix L.10 for the head-count settings used by each model, and Appendix L.11 for the rationale behind the per-scale baseline choices.

| Model
*variant* | Open LLM Leaderboard | | | | | | | General Ability | | | | | | | | | Ranking | |
|---|---|---|---|---|---|---|---|---|---|---|---|---|---|---|---|---|---|---|
| | MMLU-P
(Acc↑) | GPQA
(Acc_n↑) | BBH
(Acc_n↑) | MATH
(EM↑) | MuSR
(Acc_n↑) | IFE-I
(strict↑) | IFE-P
(strict↑) | ARC-C
(Acc_n↑) | ARC-E
(Acc_n↑) | HS
(Acc_n↑) | PIQA
(Acc_n↑) | BoolQ
(Acc↑) | WinoG
(Acc↑) | COPA
(Acc↑) | OBQA
(Acc_n↑) | SciQ
(Acc↑) | Avg
Rank↓ | #Top1
↑ |
| **1.3B Params – 100B Tokens** | | | | | | | | | | | | | | | | | | |
| DeltaNet | 0.109 | 0.263 | 0.308 | 0.011 | 0.417 | 0.288 | 0.165 | 0.266 | 0.522 | 0.502 | 0.704 | 0.611 | 0.541 | 0.740 | 0.318 | 0.761 | 4.22 | 0 |
| GSA | 0.110 | **0.270** | 0.294 | **0.013** | 0.438 | **0.300** | 0.179 | 0.287 | 0.529 | 0.510 | **0.712** | 0.541 | 0.536 | 0.760 | 0.330 | 0.773 | 2.84 | 4 |
| RetNet | 0.110 | 0.252 | 0.293 | 0.001 | 0.384 | 0.056 | 0.024 | 0.271 | 0.489 | 0.480 | 0.701 | 0.583 | 0.533 | 0.710 | 0.324 | 0.736 | 6.69 | 0 |
| HGRN | 0.114 | 0.269 | 0.297 | 0.008 | 0.409 | 0.253 | 0.122 | 0.271 | 0.518 | 0.481 | 0.707 | 0.584 | 0.515 | 0.700 | 0.326 | 0.695 | 5.12 | 0 |
| HGRN2 | 0.115 | 0.254 | 0.295 | 0.002 | 0.350 | 0.223 | 0.129 | 0.282 | 0.504 | 0.317 | 0.671 | 0.416 | 0.522 | **0.770** | 0.328 | 0.378 | 5.84 | 1 |
| SoftmaxAttn | 0.114 | 0.259 | 0.296 | 0.011 | 0.365 | 0.270 | 0.141 | 0.280 | 0.492 | 0.492 | 0.705 | **0.621** | **0.552** | 0.760 | 0.318 | 0.769 | 4.22 | 2 |
| GLA | 0.114 | 0.259 | 0.295 | 0.006 | 0.427 | 0.272 | 0.157 | 0.277 | 0.482 | 0.488 | 0.702 | 0.574 | 0.541 | 0.690 | 0.326 | 0.721 | 5.19 | 0 |
| **HyperGLU** | **0.120** | 0.268 | **0.321** | 0.012 | **0.468** | 0.286 | **0.189** | **0.366** | **0.611** | **0.516** | 0.708 | 0.613 | 0.531 | 0.760 | **0.353** | **0.788** | **1.88** | **9** |
| **340M Params – 15B Tokens** | | | | | | | | | | | | | | | | | | |
| DiffTrans | 0.109 | 0.259 | 0.299 | 0.008 | 0.390 | 0.266 | 0.133 | 0.289 | 0.531 | 0.408 | 0.668 | **0.603** | **0.534** | 0.690 | 0.330 | 0.734 | 4.13 | 2 |
| GatedDeltaNet | 0.113 | 0.260 | 0.296 | **0.010** | 0.421 | 0.258 | 0.133 | 0.276 | 0.527 | 0.396 | 0.662 | 0.588 | 0.527 | 0.710 | **0.338** | 0.735 | 3.81 | 2 |
| DeltaNet | 0.112 | 0.260 | 0.300 | 0.009 | 0.452 | **0.277** | 0.150 | 0.269 | 0.502 | 0.405 | 0.653 | 0.519 | 0.504 | 0.690 | 0.316 | 0.717 | 4.81 | 1 |
| GLA | 0.110 | 0.258 | 0.289 | 0.007 | 0.415 | 0.228 | 0.109 | 0.247 | 0.478 | 0.366 | 0.637 | 0.547 | 0.489 | 0.640 | 0.294 | 0.649 | 7.56 | 0 |
| SoftmaxAttn | 0.106 | **0.267** | 0.292 | **0.010** | 0.386 | 0.269 | 0.126 | 0.273 | 0.506 | 0.396 | 0.650 | 0.569 | 0.499 | **0.720** | 0.324 | 0.727 | 5.28 | 3 |
| **ReLUAttn+KVConv** | 0.111 | 0.253 | 0.290 | 0.008 | 0.423 | 0.229 | 0.130 | 0.268 | 0.525 | 0.412 | 0.669 | 0.524 | 0.520 | 0.660 | 0.324 | **0.774** | 5.13 | 0 |
| **HyperMLP** | **0.119** | 0.262 | 0.300 | **0.010** | **0.466** | 0.273 | 0.155 | 0.288 | **0.541** | 0.402 | 0.663 | 0.585 | 0.513 | 0.710 | 0.320 | 0.755 | 2.97 | 4 |
| **HyperGLU** | 0.117 | 0.259 | **0.313** | 0.009 | 0.451 | 0.273 | **0.157** | **0.304** | 0.538 | **0.427** | **0.674** | 0.587 | 0.518 | **0.720** | 0.336 | 0.771 | **2.31** | **6** |

used together with our low-rank sequence mixing it can still provide a performance boost. In practice, we recommend combining it with DPLR mixing. Consistent with our attention-as-MLP perspective, we apply convolutions only to the two cores $X_{t:1}$; We do the convolution first before we perform $R$ mixing, and we do not apply any mixing to the input $x_t$. In our experiments, we compare variants with (HyperMLP/HyperGLU) and without convolution (R-g-q-12o, G-g-q-12o). The results show that Hyper variants without convolution still significantly outperform ReLU/softmax attention across all metrics.

**Result analysis. (1) Softmax is not essential:** under standard feature-side upgrades, ReLU attention matches softmax at essentially the same quality–efficiency point. In particular, S-pc and R-pc achieve comparable MAD avg (80.15 vs. 80.50) and NanoGPT loss (3.0466 vs. 3.0481), suggesting that probability-simplex normalization is not required for strong performance. **(2) Sequence-space mixing is the dominant gain:** enabling temporal mixing in the score/sequence dimension yields a large jump. With the same feature modifiers and QK-side compression, adding DPLR mixing with lag layout improves R-cg-q → R-cg-q-12o from MAD avg 66.78 to 81.01 (+14.23), with particularly large gains on retrieval/copy behaviors (e.g., Fuzzy Recall 32.57 → 62.76, Selective Copy 73.90 → 99.14). NanoGPT also improves (loss 3.0828→2.9956) and reaches low-loss thresholds substantially earlier (e.g., loss<3.1: 76 → 26 steps), consistent with a strictly richer dynamic-MLP function class from learned context-wide slots. **(3) Lag layout is necessary for AR consistency:** temporal mixing without the reverse-offset (lag) layout collapses. The over-parameterized ablations show a stark failure without o: R-12! has MAD avg 46.28 and NanoGPT loss 4.3567, while R-12o! recovers to MAD avg 81.82 and loss 3.0497 (similarly G-12!:

46.18/4.3662 vs. G-12o!: 78.52/3.0386). This supports the theory that extension-consistent temporal operators must align with AR truncation semantics. **(4) Two-sided mixing helps:** using mixing in both $R^{(1)}$ and $R^{(2)}$ is generally better than mixing only one side. For instance, G-cg-q-12o outperforms G-cg-q-1o on MAD avg (81.12 vs. 77.77) with slightly lower NanoGPT loss (2.9865 vs. 2.9907), matching the intuition that routing-side and readout-side temporal mixing provide complementary flexibility. **(5) Budget asymmetry (QK vs. VO) is visible in-task:** VO compression more directly harms tasks that require diverse update directions. This matches the predicted update-subspace bottleneck from shrinking the readout rank. **(6) HyperGLU is better:** the best matched-budget configuration is the GLU variant G-cg-q-12o, which achieves the lowest NanoGPT loss (2.9865) while matching the top MAD performance (avg 81.12), consistent with decoupling routing (active set) from magnitude modulation.

**3.b. Language Modeling.** We follow the experimental protocol of (Yang et al., 2024a;c). Under identical training conditions, we train autoregressive language models with **1.3B** and **340M** parameters on the FineWeb-Edu dataset (Penedo et al., 2024), using **100B** and **15B** training tokens, respectively. All models are optimized with AdamW using a peak learning rate of $4 \times 10^{-4}$, cosine decay with a 1B-token warmup, weight decay of 0.1, and gradient clipping at 1.0, with a global batch size of 0.5M tokens. We use the LLaMA-2 tokenizer with a 32K vocabulary and train with a context length of 4096. Evaluation follows the Open LLM Leaderboard together with a standard suite of general-ability benchmarks, summarized in Tab. 2. Additional evaluation details are provided in §N. See the description of the included baselines in §M.

Across unified language-model evaluations, ReLU atten-

tion (ReLUAttn+KVConv) is already a strong non-softmax baseline, but HyperMLP and especially HyperGLU deliver consistent, high-level gains under matched parameter budgets. At 340M / 15B tokens, HyperMLP/HyperGLU improve overall standings versus the ReLU-attention baseline, indicating broadly better generalization across tasks rather than isolated wins. At the larger 1.3B / 100B tokens scale, HyperGLU further separates from softmax attention and other competitive sequence models, achieving the best aggregate ranking and the most category bests, suggesting the learned sequence mixing plus GLU-style routing yields robust improvements.

## 4. Conclusion and Limitations

We reframe autoregressive attention as a dynamic two-layer MLP whose context-instantiated scores form an ever-growing hidden representation. Based on this view, we propose HyperMLP/HyperGLU, which learns input-conditioned mixing in both feature and sequence spaces with a reverse-offset (lag) layout and low-rank parameterization. Across controlled studies and language-model evaluations under matched budgets, HyperMLP/HyperGLU consistently outperform strong softmax-attention Transformers, highlighting the benefits of learned temporal mixing and MLP-style routing. Although we provide a minimal efficient implementation, it cannot match highly optimized kernels such as FlashAttention. Additionally, due to resource constraints, we do not scale HyperMLP/HyperGLU to very large practical LLM sizes; validating performance at frontier scales is left for future work.

## Acknowledgements

This research was supported in part through research cyberinfrastructure resources and services provided by the Partnership for an Advanced Computing Environment (PACE, 2017) at the Georgia Institute of Technology, Atlanta, Georgia, USA.

This work used DeltaAI at the National Center for Supercomputing Applications through allocation MTH250051 from the Advanced Cyberinfrastructure Coordination Ecosystem: Services & Support (ACCESS) program (Boerner et al., 2023), which is supported by National Science Foundation grants #2138259, #2138286, #2138307, #2137603, and #2138296.

We thank Lambda, Inc. for their compute resource help.

## Impact Statement

This paper proposes a new architectural perspective and attention variant for autoregressive sequence models. As a fundamental modeling component, HyperMLP/HyperGLU primarily affects upstream model expressivity rather than any specific downstream application. We conduct experiments only on publicly available benchmarks and datasets, without collecting or using personal or sensitive data. While improved sequence modeling and retrieval capabilities could be misused in downstream systems (e.g., for surveillance or misleading content), such risks depend on deployment choices beyond the scope of this work. We therefore emphasize responsible use, including appropriate evaluation, bias mitigation, privacy safeguards, and energy-aware training practices.

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

# Appendix

## A. Overview of the Supporting Theoretical Results in the Appendix

In figure 4, we provide a logical map that organizes the appendix into three intertwined threads: (i) Mechanism & analysis (left) develops the dynamic-MLP and three-stage memory view (Global $\rightarrow$ Pool $\rightarrow$ Act), including gate invariance, measure/TV formulations, and NTK-style stability; (ii) Geometry & function classes (right) analyzes static vs. dynamic partition geometry, showing how sequence-space mixing yields warped routing and strictly richer function classes than token-wise polyhedral ReLU attention; (iii) Parameterization & efficiency (bottom) covers the lag layout, DPLR temporal mixing, rank allocation (shrinking QK rather than VO), HyperGLU, and the complexity/overhead bounds. Each numbered box (M1–M8, G1–G5, D1–D6) points to the precise lemmas, propositions, and theorems that support the main text.

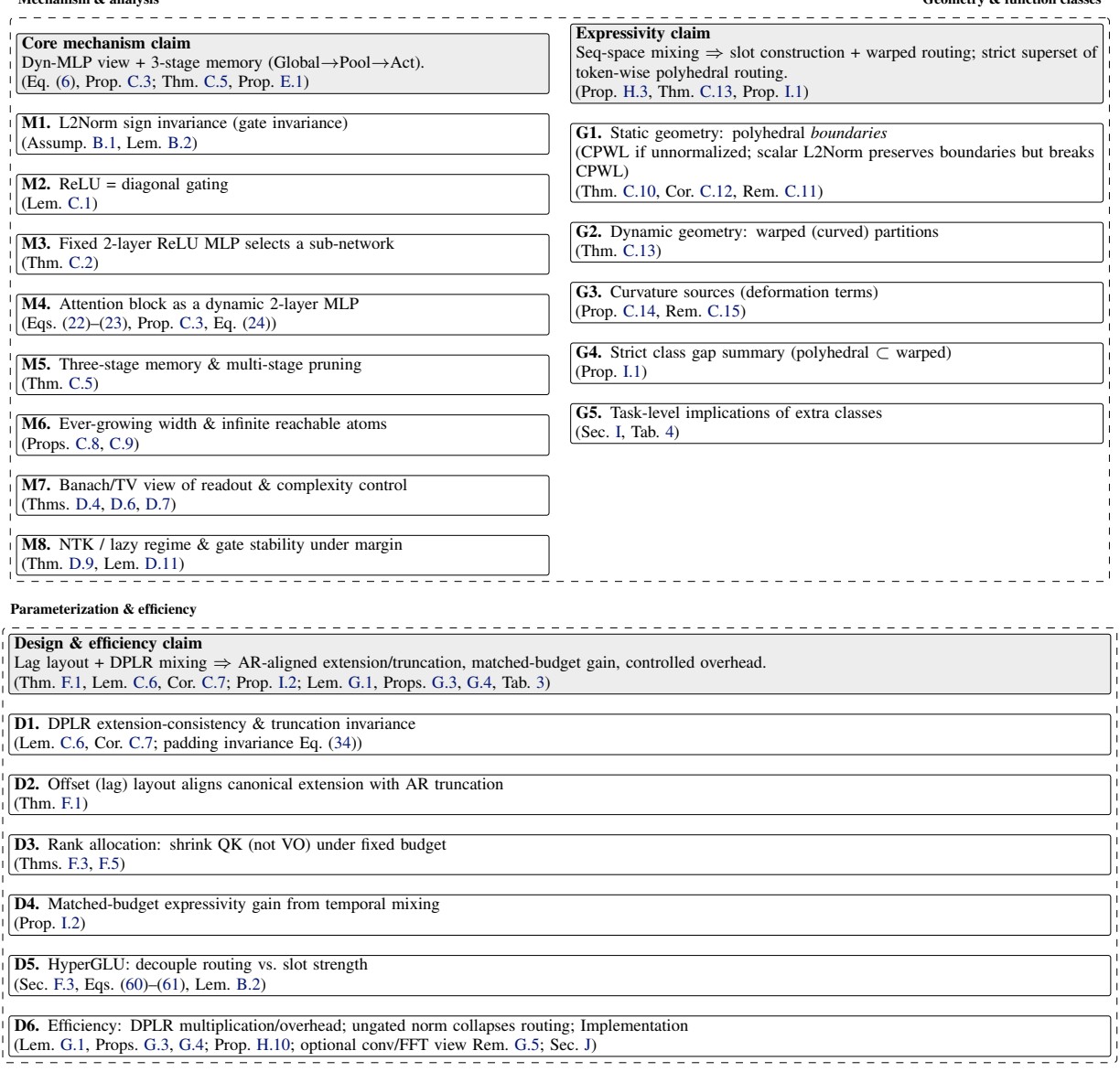

*Figure 4.* Logical map of our theoretical results.

# B. Notation and Common Setup

**Row-vector convention and indexing.** We represent tokens/states as row vectors. For $d \in \mathbb{N}$, $x \in \mathbb{R}^{1 \times d}$ and a length-$t$ context is $X \in \mathbb{R}^{t \times d}$. We write $[t] := \{1, \dots, t\}$ and denote the $i$-th row of $X$ by $X[i, :]$.

**Lag (reverse-offset) layout.** In autoregressive use at step $t$, we take the lag-ordered prefix

$$X_{t:1} := [x_t; x_{t-1}; \dots; x_1] \in \mathbb{R}^{t \times d}.$$

Unless stated otherwise we abbreviate $X := X_{t:1}$ and $x := x_t$.[4]

**One-head dynamic block.** Fix an output dimension $d$. A single head is specified by feature-space mixing maps $L^{(1)}(\cdot), L^{(2)}(\cdot) : \mathbb{R}^{1 \times d} \to \mathbb{R}^{d \times d}$ and sequence-space mixing maps $R_t^{(1)}(\cdot), R_t^{(2)}(\cdot) : \mathbb{R}^{1 \times d} \to \mathbb{R}^{t \times t}$. Given $(x, X) \in \mathbb{R}^{1 \times d} \times \mathbb{R}^{t \times d}$, define

$$h_t(x; X) := x \, L^{(1)}(x) \, X^\top \, R_t^{(1)}(x) \in \mathbb{R}^{1 \times t}, \tag{22}$$

$$o_t(x; X) := \sigma_t\big(h_t(x; X)\big) \, R_t^{(2)\top}(x) \, X \, L^{(2)\top}(x) \in \mathbb{R}^{1 \times d}. \tag{23}$$

We also use the effective (dynamically instantiated) matrices

$$W_t^{(1)}(X, x) := L^{(1)}(x) \, X^\top \, R_t^{(1)}(x) \in \mathbb{R}^{d \times t}, \qquad W_t^{(2)}(X, x) := R_t^{(2)\top}(x) \, X \, L^{(2)\top}(x) \in \mathbb{R}^{t \times d}. \tag{24}$$

Then $h_t = x W_t^{(1)}(X, x)$ and $o_t = \sigma_t(x W_t^{(1)}(X, x)) \, W_t^{(2)}(X, x)$.

**Activation and L2 normalization.** Throughout the appendix we take

$$\sigma_t(z) := \mathrm{ReLU}\big(\mathrm{L2Norm}_t(z)\big), \qquad z \in \mathbb{R}^{1 \times t}. \tag{25}$$

Here $\mathrm{L2Norm}_t$ is affine-free across the length-$t$ vector (no per-coordinate gain/bias).

**Assumption B.1** (Positive scalar L2 normalization across $t$). For each $t \geq 1$, there exists $\rho_t : \mathbb{R}^{1 \times t} \to (0, \infty)$ such that

$$\mathrm{L2Norm}_t(z) = \frac{z}{\rho_t(z)}.$$

When we need a concrete choice, we use $\rho_t(z) = \sqrt{\|z\|_2^2 + \varepsilon}$ with $\varepsilon > 0$.

**Lemma B.2** (Gate invariance under scalar L2 normalization). *Under Assumption B.1,*

$$\mathbf{1}\{\mathrm{L2Norm}_t(z) > 0\} = \mathbf{1}\{z > 0\}, \qquad \mathrm{ReLU}(\mathrm{L2Norm}_t(z)) = \frac{1}{\rho_t(z)} \, \mathrm{ReLU}(z).$$

**Context instantiation and slot notation.** Define the context-instantiation operators

$$\mathcal{E}_t^{(j)}(x)[X] := R_t^{(j)\top}(x) \, X \in \mathbb{R}^{t \times d}, \qquad j \in \{1, 2\}. \tag{26}$$

Write

$$U := \mathcal{E}_t^{(1)}(x)[X], \qquad V := \mathcal{E}_t^{(2)}(x)[X], \qquad u_i := U[i, :], \; v_i := V[i, :] \; (i \in [t]).$$

---

[4]In autoregressive inference $x$ is the newest row of $X$. We sometimes write $(X, x)$ only to expose the factorization of dynamic weights; in analyses that fix $X$ and vary $x$, this means varying only the most recent row.

**Measure/dictionary notation.** Let $\Omega := \mathbb{R}^d \times \mathbb{R}^d$ with $\omega = (u, v) \in \Omega$. Define the addressing and readout maps

$$\alpha(x; u) := x\, L^{(1)}(x)\, u^\top \in \mathbb{R}, \qquad \beta(x; v) := v\, L^{(2)\top}(x) \in \mathbb{R}^{1 \times d}. \tag{27}$$

Define the induced dictionary element

$$\Psi(x; (u, v)) := \alpha(x; u)_+\, \beta(x; v) \in \mathbb{R}^{1 \times d}, \qquad a_+ := \max\{0, a\}. \tag{28}$$

The Context-instantiated Memory Pool is the atomic measure

$$\mu_{X,x}^{\mathrm{pool}} := \sum_{i=1}^{t} \delta_{(u_i, v_i)} \qquad \text{on } \Omega. \tag{29}$$

The hard gate is $g_x(u, v) := \mathbf{1}\{\alpha(x; u) > 0\}$ and the Current-step Activated Memory is the restricted measure

$$\mu_{X,x}^{\mathrm{act}} := g_x\, \mu_{X,x}^{\mathrm{pool}}. \tag{30}$$

We freely identify finite sums with integrals against $\mu_{X,x}^{\mathrm{pool}}$, e.g., $\int f \, d\mu_{X,x}^{\mathrm{pool}} = \sum_{i=1}^{t} f(u_i, v_i)$. (Here $\mu_{X,x}^{\mathrm{pool}}$ and $\mu_{X,x}^{\mathrm{act}}$ are finite atomic measures, so the integral notation is purely shorthand for a finite sum; for vector-valued $f$ it is understood componentwise.)

## C. Redesigning Attention from First Principles: Theoretical Supports

Our methodology starts from a minimal observation: ReLU networks compute by selecting a sub-network conditioned on the input. We then re-interpret autoregressive ReLU attention as a dynamic two-layer ReLU MLP whose hidden width grows with the context length, and we formalize its memory behavior as a multi-stage pruning process: Global (parameter-defined) Memory Space $\rightarrow$ Context-instantiated Memory Pool $\rightarrow$ Current-step Activated Memory. This section provides the formal backbone for these statements, using the matrix-chain notation of our model.

### C.1. ReLU as input-conditioned sub-network selection

We start from a depth-two ReLU MLP and omit biases as in common LLM practice:

$$o = \mathrm{ReLU}\big(x W_{\mathrm{MLP}}^{(1)}\big)\, W_{\mathrm{MLP}}^{(2)}, \qquad x \in \mathbb{R}^{1 \times d}. \tag{31}$$

ReLU acts as diagonal gating (Lemma C.1): for any row vector $z \in \mathbb{R}^{1 \times m}$,

$$D(z) := \mathrm{diag}\big(\mathbf{1}\{z_1 > 0\}, \ldots, \mathbf{1}\{z_m > 0\}\big) \in \mathbb{R}^{m \times m},$$

so $\mathrm{ReLU}(z) = z D(z)$. Each input $x$ therefore induces a binary mask $D_x := D(x W_{\mathrm{MLP}}^{(1)})$ and

$$o = x\, W_{\mathrm{MLP}}^{(1)}\, D_x\, W_{\mathrm{MLP}}^{(2)}. \tag{32}$$

Hence a fixed two-layer ReLU MLP can be viewed as learning a family of linear sub-networks that are input-conditioned: on any region of input space where the sign pattern (and thus $D_x$) is constant, the map reduces to a single linear transformation, while zero entries of $D_x$ deactivate the corresponding hidden units and prune their incident connections (Saxe et al., 2022).

**Lemma C.1** (ReLU is diagonal gating). *For any row vector $z \in \mathbb{R}^{1 \times m}$, define*

$$D(z) := \mathrm{diag}(\mathbf{1}\{z_1 > 0\}, \ldots, \mathbf{1}\{z_m > 0\}) \in \mathbb{R}^{m \times m}.$$

*Then*

$$\mathrm{ReLU}(z) = z\, D(z).$$

*Proof.* For each coordinate $i$, if $z_i > 0$ then $(zD(z))_i = z_i = \max(0, z_i)$; otherwise $(zD(z))_i = 0 = \max(0, z_i)$.  □

**Theorem C.2** (Fixed 2-layer ReLU MLP selects a sub-network). *Consider a two-layer ReLU MLP*

$$f(x) = \text{ReLU}(xW_1 + b_1)\, W_2 + b_2, \qquad x \in \mathbb{R}^{1 \times d}.$$

*For every input $x$, define the diagonal binary mask*

$$D_x := D(xW_1 + b_1) \in \mathbb{R}^{h \times h},$$

*where $D(\cdot)$ is as in Lemma C.1. Then*

$$f(x) \;=\; (xW_1 + b_1)\, D_x\, W_2 + b_2 \;=\; xW_1 D_x W_2 \;+\; b_1 D_x W_2 \;+\; b_2.$$

*Moreover, on any region where $D_x$ is constant, $f$ reduces to an affine map.*

*Proof.* Apply Lemma C.1 with $z = xW_1 + b_1$ to obtain $\text{ReLU}(xW_1 + b_1) = (xW_1 + b_1)D_x$ and substitute into $f$. If $D_x \equiv D_0$ on a region, then $f(x) = xW_1 D_0 W_2 + (b_1 D_0 W_2 + b_2)$ is affine there, and any zero diagonal entry of $D_0$ forces the corresponding hidden unit output to be 0 (hence its incident connections contribute 0).  □

## C.2. Attention as a dynamic two-layer ReLU MLP

**Notation.** We use the common setup in Appendix B. In particular, the one-head block is defined by (22)–(23) with activation (25).

**Proposition C.3** (Dynamic two-layer form). *Define the effective (dynamically instantiated) weights*

$$W^{(1)}(X, x_t) \;:=\; L^{(1)}(x_t)\, X^\top\, R^{(1)}(x_t) \in \mathbb{R}^{d \times t}, \qquad W^{(2)}(X, x_t) \;:=\; R^{(2)\top}(x_t)\, X\, L^{(2)\top}(x_t) \in \mathbb{R}^{t \times d}.$$

*Then* (22)–(23) *are equivalent to*

$$o_t \;=\; \sigma\big(x_t\, W^{(1)}(X, x_t)\big)\, W^{(2)}(X, x_t).$$

*Hence, for fixed $X$, the computation $x_t \mapsto o_t$ can be written in the algebraic form of a width-$t$ two-layer ReLU network with dynamically instantiated (input-conditioned) weights.*

*Proof.* Substitute the definitions of $W^{(1)}(X, x_t)$ and $W^{(2)}(X, x_t)$ into (22)–(23).  □

## C.3. Context instantiation and multi-stage pruning

The dynamic two-layer view exposes an explicit memory interpretation. We first define the context-instantiation operators induced by the sequence-space mixings.

**Notation.** We recall $\mathcal{E}_t^{(j)}$, the mixed contexts $(U, V)$, the slot rows $(u_i, v_i)$, and the measures $(\mu_{X,x}^{\text{pool}}, \mu_{X,x}^{\text{act}})$ from Appendix B. We also adopt Assumption B.1, so L2 normalization preserves the ReLU gate pattern (Lemma B.2).

**L2 normalization vs. probability normalization.** Throughout this work, $\text{L2Norm}_t$ denotes the affine-free L2 normalization applied across the length-$t$ vector (i.e., no elementwise gain/bias). With this convention, $\text{L2Norm}_t$ acts as a positive scalar rescaling and therefore preserves sign patterns.

**Assumption C.4** (Scalar positive L2 normalization across $t$). For each $t \geq 1$, there exists $\rho_t : \mathbb{R}^{1 \times t} \to (0, \infty)$ such that

$$\text{L2Norm}_t(z) \;=\; \frac{z}{\rho_t(z)}.$$

Under this assumption, normalization does not change which paths are activated (the sign pattern); it only rescales their magnitudes. In particular, for the activation $\sigma_t(\cdot) := \text{ReLU}(\text{L2Norm}_t(\cdot))$ we have the equivalent form

$$\sigma_t(z) \;=\; \frac{1}{\rho_t(z)}\, \text{ReLU}(z),$$

which we will use below without repeatedly expanding $\text{L2Norm}_t(\cdot)$.

**Theorem C.5** (Three-stage memory and multi-stage pruning)**.** *Under Assumption B.1, the output* (23) *admits the representations*

$$
o_t \;=\; \frac{1}{\rho_t(h_t)} \int_\Omega \alpha(x_t; u)\, \beta(x_t; v)\, d\mu_{X,x_t}^{\mathrm{act}}(u, v) \;=\; \frac{1}{\rho_t(h_t)} \int_\Omega \alpha(x_t; u)_+ \, \beta(x_t; v)\, d\mu_{X,x_t}^{\mathrm{pool}}(u, v).
$$

*Equivalently, $o_t$ is a readout from the Current-step Activated Memory obtained by restricting the Context-instantiated Memory Pool.*

*Moreover, in terms of supports (set-valued, ignoring multiplicity), we have the pruning chain*

$$
\Omega \;\supset\; \mathrm{supp}\big(\mu_{X,x_t}^{\mathrm{pool}}\big) \;\supset\; \mathrm{supp}\big(\mu_{X,x_t}^{\mathrm{act}}\big),
$$

*corresponding to: Global (parameter-defined) Memory Space $\to$ Context-instantiated Memory Pool $\to$ Current-step Activated Memory.*

*Proof.* By Assumption B.1, $\mathrm{L2Norm}_t(h_t) = h_t/\rho_t(h_t)$ with $\rho_t(h_t) > 0$, hence

$$
\sigma_t(h_t) = \mathrm{ReLU}\left(\frac{h_t}{\rho_t(h_t)}\right) = \frac{1}{\rho_t(h_t)}\,\mathrm{ReLU}(h_t) = \frac{1}{\rho_t(h_t)}\big(h_t \odot \mathbf{1}\{h_t > 0\}\big).
$$

Using $(h_t)_i = \alpha(x_t; u_i)$ and $\beta(x_t; v_i) = v_i L^{(2)\top}(x_t)$, we obtain

$$
o_t = \sum_{i=1}^{t} \sigma_t(h_t)_i\, \beta(x_t; v_i) = \frac{1}{\rho_t(h_t)} \sum_{i=1}^{t} \alpha(x_t; u_i)\, \mathbf{1}\{\alpha(x_t; u_i) > 0\}\, \beta(x_t; v_i).
$$

Writing the finite sum as an integral against $\mu_{X,x_t}^{\mathrm{pool}} = \sum_{i=1}^{t} \delta_{(u_i, v_i)}$ and absorbing the indicator into the restricted measure $\mu_{X,x_t}^{\mathrm{act}} = g_{x_t} \mu_{X,x_t}^{\mathrm{pool}}$ yields the first boxed formula. The second boxed formula follows from the identity $\alpha\,\mathbf{1}\{\alpha > 0\} = \alpha_+$. The support inclusions follow immediately from the definitions of $\mu^{\mathrm{pool}}$ and $\mu^{\mathrm{act}}$. $\qquad\square$

**Lemma C.6** (DPLR temporal-mixing parameterization of $R_t^{(m)}(x)$ admits prefix-extension consistency)**.** *Fix integers $1 \le t < T$ and let*

$$
P_{t \to T} := \begin{bmatrix} I_t \\ 0 \end{bmatrix} \in \mathbb{R}^{T \times t}.
$$

*Let $s(x) := \phi(xW_S) \in \mathbb{R}^{r_s}$ and $S(x) := \mathrm{Diag}(s(x)) \in \mathbb{R}^{r_s \times r_s}$, where $\phi$ is any fixed nonlinearity (e.g. sigmoid).*

*Given any length-$t$ DPLR parameters*

$$
p_t \in \mathbb{R}^t, \qquad A_t, B_t \in \mathbb{R}^{t \times r_s}, \qquad D_t := I_t + \mathrm{Diag}(p_t),
$$

*define the length-$T$ extension by*

$$
p_T := \begin{bmatrix} p_t \\ -\mathbf{1}_{T-t} \end{bmatrix} \in \mathbb{R}^T, \qquad A_T := P_{t \to T} A_t \in \mathbb{R}^{T \times r_s}, \qquad B_T := P_{t \to T} B_t \in \mathbb{R}^{T \times r_s}, \qquad D_T := I_T + \mathrm{Diag}(p_T).
$$

*Define*

$$
R_t(x) := D_t + A_t\, S(x)\, B_t^\top \in \mathbb{R}^{t \times t}, \qquad R_T(x) := D_T + A_T\, S(x)\, B_T^\top \in \mathbb{R}^{T \times T}.
$$

*Then for all $x$,*

$$
R_T(x) \;=\; P_{t \to T}\, R_t(x)\, P_{t \to T}^\top.
$$

*Proof.* First, since $p_{t+1:T} = -\mathbf{1}$, the diagonal matrix $D_T = I_T + \mathrm{Diag}(p_T)$ has diagonal entries $(1 + p_{t,1}, \ldots, 1 + p_{t,t}, 0, \ldots, 0)$, hence

$$
D_T = P_{t \to T} D_t P_{t \to T}^\top.
$$

Second,

$$
A_T S(x) B_T^\top = (P_{t \to T} A_t)\, S(x)\, (P_{t \to T} B_t)^\top = P_{t \to T}\big(A_t S(x) B_t^\top\big) P_{t \to T}^\top.
$$

Adding the two identities yields $R_T(x) = P_{t \to T} R_t(x) P_{t \to T}^\top$. $\qquad\square$

**Corollary C.7** (Extension consistency, monotone expressivity, and non-instantiated pool slots). *Fix integers $1 \leq t < T$. For each length $\ell \in \{t, T\}$, let $F_\ell : \mathbb{R}^{1 \times d} \times \mathbb{R}^{\ell \times d} \to \mathbb{R}^{1 \times d}$ denote the length-$\ell$ block map*

$$F_\ell(x, X) := o_\ell(x; X),$$

*where $o_\ell(x; X)$ is computed by (35)–(36) with $R_\ell^{(m)}(x) \in \mathbb{R}^{\ell \times \ell}$ for $m \in \{1, 2\}$ and the activation*

$$\sigma_\ell(z) := \frac{1}{\rho_\ell(z)} \operatorname{ReLU}(z),$$

*as in Assumption B.1.*

*Let $P_{t \to T} \in \mathbb{R}^{T \times t}$ be the canonical injection*

$$P_{t \to T} := \begin{bmatrix} I_t \\ 0 \end{bmatrix}, \qquad P_{t \to T}^\top = \begin{bmatrix} I_t & 0 \end{bmatrix},$$

*and define the reverse-offset truncation*

$$\tau_t(\widetilde{X}) := P_{t \to T}^\top \widetilde{X} = \widetilde{X}[1\!:\!t, :] \in \mathbb{R}^{t \times d}.$$

*Assume the sequence-space mixings are extension-consistent in the sense that for each $m \in \{1, 2\}$, for any chosen length-$t$ operators $R_t^{(m)}(\cdot)$ with our learnable DPLR parameterization, there exists a length-$T$ choice $R_T^{(m)}(\cdot)$ such that*

$$R_T^{(m)}(x) = P_{t \to T} R_t^{(m)}(x) P_{t \to T}^\top, \qquad \forall x, \ m \in \{1, 2\}. \tag{33}$$

*Assume moreover the L2 normalization scalar satisfies a padding invariance property:*

$$\rho_T([z, 0]) = \rho_t(z) \qquad \forall z \in \mathbb{R}^{1 \times t}, \tag{34}$$

*where $[z, 0] \in \mathbb{R}^{1 \times T}$ denotes zero-padding.*

*Then, for every current input $x \in \mathbb{R}^{1 \times d}$ and every longer history $\widetilde{X} \in \mathbb{R}^{T \times d}$, letting $X := \tau_t(\widetilde{X})$, we have*

$$F_T(x, \widetilde{X}) = F_t(x, X).$$

*In particular, any computation realizable at length $t$ is realizable at length $T$ (by an appropriate extension of the sequence-mixing parameters), so the realizable function family is monotone in context length.*

*In the memory-pool formulation, write $(u_i^{(\ell)}, v_i^{(\ell)})$ for the slots instantiated into the Context-instantiated Memory Pool at length $\ell$, and define*

$$\mu_{X,x}^{\mathrm{pool},\ell} := \sum_{i=1}^{\ell} \delta_{(u_i^{(\ell)}, v_i^{(\ell)})}.$$

*Let $\mu_{X,x}^{\mathrm{act},\ell}$ denote the corresponding Current-step Activated Memory measure obtained by hard-gating $\mu_{X,x}^{\mathrm{pool},\ell}$ via $g_x(u, v) = \mathbf{1}\{\alpha(x; u) > 0\}$.*

*Under (33), the length-$T$ slots satisfy*

$$(u_i^{(T)}, v_i^{(T)}) = \begin{cases} (u_i^{(t)}, v_i^{(t)}), & i \in [t], \\ (0, 0), & i \in \{t+1, \dots, T\}, \end{cases}$$

*hence*

$$\mu_{\widetilde{X},x}^{\mathrm{pool},T} = \mu_{X,x}^{\mathrm{pool},t} + \sum_{i=t+1}^{T} \delta_{(0,0)} \qquad \text{and} \qquad \mu_{\widetilde{X},x}^{\mathrm{act},T} = \mu_{X,x}^{\mathrm{act},t}.$$

*Therefore, tokens appearing only in the discarded suffix $\widetilde{X}[t+1\!:\!T, :]$ induce no nontrivial pool atoms (only dummy $(0, 0)$ slots) and contribute exactly zero to the readout.*

*Proof.* Fix $x \in \mathbb{R}^{1 \times d}$ and $\widetilde{X} \in \mathbb{R}^{T \times d}$, and let $X := \tau_t(\widetilde{X}) = P_{t \to T}^\top \widetilde{X}$. Under (33), for each $m \in \{1, 2\}$,

$$R_T^{(m)\top}(x) \widetilde{X} = \left(P_{t \to T} R_t^{(m)}(x) P_{t \to T}^\top\right)^\top \widetilde{X} = P_{t \to T} R_t^{(m)\top}(x) P_{t \to T}^\top \widetilde{X} = P_{t \to T} R_t^{(m)\top}(x) X.$$

Therefore, defining $U_t := R_t^{(1)\top}(x)X$, $V_t := R_t^{(2)\top}(x)X$; $U_T := R_T^{(1)\top}(x)\widetilde{X}$, $V_T := R_T^{(2)\top}(x)\widetilde{X}$, we have

$$U_T = \begin{bmatrix} U_t \\ 0 \end{bmatrix}, \qquad V_T = \begin{bmatrix} V_t \\ 0 \end{bmatrix}.$$

Consequently, the pre-activation at length $T$ is a zero-padded version of the length-$t$ one:

$$h_T(x; \widetilde{X}) = x\, L^{(1)}(x)\, \widetilde{X}^\top R_T^{(1)}(x) = x\, L^{(1)}(x)\, X^\top R_t^{(1)}(x) P_{t \to T}^\top = [h_t(x; X), 0].$$

By Assumption B.1 (applied at the corresponding lengths),

$$\sigma_T(h_T) = \frac{1}{\rho_T([h_t, 0])}\, [\text{ReLU}(h_t), 0], \qquad \text{and by (34),}\ \ \rho_T([h_t, 0]) = \rho_t(h_t).$$

Using $V_T = \begin{bmatrix} V_t \\ 0 \end{bmatrix}$,

$$\sigma_T(h_T)\, V_T = \frac{1}{\rho_t(h_t)}[\text{ReLU}(h_t), 0] \begin{bmatrix} V_t \\ 0 \end{bmatrix} = \frac{1}{\rho_t(h_t)}\, \text{ReLU}(h_t)\, V_t = \sigma_t(h_t)\, V_t.$$

Multiplying by the shared $L^{(2)\top}(x)$ yields $F_T(x, \widetilde{X}) = F_t(x, X)$.

The slot identities follow from $U_T = [U_t; 0]$ and $V_T = [V_t; 0]$. In particular, for $i > t$ we have $u_i^{(T)} = v_i^{(T)} = 0$, hence $\alpha(x; u_i^{(T)}) = 0$ and the dummy atoms contribute zero; moreover they are excluded from the Current-step Activated Memory measure because the hard gate is $\mathbf{1}\{\alpha(x; u) > 0\}$. $\square$

### C.4. Ever-growing hidden representation and an implicit infinite Global (parameter-defined) Memory Space

Finally, the dynamic formulation clarifies why ReLU-based attention behaves fundamentally differently from a fixed-width ReLU MLP.

**Proposition C.8** (Ever-growing hidden width and context-instantiated pool slots)**.** *For each context length $t \geq 1$, the hidden pre-activation satisfies $h_t \in \mathbb{R}^{1 \times t}$ and the Context-instantiated Memory Pool measure is atomic with at most $t$ atoms:*

$$\mu_{X, x_t}^{\text{pool}} = \sum_{i=1}^{t} \delta_{(u_i, v_i)}.$$

*Consequently, the block can instantiate at most $t$ candidate memory slots (and then activates a subset of them) at step $t$.*

*Proof.* Immediate from the definitions: $h_t = x_t L^{(1)}(x_t) X^\top R^{(1)}(x_t)$ has length $t$, and $\mu_{X, x_t}^{\text{pool}}$ places one atom per row pair $(u_i, v_i)$ of $U = R^{(1)\top}(x_t)X$ and $V = R^{(2)\top}(x_t)X$. $\square$

**Proposition C.9** (Implicit infinite reachable atom set across contexts)**.** *Assume the token/state space contains infinitely many distinct vectors (e.g., an infinite subset of $\mathbb{R}^d$). Assume moreover that the parameterization (pure ReLU attention) of the sequence-mixing operators contains the identity mapping, i.e., for each $t$ there exists a choice of parameters such that $R_\theta^{(1)}(x) \equiv I_t$, $R_\theta^{(2)}(x) \equiv I_t$, for all $x$. Define the reachable atom set (across all contexts and lengths)*

$$\mathcal{A}_\theta := \bigcup_{t \geq 1}\ \bigcup_{X \in \mathbb{R}^{t \times d}}\ \bigcup_{x \in \mathbb{R}^{1 \times d}} \text{supp}(\mu_{X, x}^{\text{pool}}) \subseteq \Omega := \mathbb{R}^d \times \mathbb{R}^d.$$

*Then $\mathcal{A}_\theta$ is infinite (indeed, uncountable whenever the token/state space contains a continuum).*

*Proof.* It suffices to consider $t = 1$. Under $R_\theta^{(1)} \equiv R_\theta^{(2)} \equiv I_1$ and any context matrix $X = [x] \in \mathbb{R}^{1 \times d}$, we have $u_1 = v_1 = x$, hence $\mu_{X, x}^{\text{pool}} = \delta_{(x, x)}$ and $\text{supp}(\mu_{X, x}^{\text{pool}}) = \{(x, x)\} \subset \Omega$. As $x$ ranges over an infinite set of distinct vectors, we obtain infinitely many distinct atoms $(x, x)$, so $\mathcal{A}_\theta$ is infinite. If the token/hidden state space contains a continuum, the set $\{(x, x)\}$ is uncountable. $\square$

**Discussion.** Above formalizes the key methodological shift: instead of treating attention weights as probabilities that must be normalized, we treat the block as a dynamic ReLU MLP that (i) specifies a Global (parameter-defined) Memory Space (an ambient atom space together with a parameter-defined address/read dictionary), (ii) instantiates a finite Context-instantiated Memory Pool from the current history, and (iii) performs input-conditioned sub-network selection via Current-step Activated Memory over that pool. Assumption B.1 makes explicit why replacing probability normalization with L2-based normalization can preserve the selection mechanism (the activated set) while only rescaling magnitudes, aligning the block with standard MLP practice.

## C.5. Partition Geometry: From Polyhedral Splines to Warped Splines

We summarize the geometric view of our dynamic two-layer block. For one head, given the current token $x \in \mathbb{R}^{1 \times d}$ and the (reverse-offset) context matrix $X := X_{t:1} \in \mathbb{R}^{t \times d}$, define

$$h(x; X) := x \, L^{(1)}(x) \, X^\top R^{(1)}(x) \in \mathbb{R}^{1 \times t}, \tag{35}$$

$$o(x; X) := \sigma\big(h(x; X)\big) \, R^{(2)\top}(x) \, X \, L^{(2)\top}(x) \in \mathbb{R}^{1 \times d}. \tag{36}$$

Throughout this section we focus on the sign pattern of $h(x; X)$, which fully determines the ReLU mask. If $\mathrm{L2Norm}_t$ is a positive scalar rescaling across the length-$t$ vector, then it preserves sign patterns and does not affect the partition geometry (Remark C.16); however, it can change whether the map is piecewise linear by introducing a nonconstant positive scalar factor (Remark C.11).

**Notation.** For fixed $X$, let $h_j(x; X)$ denote the $j$-th coordinate of $h(x; X)$, and define the boundary set

$$\mathcal{B}_X := \bigcup_{j=1}^t \{x : \ h_j(x; X) = 0\}. \tag{37}$$

On $\mathbb{R}^d \setminus \mathcal{B}_X$, the sign pattern $s_X(x) := \mathbf{1}\{h(x; X) > 0\} \in \{0, 1\}^t$ is locally constant.

### C.5.1. STATIC MIXING: POLYHEDRAL (PIECEWISE-LINEAR) GEOMETRY

We first consider the "static" case, which recovers the classical polyhedral geometry of two-layer ReLU networks.

**Theorem C.10** (Polyhedral spline under static mixing). *Assume $L^{(1)}(x) \equiv L^{(1)}$, $L^{(2)}(x) \equiv L^{(2)}$ and $R^{(1)}(x) \equiv R^{(1)}$, $R^{(2)}(x) \equiv R^{(2)}$ are constant w.r.t. $x$. Define the effective context matrices*

$$U := R^{(1)\top} X, \qquad V := R^{(2)\top} X, \tag{38}$$

*and let*

$$B(X) := L^{(1)} U^\top = L^{(1)} X^\top R^{(1)} \in \mathbb{R}^{d \times t}. \tag{39}$$

*Let $b_j \in \mathbb{R}^d$ be the $j$-th column of $B(X)$ and define hyperplanes $H_j := \{x : \ x \, b_j = 0\}$. Then, for fixed $X$, the map $x \mapsto o(x; X)$ is continuous piecewise linear (CPWL). Moreover, on each open cell (possibly empty)*

$$\mathcal{C}_s(X) := \Big( \bigcap_{j:s_j=1} \{x : \ x \, b_j > 0\} \Big) \cap \Big( \bigcap_{j:s_j=0} \{x : \ x \, b_j < 0\} \Big), \qquad s \in \{0, 1\}^t, \tag{40}$$

*the output is linear:*

$$\forall x \in \mathcal{C}_s(X): \qquad o(x; X) = x \, A_s(X), \quad \text{where} \quad A_s(X) = B(X) \, D_s \, V \, L^{(2)\top}, \quad D_s := \mathrm{diag}(s). \tag{41}$$

*Proof.* Under the stated assumptions, $h(x; X) = x \, B(X)$ and $o(x; X) = \mathrm{ReLU}(x B(X)) \, V \, L^{(2)\top}$. On $\mathcal{C}_s(X)$, the sign pattern of $x B(X)$ is fixed to $s$, hence $\mathrm{ReLU}(x B(X)) = (x B(X)) D_s$. Substituting yields (41). Continuity follows from continuity of ReLU and matrix multiplication. $\square$

*Remark* C.11 (Scalar L2 normalization preserves boundaries but generally breaks CPWL). In our implementation we often use the normalized activation

$$\sigma_t(z) := \mathrm{ReLU}(\mathrm{L2Norm}_t(z)) = \frac{1}{\rho_t(z)} \mathrm{ReLU}(z), \qquad \rho_t(z) > 0 \text{ a scalar,}$$

which preserves sign patterns (hence leaves $\mathcal{B}_X$ unchanged) but introduces a nonconstant positive rescaling. In the static setting of Theorem C.10, replacing ReLU by $\sigma_t$ yields

$$o_{\text{norm}}(x; X) = \sigma_t(xB(X)) \, V \, L^{(2)\top}.$$

On any cell $\mathcal{C}_s(X)$, since $\text{ReLU}(xB(X)) = (xB(X))D_s$, we obtain

$$\forall x \in \mathcal{C}_s(X): \qquad o_{\text{norm}}(x; X) = \frac{1}{\rho_t(xB(X))} \, x \, A_s(X).$$

Thus the partition geometry (the hyperplanes $H_j$) is unchanged, but the map is generally not linear on $\mathcal{C}_s(X)$ (hence not CPWL) unless $\rho_t$ is constant on that cell. When $\rho_t$ is smooth and bounded away from 0 (e.g., $\rho_t(z) = \sqrt{\|z\|_2^2 + \varepsilon}$ with $\varepsilon > 0$), $o_{\text{norm}}(\cdot; X)$ is smooth on each cell.

**What $R^{(1)}$ changes in the static regime.** Theorem C.10 shows that the partition boundaries are exactly the hyperplanes $H_j$, whose normals are the columns of $B(X) = L^{(1)}X^\top R^{(1)}$. Thus $R^{(1)}$ acts in the hidden/sequence dimension by mixing these normals.

**Corollary C.12** (Diagonal vs. mixing effects of $R^{(1)}$ (static)). *In Theorem C.10, write $B(X) = B_0(X)R^{(1)}$ with $B_0(X) := L^{(1)}X^\top$. If $R^{(1)}$ is diagonal with all diagonal entries nonzero, then the set of partition hyperplanes $\{H_j\}_{j=1}^t$ is unchanged (each normal is only rescaled), i.e., the geometry of the knots is preserved. In contrast, any off-diagonal mixing in $R^{(1)}$ replaces each normal by a linear combination of the base normals:*

$$b_j = B_0(X) \, r_j = \sum_{i=1}^t (r_j)_i \, b_i^{(0)}, \qquad r_j := R^{(1)}[:, j], \;\; b_i^{(0)} := B_0(X)[:, i], \tag{42}$$

*thereby changing the hyperplane arrangement and hence the partition geometry.*

*Proof.* If $R^{(1)} = \text{diag}(d)$ with $d_j \neq 0$, then $b_j = d_j b_j^{(0)}$ and $\{x : x b_j = 0\} = \{x : x b_j^{(0)} = 0\}$, so the hyperplanes are identical. For general $R^{(1)}$, the column identity follows from $B = B_0 R^{(1)}$. $\qquad\square$

### C.5.2. FULLY DYNAMIC MIXING: WARPED (PIECEWISE-SMOOTH) GEOMETRY

We now turn to the regime relevant to our Global (parameter-defined) Memory Space $\to$ Context-instantiated Memory Pool $\to$ Current-step Activated Memory view: both feature mixing ($L$) and sequence mixing ($R$) depend on the current input $x$. In this case the partition is no longer polyhedral; it becomes the pullback of the orthant partition under a nonlinear map $x \mapsto h(x; X)$.

**Theorem C.13** (Pullback orthant partition and piecewise-smoothness). *Fix $X$. Assume that each coordinate $h_j(\cdot; X)$ in (35) is continuous. Then $\mathbb{R}^d \setminus \mathcal{B}_X$ (with $\mathcal{B}_X$ from (37)) decomposes into open connected components on each of which the sign pattern $s_X(x) = \mathbf{1}\{h(x; X) > 0\}$ is constant.*

*If moreover $L^{(1)}(\cdot), R^{(1)}(\cdot), R^{(2)}(\cdot), L^{(2)}(\cdot)$ are $C^k$ ($k \geq 1$), then $o(\cdot; X)$ in (36) is $C^k$ on each connected component of $\mathbb{R}^d \setminus \mathcal{B}_X$. Furthermore, at any point $x_0 \in \mathcal{B}_X$ where $h_j(x_0; X) = 0$ and $\nabla_x h_j(x_0; X) \neq 0$, the set $\{x : h_j(x; X) = 0\}$ is a $C^k$ hypersurface in a neighborhood of $x_0$.*

*Proof.* Let $x_0 \notin \mathcal{B}_X$. Then $h_j(x_0; X) \neq 0$ for all $j$. By continuity of each $h_j(\cdot; X)$, there exists a neighborhood of $x_0$ on which every $h_j$ preserves its sign, hence the sign pattern is constant locally. Therefore the sign pattern is constant on each connected component of $\mathbb{R}^d \setminus \mathcal{B}_X$.

On such a component, the ReLU mask is constant: there exists $s \in \{0, 1\}^t$ such that $\text{ReLU}(h(x; X)) = h(x; X)D_s$ holds throughout the component. Substituting into (36) shows $o(\cdot; X)$ is a product/compose of $C^k$ maps, hence $C^k$ on the component.

Finally, if $h_j(\cdot; X)$ is $C^k$ and $\nabla_x h_j(x_0; X) \neq 0$ at a zero $h_j(x_0; X) = 0$, the implicit function theorem implies $\{x : h_j(x; X) = 0\}$ is a $C^k$ hypersurface near $x_0$. $\qquad\square$

**Local geometry: "moving" hyperplanes and curvature sources.** Theorem C.13 implies that, in the fully dynamic regime, partition boundaries are generally curved (level sets), not fixed hyperplanes. Locally, however, every smooth boundary admits a tangent hyperplane.

**Proposition C.14** (Local linearization and deformation terms). *Fix $X$ and define*

$$z(x; X) := x\, L^{(1)}(x)\, X^\top \in \mathbb{R}^{1 \times t}, \qquad \text{so that} \qquad h(x; X) = z(x; X)\, R^{(1)}(x).$$

*Then the first-order differential (with $X$ fixed) is*

$$dh = (dx)\, L^{(1)}(x)\, X^\top R^{(1)}(x) +\ x\, (dL^{(1)}(x))\, X^\top R^{(1)}(x) +\ x\, L^{(1)}(x)\, X^\top (dR^{(1)}(x)). \tag{43}$$

*Consequently, even when the "base" term $(dx) L^{(1)} X^\top R^{(1)}$ would induce polyhedral boundaries, input-dependent feature mixing $(dL^{(1)})$ and, critically, input-dependent sequence mixing $(dR^{(1)})$ introduce additional deformation terms that bend and move the partition surfaces.*

*Proof.* Apply the product rule to $h = z\, R^{(1)}$ to get $dh = (dz)R^{(1)} + z(dR^{(1)})$. Then apply the product rule to $z = x\, L^{(1)}\, X^\top$ to get $dz = (dx)L^{(1)}X^\top + x(dL^{(1)})X^\top$, and substitute. $\qquad\square$

*Remark* C.15 (Low-rank dynamic $R$ yields controlled smooth deformations). In HyperMLP, $R^{(1)}(x) = D + A\, S(x)\, B^\top$ with diagonal $S(x)$ (e.g., $S(x) = \text{Diag}(\phi(xW_S))$). Then $dR^{(1)}(x) = A\, (dS(x))\, B^\top$ has rank at most $r_s$. Combined with (43), this shows the geometry is deformed through a low-dimensional, smooth mechanism, suggesting a favorable expressivity–stability tradeoff: we can warp the gating surfaces without introducing arbitrary high-curvature boundaries.

*Remark* C.16 (L2 normalization and sign patterns). If $\text{L2Norm}_t(h) = h/\rho(h)$ with $\rho(h) > 0$ a scalar, then $\mathbf{1}\{\text{L2Norm}_t(h) > 0\} = \mathbf{1}\{h > 0\}$. Hence inserting $\text{L2Norm}_t$ before ReLU does not change the partition geometry; it only rescales magnitudes.

**Takeaway.** Static attention-like blocks induce polyhedral (piecewise-linear) partitions tied to a fixed hyperplane arrangement. Our fully dynamic design replaces this with a warped orthant partition: the context instantiates a finite Context-instantiated Memory Pool of slots, while input-dependent feature and sequence mixing (especially $R^{(1)}(x)$) smoothly deforms the activation boundaries that determine the Current-step Activated Memory, yielding a piecewise-smooth, input-conditioned "spline" geometry.

# D. Functional-Analytic and NTK Connections

This section summarizes how our memory-measure formulation of the Global (parameter-defined) Memory Space $\to$ Context-instantiated Memory Pool $\to$ Current-step Activated Memory decomposition relates to (i) Banach-space function classes (variation-norm/Barron-type views of two-layer models) and (ii) the Neural Tangent Kernel (NTK) / lazy-training regime.

## D.1. A Banach-space view on reachable Context-instantiated Memory Pool measures

**Notation.** We use the common setup in Appendix B. In particular, the block is defined by (22)–(23), and the pool/activated measures and dictionary maps are (B)–(30) and (B)–(28).

**Restricting to a reachable compact support.** The key point is that the instantiated pool atoms $(u_j, v_j)$ do not range over all of the Global (parameter-defined) Memory Space $\Omega$; they are restricted by the norms of $(x_t, X)$ and the operator norms of the mixing maps.

**Assumption D.1** (Bounded contexts and (measurable) uniformly bounded mixings). Fix a bounded input domain $\mathcal{X} \subset \mathbb{R}^{1 \times d}$ and a bounded set of context matrices $\mathcal{C}_t \subset \mathbb{R}^{t \times d}$ with

$$B_X := \sup_{X \in \mathcal{C}_t} \|X\|_F\ < \infty, \qquad B_x := \sup_{x \in \mathcal{X}} \|x\|_2\ < \infty.$$

Assume the mixing operators are Borel measurable and uniformly bounded on $\mathcal{X}$:

$$\sup_{x \in \mathcal{X}} \|R_\theta^{(j)}(x)\|_{\text{op}}\ \le\ C_{Rj}, \qquad \sup_{x \in \mathcal{X}} \|L_\theta^{(j)}(x)\|_{\text{op}}\ \le\ C_{Lj}, \qquad j \in \{1, 2\}.$$

**Lemma D.2** (Reachable atoms lie in a compact set). *Under Assumption D.1, for all $x \in \mathcal{X}$ and $X \in \mathcal{C}_t$, the instantiated atoms satisfy*

$$\|u_j\|_2 \le C_{R1} B_X, \qquad \|v_j\|_2 \le C_{R2} B_X, \qquad j \in [t].$$

*Hence $\mu_{X,x}^{\text{pool}}$ is supported on the compact set*

$$\Omega_t := \left\{ (u,v) \in \mathbb{R}^d \times \mathbb{R}^d : \|u\|_2 \le C_{R1} B_X,\ \|v\|_2 \le C_{R2} B_X \right\} \subset \Omega.$$

*Proof.* Since $U = R_\theta^{(1)\top}(x) X$, we have $\|U\|_F \le \|R_\theta^{(1)}(x)\|_{\text{op}} \|X\|_F \le C_{R1} B_X$. For each row, $\|u_j\|_2 \le \|U\|_F \le C_{R1} B_X$. The argument for $V$ is identical. □

**Lemma D.3** (Measurability and boundedness of the feature map on $\mathcal{X} \times \Omega_t$). *Under Assumption D.1, the feature map $\Psi_\theta(x; (u,v)) := \alpha_\theta(x; u)_+ \beta_\theta(x; v)$ is Borel measurable and bounded on $\mathcal{X} \times \Omega_t$. In particular,*

$$\|\Psi_\theta\|_{\infty,2} := \sup_{x \in \mathcal{X}, (u,v) \in \Omega_t} \|\Psi_\theta(x; (u,v))\|_2 \le B_x\, C_{L1}\, C_{L2}\, (C_{R1} B_X)\, (C_{R2} B_X) < \infty.$$

*Proof.* Measurability follows from measurability of $x \mapsto L_\theta^{(j)}(x)$ and matrix products. For the bound, $\alpha_+(x; u) \le |\alpha_\theta(x; u)| \le \|x\|_2 \|L_\theta^{(1)}(x)\|_{\text{op}} \|u\|_2 \le B_x C_{L1} \|u\|_2$, and $\|\beta_\theta(x; v)\|_2 \le \|v\|_2 \|L_\theta^{(2)}(x)\|_{\text{op}} \le C_{L2} \|v\|_2$. Use Lemma D.2 to bound $\|u\|_2, \|v\|_2$ on $\Omega_t$ and multiply the inequalities. □

**Feature map and bounded linear readout on $\mathcal{M}(\Omega_t)$.** Let $\mathcal{M}(\Omega_t)$ be the Banach space of finite signed (Radon) measures on $\Omega_t$ equipped with the total variation norm $\|\cdot\|_{\text{TV}}$. Define the feature map

$$\Psi_\theta(x; (u,v)) := \alpha_\theta(x; u)_+ \beta_\theta(x; v) \in \mathbb{R}^{1 \times d}, \qquad \alpha_+(x; u) := \max\{0, \alpha_\theta(x; u)\}.$$

Define the (vector-valued) linear operator $T_{\theta,t} : \mathcal{M}(\Omega_t) \to L_\infty(\mathcal{X}; \mathbb{R}^d)$ by

$$(T_{\theta,t}\mu)(x) := \int_{\Omega_t} \Psi_\theta(x; (u,v))\, d\mu(u,v), \qquad x \in \mathcal{X}, \tag{44}$$

where the integral is understood componentwise (equivalently as a Bochner integral).

**Theorem D.4** (Bounded linear readout over reachable measures). *Under Assumption D.1, the operator $T_{\theta,t}$ is well-defined, linear, and bounded:*

$$\|T_{\theta,t}\mu\|_{\infty,2} \le \|\Psi_\theta\|_{\infty,2} \|\mu\|_{\text{TV}}, \qquad \|\Psi_\theta\|_{\infty,2} := \sup_{x \in \mathcal{X}, (u,v) \in \Omega_t} \|\Psi_\theta(x; (u,v))\|_2,$$

*where $\|\Psi_\theta\|_{\infty,2} < \infty$ by Lemma D.3. Moreover, under Assumption B, for any $x_t \in \mathcal{X}$ and $X \in \mathcal{C}_t$, the block output satisfies*

$$o_t = \frac{1}{\rho(h_t)} (T_{\theta,t}\mu_{X,x_t}^{\text{pool}})(x_t), \qquad h_t \text{ as in (B)}. \tag{45}$$

*Proof.* By Lemma D.3, $\Psi_\theta$ is measurable and bounded on $\mathcal{X} \times \Omega_t$, hence the (Bochner/componentwise) integral in (44) is well-defined for any $\mu \in \mathcal{M}(\Omega_t)$ and is absolutely convergent. The stated bound follows from the standard estimate

$$\|(T_{\theta,t}\mu)(x)\|_2 \le \int_{\Omega_t} \|\Psi_\theta(x; (u,v))\|_2\, d|\mu|(u,v) \le \|\Psi_\theta\|_{\infty,2} \|\mu\|_{\text{TV}}.$$

For (45), expand the forward pass: $o_t = \sum_{j=1}^t \text{ReLU}(\text{L2Norm}_t(h_t))_j\, \beta_\theta(x_t; v_j)$. By Lemma B.2, $\text{ReLU}(\text{L2Norm}_t(h_t)) = \rho(h_t)^{-1}\text{ReLU}(h_t)$. Since $(h_t)_j = \alpha_\theta(x_t; u_j)$, we get

$$o_t = \frac{1}{\rho(h_t)} \sum_{j=1}^t \alpha_\theta(x_t; u_j)_+ \beta_\theta(x_t; v_j) = \frac{1}{\rho(h_t)} \int_{\Omega_t} \Psi_\theta(x_t; (u,v))\, d\mu_{X,x_t}^{\text{pool}}(u,v),$$

which is exactly (45). □

*Remark* D.5 (Banach/variation view for the pre-normalized readout). Our linear operator $T_{\theta,t}$ captures the *pre-normalized* measure readout. Define the pre-normalized output

$$\widetilde{o}_t \ := \ \rho_t(h_t)\, o_t, \qquad \text{where} \qquad o_t = \frac{1}{\rho_t(h_t)}\, (T_{\theta,t}\mu^{\mathrm{pool}}_{X,x_t})(x_t)$$

and $h_t$ is the length-$t$ pre-activation vector. Then

$$\widetilde{o}_t \ = \ (T_{\theta,t}\mu^{\mathrm{pool}}_{X,x_t})(x_t),$$

so $\widetilde{o}_t$ is a bounded linear functional of the instantiated pool measure (and thus lies in $\mathrm{Ran}(T_{\theta,t})$ for fixed $t$).

The actual head output rescales this readout by the positive scalar $1/\rho_t(h_t)$. For our choice $\rho_t(z) = \sqrt{\|z\|_2^2 + \varepsilon}$ with $\varepsilon > 0$, we have $\rho_t(z) \geq \sqrt{\varepsilon}$ for all $z$, hence

$$\|o_t\|_2 \ \leq \ \frac{1}{\sqrt{\varepsilon}}\|\widetilde{o}_t\|_2 \ \leq \ \frac{1}{\sqrt{\varepsilon}}\|\Psi_\theta\|_{\infty,2}\,\|\mu^{\mathrm{pool}}_{X,x_t}\|_{\mathrm{TV}}.$$

Therefore, all TV/variation-norm bounds proved for the linear readout extend to the normalized head up to a fixed multiplicative constant (and the scalar normalization does not affect gating/sign patterns; cf. Lemma B.2).

**Theorem D.6** (Restriction/activation is a TV contraction). *Let $\mu \in \mathcal{M}(\Omega_t)$ and let $g : \Omega_t \to \mathbb{R}$ be measurable with $|g| \leq 1$. Define $(g\mu)(A) := \int_A g\, d\mu$. Then*

$$\|g\mu\|_{\mathrm{TV}} \ \leq \ \|\mu\|_{\mathrm{TV}}.$$

*In particular, for the hard gate $g_x(u,v) := \mathbf{1}\{\alpha_\theta(x;u) > 0\}$, the Current-step Activated Memory measure $\mu^{\mathrm{act}} := g_x\,\mu^{\mathrm{pool}}$ satisfies $\|\mu^{\mathrm{act}}\|_{\mathrm{TV}} \leq \|\mu^{\mathrm{pool}}\|_{\mathrm{TV}}$.*

**Induced variation norm on the reachable function class.** Define the reachable function class $\mathcal{F}_{\theta,t} := \mathrm{Ran}(T_{\theta,t}) \subset L_\infty(\mathcal{X};\mathbb{R}^d)$ and equip it with the quotient norm

$$\|f\|_{\mathrm{var},\theta,t} \ := \ \inf\{\|\mu\|_{\mathrm{TV}} : \ T_{\theta,t}\mu = f\}.$$

**Theorem D.7** (Variation norm yields a Banach function space). *The space $(\mathcal{F}_{\theta,t}, \|\cdot\|_{\mathrm{var},\theta,t})$ is a Banach space and satisfies*

$$\|f\|_{\infty,2} \ \leq \ \|\Psi_\theta\|_{\infty,2}\,\|f\|_{\mathrm{var},\theta,t}, \qquad \forall f \in \mathcal{F}_{\theta,t}.$$

*Proof.* The inequality follows directly from Theorem D.4 by taking the infimum over representing measures. Banach-ness follows from the fact that $\mathcal{F}_{\theta,t}$ endowed with $\|\cdot\|_{\mathrm{var},\theta,t}$ is isometrically isomorphic to the quotient Banach space $\mathcal{M}(\Omega_t)/\ker(T_{\theta,t})$, and quotients of Banach spaces are Banach. $\square$

### D.2. NTK and the lazy-training regime

We next summarize the NTK connection for our dynamic block. Fix $t$ and let $z$ denote the (vectorized) input comprising $(x_t, X) \in \mathbb{R}^{(t+1)\times d}$, so $z \in \mathbb{R}^{(t+1)d}$. Here we treat the block as a function of two independent arguments $(x, X)$ (query and memory). Let $f_\theta(z) := o_t$ denote the block output (possibly after head aggregation). Assume $f_\theta$ is differentiable in $\theta$ at the inputs of interest (this holds almost surely under continuous initialization, since nondifferentiability occurs only on measure-zero boundaries of ReLU gates).

**Definition D.8** (Neural Tangent Kernel (matrix-valued)). Let $J_\theta(z) := \nabla_\theta f_\theta(z) \in \mathbb{R}^{d\times|\theta|}$ be the Jacobian. Define the (matrix-valued) NTK

$$\Theta_\theta(z,z') \ := \ J_\theta(z)\, J_\theta(z')^\top \in \mathbb{R}^{d\times d}.$$

**Theorem D.9** (Exact prediction dynamics under gradient flow). *Consider squared loss on a dataset $\{(z_i, y_i)\}_{i=1}^n$:*

$$\mathcal{L}(\theta) = \frac{1}{2}\sum_{i=1}^n \|f_\theta(z_i) - y_i\|_2^2, \qquad \dot{\theta}(t) = -\nabla_\theta\mathcal{L}(\theta(t)).$$

*Then for any input $z$,*

$$\frac{d}{dt}f_{\theta(t)}(z) = -\sum_{i=1}^n \Theta_{\theta(t)}(z,z_i)\,\big(f_{\theta(t)}(z_i) - y_i\big).$$

*Proof.* By the chain rule, $\frac{d}{dt} f_{\theta(t)}(z) = J_{\theta(t)}(z)\,\dot{\theta}(t)$. Also, $\nabla_\theta \mathcal{L}(\theta) = \sum_{i=1}^n J_\theta(z_i)^\top (f_\theta(z_i) - y_i)$. Substitute $\dot{\theta}(t) = -\nabla_\theta \mathcal{L}(\theta(t))$ and regroup:

$$\frac{d}{dt} f_{\theta(t)}(z) = -\sum_{i=1}^n J_{\theta(t)}(z)\,J_{\theta(t)}(z_i)^\top (f_{\theta(t)}(z_i) - y_i) = -\sum_{i=1}^n \Theta_{\theta(t)}(z, z_i)\,(f_{\theta(t)}(z_i) - y_i).$$

$\square$

**Corollary D.10** (Kernel (lazy) regime). *If along training $\Theta_{\theta(t)}(z, z') \approx \Theta_{\theta(0)}(z, z')$ for all relevant $(z, z')$, then the prediction dynamics are approximately those of kernel gradient flow driven by the fixed kernel $\Theta_{\theta(0)}$. In particular, when $\Theta_{\theta(t)} \equiv \Theta_{\theta(0)}$ is exactly constant, the dynamics are linear in function space and coincide with kernel regression/gradient flow in the (matrix-valued) RKHS induced by $\Theta_{\theta(0)}$.*

**A deterministic gate-stability lemma (linking to Current-step Activated Memory selection).** The NTK regime is often associated with small parameter movement. A direct consequence in our formulation is stability of the Current-step Activated Memory selection whenever pre-activations stay away from $0$.

**Lemma D.11** (Current-step Activated Memory stability under a margin). *Fix an input $(x_t, X)$ and define $h_t(\theta)$ by (B). Suppose that at some reference parameters $\theta_0$ there exists $\delta > 0$ such that*

$$\min_{j \in [t]} |(h_t(\theta_0))_j| \;\geq\; \delta.$$

*If $\theta$ satisfies $\|h_t(\theta) - h_t(\theta_0)\|_\infty < \delta$, then the gate pattern is unchanged:*

$$\mathbf{1}\{h_t(\theta) > 0\} \;=\; \mathbf{1}\{h_t(\theta_0) > 0\},$$

*and hence the Current-step Activated Memory restriction (as defined in Theorem D.6) selects the same subset of atoms.*

*Proof.* For each coordinate $j$, $|(h_t(\theta))_j - (h_t(\theta_0))_j| < \delta \leq |(h_t(\theta_0))_j|$ implies $(h_t(\theta))_j$ cannot cross $0$, so its sign is preserved. Therefore the indicator mask (and the induced restriction) is identical. $\square$

**Summary.** Theorems D.4–D.9 formalize two complementary perspectives: (i) our forward pass is a bounded linear readout over a Banach space of measures (with restriction/activation as a TV contraction), aligning with variation-norm tools for two-layer models; and (ii) in regimes where training linearizes the model (NTK/lazy training), learning is governed by a fixed kernel in an RKHS, and the Current-step Activated Memory selection can become stable under a margin condition (Lemma D.11), thereby limiting changes in dynamic routing on the training set.

# E. Mechanistic Implications: Context-Instantiated Dictionaries and Multi-Stage Selection

This section distills the measure and operator views developed in the previous section into a mechanistic picture that is both (i) mathematically checkable and (ii) directly interpretable. The key message is: parameters define a Global (parameter-defined) Memory Space (how to use memory), the context instantiates a finite Context-instantiated Memory Pool (what memory is), and ReLU induces Current-step Activated Memory by selecting over that pool.

**Standing assumption (L2 normalization sign invariance).** Throughout this section we adopt the common regime where $\text{L2Norm}_t$ is a positive scalar rescaling across the length-$t$ vector: there exists a map $\rho : \mathbb{R}^{1 \times t} \to (0, \infty)$ such that

$$\text{L2Norm}_t(z) = \frac{z}{\rho(z)}.$$

Consequently, $\text{ReLU}(\text{L2Norm}_t(z)) = \rho(z)^{-1}\text{ReLU}(z)$ and the ReLU gate pattern is unchanged.

## E.1. Dictionary–measure decomposition of the block

**Notation.** We use the common setup in Appendix B. In particular, the block is (22)–(23), the dictionary element is (28), and the pool/activated measures are (B)–(30).

**Context-instantiation operators and pool slots.** Define the context-instantiation operators

$$\mathcal{E}_{\theta,t}^{(j)}(x)[X] := R_\theta^{(j)\top}(x)\, X \in \mathbb{R}^{t\times d}, \qquad j \in \{1, 2\},$$

and set

$$U_\theta(x; X) := \mathcal{E}_{\theta,t}^{(1)}(x)[X], \qquad V_\theta(x; X) := \mathcal{E}_{\theta,t}^{(2)}(x)[X].$$

Write their rows as $u_j := U_\theta(x; X)[j, :] \in \mathbb{R}^{1\times d}$ and $v_j := V_\theta(x; X)[j, :] \in \mathbb{R}^{1\times d}$.

**Global (parameter-defined) Memory Space and parameter-defined dictionary elements.** Let

$$\Omega := \mathbb{R}^d \times \mathbb{R}^d,$$

whose elements $\omega = (u, v)$ represent an address-side vector $u$ and a content-side vector $v$. Define the (parameter-induced) addressing and readout maps

$$\alpha_\theta(x; u) := x\, L_\theta^{(1)}(x)\, u^\top \in \mathbb{R}, \qquad \beta_\theta(x; v) := v\, L_\theta^{(2)\top}(x) \in \mathbb{R}^{1\times d}, \tag{46}$$

and the corresponding (vector-valued) dictionary element as in (28). The family $\{\Psi_\theta(\cdot; \omega) : \omega \in \Omega\}$ is a parameter-defined global dictionary over the Global (parameter-defined) Memory Space $\Omega$.

**Context-instantiated Memory Pool as an atomic measure.** Define the atomic (finite) measure as in (29), which records the Context-instantiated Memory Pool of $(u_j, v_j)$ slots instantiated from $(X, x)$ through sequence-space mixing. Note that in the fully dynamic regime, $R_\theta^{(j)}(x)$ depends on $x$, hence $\mu_{X,x}^{\mathrm{pool}}$ is input-dependent; the statements below are therefore understood pointwise in $(X, x)$.

**Proposition E.1** (Pointwise measure readout and current-step activation). *Let $h(x; X)$ and $o(x; X)$ be defined by* (B). *Under the standing L2-normalization sign-invariance assumption,*

$$o(x; X) = \frac{1}{\rho(h(x; X))} \int_\Omega \Psi_\theta(x; (u, v))\, d\mu_{X,x}^{\mathrm{pool}}(u, v). \tag{47}$$

*Equivalently,*

$$o(x; X) = \frac{1}{\rho(h(x; X))} \sum_{j=1}^t \alpha_\theta(x; u_j)_+\, \beta_\theta(x; v_j).$$

*Proof.* Since $X^\top R_\theta^{(1)}(x) = (R_\theta^{(1)\top}(x)X)^\top = U_\theta(x; X)^\top$, we have $(h(x; X))_j = \alpha_\theta(x; u_j)$ for all $j$. Moreover, $R_\theta^{(2)\top}(x)X = V_\theta(x; X)$ gives

$$o(x; X) = \sum_{j=1}^t \sigma(h(x; X))_j\, \beta_\theta(x; v_j).$$

By the standing assumption, $\sigma(h) = \rho(h)^{-1}\mathrm{ReLU}(h) = \rho(h)^{-1}(h)_+$ elementwise, hence

$$o(x; X) = \frac{1}{\rho(h(x; X))} \sum_{j=1}^t \alpha_\theta(x; u_j)_+\, \beta_\theta(x; v_j).$$

Writing the finite sum as an integral against the atomic measure (29) yields (47). $\qquad\square$

**Lemma E.2** (Pushforward form of pool instantiation). *Let $\nu_t := \sum_{j=1}^t \delta_j$ be the counting measure on $[t]$. Define the (instantiation) map*

$$\Gamma_\theta(x; X) : [t] \to \Omega, \qquad \Gamma_\theta(x; X)(j) := (u_j, v_j),$$

*where $(u_j, v_j)$ are induced by $U_\theta(x; X)$ and $V_\theta(x; X)$. Then*

$$\mu_{X,x}^{\mathrm{pool}} = (\Gamma_\theta(x; X))_\#\nu_t.$$

*Proof.* For any measurable $A \subseteq \Omega$, $(\Gamma_\#\nu_t)(A) = \nu_t(\Gamma^{-1}(A)) = \sum_{j=1}^t \mathbf{1}\{\Gamma(j) \in A\} = \sum_{j=1}^t \delta_{(u_j,v_j)}(A) = \mu_{X,x}^{\mathrm{pool}}(A)$. $\qquad\square$

### E.2. Global (parameter-defined) Memory Space vs. instantiated pools

The decomposition above suggests three distinct objects:

- **Global (parameter-defined) Memory Space:** $\Omega = \mathbb{R}^d \times \mathbb{R}^d$ (potential atoms), together with parameter-induced dictionary elements $\Psi_\theta(\cdot;\omega)$.

- **Context-instantiated Memory Pool:** the (multi)set of slots $\{(u_j, v_j)\}_{j=1}^t$, equivalently $\mathrm{supp}(\mu_{X,x}^{\mathrm{pool}})$ (with multiplicities).

- **Current-step Activated Memory:** the subpool selected by ReLU gating (equivalently, $\mathrm{supp}(\mu_{X,x}^{\mathrm{act}})$).

**Definition E.3** (Current-step activation restriction and activated pool). Define the hard gate

$$g_x(u, v) := \mathbf{1}\{\alpha_\theta(x; u) > 0\} \in \{0, 1\},$$

and the Current-step Activated Memory measure

$$\mu_{X,x}^{\mathrm{act}} := g_x \, \mu_{X,x}^{\mathrm{pool}}.$$

The activated pool is $\mathrm{supp}(\mu_{X,x}^{\mathrm{act}}) \subseteq \mathrm{supp}(\mu_{X,x}^{\mathrm{pool}})$.

*Remark* E.4 (Multi-stage selection as support pruning). By construction,

$$\Omega \supset \mathrm{supp}(\mu_{X,x}^{\mathrm{pool}}) \supset \mathrm{supp}(\mu_{X,x}^{\mathrm{act}}),$$

capturing the chain Global (parameter-defined) Memory Space $\to$ Context-instantiated Memory Pool $\to$ Current-step Activated Memory.

*Remark* E.5 (Finite pool per step, potentially infinite across contexts). For any fixed $t$ and any $(X, x)$, the Context-instantiated Memory Pool contains at most $t$ atoms (counting multiplicities), hence $|\mathrm{supp}(\mu_{X,x}^{\mathrm{pool}})| \leq t$ and $|\mathrm{supp}(\mu_{X,x}^{\mathrm{act}})| \leq t$. In contrast, across all contexts and lengths, the reachable atom set

$$\mathcal{A}_\theta := \bigcup_{t \geq 1} \bigcup_{X \in \mathbb{R}^{t \times d}} \bigcup_{x \in \mathbb{R}^{1 \times d}} \mathrm{supp}(\mu_{X,x}^{\mathrm{pool}}) \subseteq \Omega$$

can be infinite (indeed uncountable under mild conditions). Thus "infinite dictionary" refers to capacity across contexts, not to per-step computation.

### E.3. Activation gating as restriction and complexity control via TV/variation norms

We now connect the mechanistic decomposition to the functional-analytic framework. To avoid pathologies from $\Omega = \mathbb{R}^{2d}$ being unbounded, we follow the previous section and work on a reachable compact set of atoms.

**Reachable atom set.** Assume $(x, X)$ range over bounded sets and the mixing operators are uniformly bounded in operator norm; then every instantiated atom $(u_j, v_j)$ lies in a compact set $\Omega_t \subseteq \Omega$ (as shown in the previous section). Let $\mathcal{M}(\Omega_t)$ be the Banach space of finite signed Radon measures on $\Omega_t$ with total variation norm $\|\cdot\|_{\mathrm{TV}}$.

**Pointwise linear functionals.** For each fixed $x \in \mathcal{X}$, define the linear functional on measures

$$\ell_{\theta,x} : \mathcal{M}(\Omega_t) \to \mathbb{R}^{1 \times d}, \qquad \ell_{\theta,x}(\mu) := \int_{\Omega_t} \Psi_\theta(x; \omega) \, d\mu(\omega),$$

(where the integral is understood componentwise, equivalently as a Bochner integral). Linearity follows immediately from linearity of the integral in $\mu$.

**Lemma E.6** (Restriction (activation gating) is a TV contraction). *Let $\mu \in \mathcal{M}(\Omega_t)$ and let $g : \Omega_t \to \mathbb{R}$ be measurable with $|g| \leq 1$. Define $(g\mu)(A) := \int_A g \, d\mu$. Then*
$$\|g\mu\|_{\mathrm{TV}} \leq \|\mu\|_{\mathrm{TV}}.$$

*In particular, $\|\mu_{X,x}^{\mathrm{act}}\|_{\mathrm{TV}} \leq \|\mu_{X,x}^{\mathrm{pool}}\|_{\mathrm{TV}}$.*

*Proof.* This is the standard total-variation contraction under multiplication by a bounded function: for any measurable partition $\{A_i\}$,

$$\sum_i |(g\mu)(A_i)| = \sum_i \left| \int_{A_i} g \, d\mu \right| \leq \sum_i \int_{A_i} |g| \, d|\mu| \leq \sum_i \int_{A_i} 1 \, d|\mu| = |\mu|(\Omega_t) = \|\mu\|_{\text{TV}}.$$

Taking the supremum over partitions yields the claim. □

*Remark* E.7 (Atomic measures: TV equals pool size). For the unweighted atomic Context-instantiated Memory Pool measure $\mu_{X,x}^{\text{pool}} = \sum_{j=1}^{t} \delta_{(u_j, v_j)}$,

$$\|\mu_{X,x}^{\text{pool}}\|_{\text{TV}} = t, \qquad \|\mu_{X,x}^{\text{act}}\|_{\text{TV}} = \#\{j \in [t] : \alpha_\theta(x; u_j) > 0\},$$

i.e., TV counts the number of instantiated (resp. activated) slots, including multiplicities.

**Proposition E.8** (A priori output bound via TV mass). *Assume $\Psi_\theta$ is bounded on $\mathcal{X} \times \Omega_t$ with $\|\Psi_\theta\|_{\infty,2} := \sup_{x \in \mathcal{X}, \omega \in \Omega_t} \|\Psi_\theta(x; \omega)\|_2 < \infty$. Then for any $(X, x)$ with $\text{supp}(\mu_{X,x}^{\text{pool}}) \subseteq \Omega_t$,*

$$\|\ell_{\theta,x}(\mu_{X,x}^{\text{pool}})\|_2 \leq \|\Psi_\theta\|_{\infty,2} \|\mu_{X,x}^{\text{pool}}\|_{\text{TV}} = t \|\Psi_\theta\|_{\infty,2}.$$

*Moreover, by TV contraction,*

$$\|\ell_{\theta,x}(\mu_{X,x}^{\text{act}})\|_2 \leq \|\Psi_\theta\|_{\infty,2} \|\mu_{X,x}^{\text{act}}\|_{\text{TV}} \leq \|\Psi_\theta\|_{\infty,2} \|\mu_{X,x}^{\text{pool}}\|_{\text{TV}}.$$

*Proof.* For any fixed $x$,

$$\|\ell_{\theta,x}(\mu)\|_2 = \left\| \int_{\Omega_t} \Psi_\theta(x; \omega) \, d\mu(\omega) \right\|_2 \leq \int_{\Omega_t} \|\Psi_\theta(x; \omega)\|_2 \, d|\mu|(\omega) \leq \|\Psi_\theta\|_{\infty,2} |\mu|(\Omega_t) = \|\Psi_\theta\|_{\infty,2} \|\mu\|_{\text{TV}}.$$

Apply to $\mu_{X,x}^{\text{pool}}$ and $\mu_{X,x}^{\text{act}}$ and use Lemma E.6. □

*Remark* E.9 (Variation-norm interpretation (static vs. dynamic instantiation)). When the sequence-space mixings $R_\theta^{(1)}, R_\theta^{(2)}$ are static in $x$, the pool measure $\mu_X^{\text{pool}}$ depends only on $X$, and the map $x \mapsto \ell_{\theta,x}(\mu_X^{\text{pool}})$ lies in the range of the bounded linear operator $T_{\theta,t} : \mathcal{M}(\Omega_t) \to L_\infty(\mathcal{X}; \mathbb{R}^d)$ defined by $(T_{\theta,t}\mu)(x) := \ell_{\theta,x}(\mu)$. In this regime, the induced quotient (variation) norm satisfies $\|T_{\theta,t}\mu\|_{\text{var}} \leq \|\mu\|_{\text{TV}}$, so the instantiated pool size $t$ provides an explicit complexity bound.

In the fully dynamic regime, $\mu_{X,x}^{\text{pool}}$ may depend on $x$, so the representation is best understood pointwise: for each fixed $x$, the readout is a bounded linear functional of the instantiated measure, and activation remains a TV contraction. This preserves the mechanistic interpretation (space/pool/activation) while the "function-class" viewpoint requires handling input-dependent measures.

### E.4. Long-context behavior as enlarging candidate pools (not enforcing probabilities)

The dictionary–measure decomposition clarifies a long-context mechanism that is orthogonal to probability normalization.

**Candidate pool growth.** At step $t$, the model instantiates a Context-instantiated Memory Pool with at most $t$ atoms and then selects a Current-step Activated Memory subpool. Thus increasing context length enlarges the candidate pool (the number of available slots) even before activation:

$$\|\mu_{X,x}^{\text{pool}}\|_{\text{TV}} = t, \qquad \|\mu_{X,x}^{\text{act}}\|_{\text{TV}} \leq t.$$

**A minimal monotonicity principle (under extension consistency).** To formalize the statement "longer contexts cannot reduce expressivity," one may impose an extension-consistency condition: for $t < T$, there exists an embedding $\iota_{t \to T} : \mathbb{R}^{t \times d} \to \mathbb{R}^{T \times d}$ such that the length-$T$ block can ignore the additional $T - t$ tokens (e.g., by padding zeros and choosing sequence mixings that act as identity on the first $t$ slots and do not mix in the padded slots). Under this condition, any computation realizable at length $t$ is realizable at length $T$ by an appropriate extension, so the attainable set of outputs expands with $t$.

**Role of sequence-space mixing.** The instantiation map $\Gamma_\theta(x; X)$ (Lemma E.2) makes explicit how sequence-space mixing changes which atoms are reachable from a given context: $R_\theta^{(1)}(x)$ and $R_\theta^{(2)}(x)$ alter $(u_j, v_j)$ by mixing the rows of $X$ before they are used for addressing and readout. Learning these mixings therefore modifies the reachable Context-instantiated Memory Pool and, in the fully dynamic regime, deforms the activation geometry (cf. the "warped partition" analysis earlier), providing a direct mechanism for expressivity gains that does not rely on preserving any probabilistic interpretation of weights.

### E.5. Connection back to NTK: stable activation under small parameter movement

The NTK/lazy-training discussion implies a complementary mechanistic statement: when pre-activations stay away from zero by a margin, the Current-step Activated Memory is stable.

*Remark* E.10 (Gate stability implies activated-pool stability). Fix $(X, x)$ and consider the pre-activation vector $h(\theta) :=$ $h(x; X)$. If there exists $\delta > 0$ such that $\min_{j \in [t]} |h(\theta_0)_j| \geq \delta$ at some reference $\theta_0$ and $\|h(\theta) - h(\theta_0)\|_\infty < \delta$, then the sign pattern $\mathbf{1}\{h(\theta) > 0\}$ is unchanged. Equivalently, the restriction $\mu_{X,x}^{\text{act}} = g_x \mu_{X,x}^{\text{pool}}$ selects the same subset of atoms. Thus, in regimes where training induces small changes in $h$ on the data (as in lazy/NTK analyses), the model's routing/selection over the Context-instantiated Memory Pool can remain stable.

**Takeaway.** The block can be viewed as operating on a Global (parameter-defined) Memory Space $\Omega$ specified by parameters, instantiating a finite Context-instantiated Memory Pool via $\mu_{X,x}^{\text{pool}}$, and selecting a Current-step Activated Memory subpool via the restriction $\mu_{X,x}^{\text{act}} = g_x \mu_{X,x}^{\text{pool}}$. The Banach/TV tools quantify this activation as a contraction (complexity does not increase under gating), while the NTK margin condition quantifies when the activated pool remains stable over training.

## F. Parameterization of HyperMLP

### F.1. Offset (lag) Layout

**Theorem F.1** (Offset (lag) layout aligns canonical extension with autoregressive truncation). *Fix integers $1 \leq t < T$. Let $\widetilde{X}^\rightarrow \in \mathbb{R}^{T \times d}$ denote a length-$T$ history in forward order (oldest→newest),*

$$\widetilde{X}^\rightarrow := X_{1:T} = \begin{bmatrix} x_1 \\ \vdots \\ x_T \end{bmatrix},$$

*and let $\widetilde{X}^\leftarrow \in \mathbb{R}^{T \times d}$ denote the same history in offset/lag order (newest→oldest),*

$$\widetilde{X}^\leftarrow := X_{T:1} = \begin{bmatrix} x_T \\ x_{T-1} \\ \vdots \\ x_1 \end{bmatrix}.$$

*For each length $\ell \in \{t, T\}$, consider the (one-head) dynamic block map*

$$h_\ell(x; X) := x\, L^{(1)}(x)\, X^\top R_\ell^{(1)}(x) \in \mathbb{R}^{1 \times \ell}, \qquad o_\ell(x; X) := \sigma_\ell\big(h_\ell(x; X)\big)\, R_\ell^{(2)\top}(x)\, X\, L^{(2)\top}(x), \qquad (48)$$

*where $L^{(1)}(\cdot), L^{(2)}(\cdot) \in \mathbb{R}^{d \times d}$ are length-independent feature-space mixings, and $R_\ell^{(1)}(\cdot), R_\ell^{(2)}(\cdot) \in \mathbb{R}^{\ell \times \ell}$ are length-$\ell$ sequence-space mixings.*

*The activation is $\sigma_\ell(z) = \text{ReLU}(\text{L2Norm}_\ell(z))$ applied elementwise across the length-$\ell$ vector. Assume $\text{L2Norm}_\ell$ is a positive scalar rescaling with padding invariance: there exists $\varepsilon > 0$ such that*

$$\text{L2Norm}_\ell(z) = \frac{z}{\rho_\ell(z)}, \qquad \rho_\ell(z) := \sqrt{\|z\|_2^2 + \varepsilon}, \qquad z \in \mathbb{R}^{1 \times \ell}, \qquad (49)$$

*so in particular $\rho_\ell(z) > 0$ and $\rho_T([z, 0]) = \rho_t(z)$ for all $z \in \mathbb{R}^{1 \times t}$.*

*Let the canonical injection be*

$$P_{t\to T} := \begin{bmatrix} I_t \\ 0 \end{bmatrix} \in \mathbb{R}^{T\times t}, \qquad P_{t\to T}^\top = \begin{bmatrix} I_t & 0 \end{bmatrix}.$$

*Assume the sequence-space mixings are prefix-extension consistent: for each $m \in \{1,2\}$ and all $x$,*

$$R_T^{(m)}(x) \;=\; P_{t\to T}\, R_t^{(m)}(x)\, P_{t\to T}^\top. \tag{50}$$

*Then the following hold.*

*(i) Offset layout gives autoregressive truncation invariance. Let $X^\leftarrow := P_{t\to T}^\top \widetilde{X}^\leftarrow = \widetilde{X}^\leftarrow[1\!:\!t,:]$ (the most recent $t$ tokens, in lag order). Then for all $x$,*

$$o_T(x; \widetilde{X}^\leftarrow) \;=\; o_t(x; X^\leftarrow).$$

*Equivalently, extending the context by appending older tokens (far past) can be made to have exactly zero effect.*

*(ii) The same canonical extension yields the wrong invariance under forward layout. Let $X^\rightarrow := P_{t\to T}^\top \widetilde{X}^\rightarrow = \widetilde{X}^\rightarrow[1\!:\!t,:]$ (the oldest $t$ tokens, in forward order). Then for all $x$,*

$$o_T(x; \widetilde{X}^\rightarrow) \;=\; o_t(x; X^\rightarrow),$$

*i.e., invariance to keeping the oldest $t$ tokens (dropping the newest), not invariance to keeping the most recent $t$ tokens.*

*Consequently, for any parameterization that is naturally prefix-stationary across lengths (i.e., implements $R_T$ by embedding $R_t$ into the top-left block via (50), as in many length-agnostic constructions), the offset/lag layout is the coordinate system in which this canonical extension-consistency matches autoregressive truncation semantics:*

*"add more far-past tokens" $\iff$ "append rows at the end" (no reindexing of recent lags).*

*Proof.* We prove the more general identity that under (50),

$$o_T(x; \widetilde{X}) = o_t\big(x; P_{t\to T}^\top \widetilde{X}\big) \qquad \forall x, \; \forall \widetilde{X} \in \mathbb{R}^{T\times d}, \tag{51}$$

from which (i) and (ii) follow by choosing $\widetilde{X} = \widetilde{X}^\leftarrow$ or $\widetilde{X} = \widetilde{X}^\rightarrow$ and interpreting the truncation.

Fix $\widetilde{X} \in \mathbb{R}^{T\times d}$ and let $X := P_{t\to T}^\top \widetilde{X} \in \mathbb{R}^{t\times d}$. Under (50), for each $m \in \{1,2\}$,

$$R_T^{(m)\top}(x)\, \widetilde{X} = \big(P_{t\to T} R_t^{(m)}(x) P_{t\to T}^\top\big)^\top \widetilde{X} = P_{t\to T}\, R_t^{(m)\top}(x)\, P_{t\to T}^\top \widetilde{X} = P_{t\to T}\, R_t^{(m)\top}(x)\, X.$$

Hence the mixed contexts at length $T$ are zero-padded versions of those at length $t$:

$$U_T := R_T^{(1)\top}(x)\widetilde{X} = \begin{bmatrix} U_t \\ 0 \end{bmatrix}, \qquad V_T := R_T^{(2)\top}(x)\widetilde{X} = \begin{bmatrix} V_t \\ 0 \end{bmatrix},$$

where $U_t := R_t^{(1)\top}(x)X$ and $V_t := R_t^{(2)\top}(x)X$.

Using $\widetilde{X}^\top R_T^{(1)}(x) = U_T^\top$ and $X^\top R_t^{(1)}(x) = U_t^\top$, the pre-activations satisfy

$$h_T(x; \widetilde{X}) = x\, L^{(1)}(x)\, \widetilde{X}^\top R_T^{(1)}(x) = x\, L^{(1)}(x)\, U_T^\top = \big[x\, L^{(1)}(x)\, U_t^\top,\; 0\big] = \big[h_t(x; X),\; 0\big].$$

By (49), $\sigma_\ell(z) = \rho_\ell(z)^{-1}\mathrm{ReLU}(z)$ and $\rho_T([h_t, 0]) = \rho_t(h_t)$, hence

$$\sigma_T\big(h_T(x; \widetilde{X})\big) = \frac{1}{\rho_T([h_t, 0])} \left[\mathrm{ReLU}(h_t), 0\right] = \big[\sigma_t(h_t(x; X)),\; 0\big].$$

Therefore,

$$\sigma_T(h_T)\, V_T = [\sigma_t(h_t), 0] \begin{bmatrix} V_t \\ 0 \end{bmatrix} = \sigma_t(h_t)\, V_t.$$

Multiplying the (shared) feature-space readout $L^{(2)\top}(x)$ yields

$$o_T(x; \widetilde{X}) = \sigma_T(h_T)\, V_T\, L^{(2)\top}(x) = \sigma_t(h_t)\, V_t\, L^{(2)\top}(x) = o_t(x; X),$$

which proves (51). Statements (i) and (ii) follow immediately by substituting $X = P_{t\to T}^\top \widetilde{X}^\leftarrow$ and $X = P_{t\to T}^\top \widetilde{X}^\rightarrow$, respectively. $\square$

*Remark* F.2 (Why this is the right notion of "better"). The theorem does not claim that forward order is intrinsically incapable of representing the same functions. Rather, it isolates a precise coordinate mismatch under the canonical (top-left) extension (50): the identity $o_T(x; \widetilde{X}) = o_t(x; P_{t \to T}^\top \widetilde{X})$ always keeps the prefix of the row ordering. Under lag order, this prefix is the most recent tokens (autoregressive truncation window); under forward order, it is the oldest tokens. Forward order can recover the same autoregressive semantics by using a suffix embedding $Q_{t \to T} = \begin{bmatrix} 0 \\ I_t \end{bmatrix}$ and the corresponding extension rule $R_T^{(m)}(x) = Q_{t \to T} R_t^{(m)}(x) Q_{t \to T}^\top$, but this requires an explicit choice that is not the canonical top-left embedding.

## F.2. Parameter Budget

### F.2.1. RANK ALLOCATION IN A RESIDUAL TWO-LAYER MLP: COMPRESS $W_1$ OR $W_2$?

We consider the residual MLP block (row-vector convention)

$$f(x) = x + \sigma(xW_1)W_2, \tag{52}$$

where $x \in \mathbb{R}^{1 \times d}$, $W_1 \in \mathbb{R}^{d \times h}$, $W_2 \in \mathbb{R}^{h \times d}$, and $\sigma : \mathbb{R}^{1 \times h} \to \mathbb{R}^{1 \times h}$ is an arbitrary (possibly nonlinear) map (e.g., $\sigma = \text{ReLU} \circ \text{Norm}$). We identify $\mathbb{R}^{1 \times d}$ with $\mathbb{R}^d$ equipped with the standard Euclidean inner product.

**Theorem F.3** (Output-subspace invariance under low-rank $W_2$). *Assume* $\text{rank}(W_2) \le r$ *and define* $U := \text{Row}(W_2) \subseteq \mathbb{R}^d$. *Then for all* $x$,

$$f(x) - x \in U. \tag{53}$$

*Equivalently, letting* $\Pi_{U^\perp}$ *be the orthogonal projection onto* $U^\perp$,

$$\Pi_{U^\perp} f(x) = \Pi_{U^\perp} x \qquad \forall x. \tag{54}$$

*In particular, the MLP branch leaves invariant a subspace of dimension* $\dim(U^\perp) = d - \text{rank}(W_2) \ge d - r$.

*Proof.* Let $\Delta(x) := f(x) - x = \sigma(xW_1)W_2$. For any $u \in \mathbb{R}^{1 \times h}$, the product $uW_2$ is a linear combination of the rows of $W_2$, hence $uW_2 \in \text{Row}(W_2) = U$. Taking $u = \sigma(xW_1)$ yields $\Delta(x) \in U$, proving (53). Applying $\Pi_{U^\perp}$ gives (54). The dimension claim follows from $\dim(U) = \text{rank}(W_2) \le r$. □

**Corollary F.4** (Invariant linear functionals). *Under the assumptions of Theorem F.3, for any* $v \in U^\perp$ *and any* $x$, $\langle f(x), v \rangle = \langle x, v \rangle$.

*Proof.* Since $f(x) - x \in U$ and $v \in U^\perp$, we have $\langle f(x) - x, v \rangle = 0$. □

**Theorem F.5** (Quotient structure under low-rank $W_1$). *Assume* $\text{rank}(W_1) \le r$, *i.e.,* $W_1$ *admits a factorization* $W_1 = LR$ *with* $L \in \mathbb{R}^{d \times r}$ *and* $R \in \mathbb{R}^{r \times h}$. *Then there exists a function* $\psi : \mathbb{R}^{1 \times r} \to \mathbb{R}^{1 \times d}$ *such that*

$$f(x) = x + \psi(xL) \qquad \forall x. \tag{55}$$

*Consequently, for any* $\delta \in \mathbb{R}^{1 \times d}$ *satisfying* $\delta W_1 = 0$ *(equivalently,* $\delta \in \text{Null}(W_1^\top)$*),*

$$f(x + \delta) = f(x) + \delta \qquad \forall x. \tag{56}$$

*Moreover,* $\text{Null}(W_1^\top)$ *has dimension* $d - \text{rank}(W_1) \ge d - r$.

*Proof.* Substitute $W_1 = LR$ into (52): $f(x) = x + \sigma((xL)R)W_2$. Define $\psi(p) := \sigma(pR)W_2$, giving (55). If $\delta W_1 = 0$, then $(x + \delta)W_1 = xW_1$ and thus $\sigma((x + \delta)W_1) = \sigma(xW_1)$, i.e., $f(x + \delta) = x + \delta + \sigma(xW_1)W_2 = f(x) + \delta$. The dimension statement is standard: $\dim \text{Null}(W_1^\top) = d - \text{rank}(W_1)$. □

**Discussion.** Theorems F.3 and F.5 reveal a structural asymmetry: low-rank $W_2$ enforces a hard restriction on the output update $\Delta(x)$, confining it to a fixed subspace $U$, whereas low-rank $W_1$ enforces a restriction on the conditioning variables of the update, since $\Delta(x)$ depends only on $xL$. In residual blocks, the identity shortcut transmits all components of $x$ unchanged, so restricting the conditioning (via $W_1$) is often empirically less damaging than restricting the action subspace (via $W_2$), although the relative impact can still be task-dependent.

F.2.2. Parameter Budget: Paying for Temporal Mixing by Shrinking the First (QK) Layer

HyperMLP introduces additional sequence-space capacity through the learned mixing operators $R^{(1)}(x)$ and $R^{(2)}(x)$, implemented via a diagonal-plus-low-rank parameterization. Conceptually, we view this added capacity as a benefit rather than a drawback. In an ideal setting, all mixing operators would be parameterized directly with respect to the token dimension, without aggressive rank compression. From this viewpoint, allocating more parameters to attention or HyperMLP blocks than to static MLP blocks is a natural and desirable choice, since sequence modeling inherently requires richer, context-conditioned structure.

However, for the purpose of controlled and fair empirical comparisons, we adopt a fixed parameter budget in our experiments. Under this constraint, the practical question becomes: which part of the dynamic two-layer block should be compressed to make room for learnable temporal mixing?

The dynamic-MLP view makes the answer immediate. Per head, the block is a residual depth-two map whose first layer instantiates addressing/selection (the "QK" path that produces the length-$t$ hidden vector), and whose second layer instantiates readout/action (the "VO" path that maps the selected hidden coordinates back into $\mathbb{R}^d$). The rank-allocation results above formalize a strong asymmetry: in a residual two-layer block $f(x) = x + \sigma(xW_1)W_2$, a low-rank $W_2$ enforces an output-subspace invariance (Theorem F.3), i.e., the update $f(x) - x$ is confined to a fixed low-dimensional subspace. In contrast, a low-rank $W_1$ induces a quotient structure (Theorem F.5): the update depends only on a low-dimensional projection of the input, while the residual connection preserves all other directions unchanged. Translated to our setting, shrinking the second-layer rank $d_{vo}$ would directly restrict the space of possible token updates, whereas shrinking the first-layer rank $d_{qk}$ primarily limits the conditioning variables used for routing over the (context-instantiated) pool. Therefore, to "pay" for temporal mixing while preserving action capacity, we reduce the first-layer/QK rank.

Concretely, throughout our experiments we use the following allocation:

$$d_{qk} = \frac{d}{4\,n_{\text{head}}}, \qquad d_{vo} = \frac{d}{n_{\text{head}}}, \qquad r_s = 16, \tag{57}$$

so that the first-layer core is $4\times$ narrower than the second-layer core (per head), freeing capacity for the two sequence-mixing operators. Finally, since each head incurs an additional $O(t\,r_s)$ sequence-mixing cost (Appendix G), we fix $n_{\text{head}} = 2$ for HyperMLP/HyperGLU, i.e., we use two parallel dynamic-MLP heads and aggregate them in the standard multi-head manner.

### F.3. Why HyperGLU: Decoupling Routing (Selection) from Slot Strength (Weighting)

**Motivation from the ReLU gating identity.** A depth-two ReLU MLP reuses the same first-layer pre-activation to serve two roles. In our row-vector convention, for $z \in \mathbb{R}^{1 \times t}$ we can write $\text{ReLU}(z) = z\,D(z)$ where $D(z) = \text{diag}(\mathbf{1}\{z_1 > 0\}, \ldots, \mathbf{1}\{z_t > 0\})$. Thus the sign pattern of $z$ selects which hidden coordinates are active, while the values of $z$ on the active coordinates also determine their coefficients in the readout. This coupling is fundamental to piecewise-linear networks, but in dynamic, context-instantiated blocks it can be useful to separate "which slots are used" from "how strongly they are used".

**Coupling in HyperMLP under the measure view.** Under the standing assumption that $\text{L2Norm}_t$ is a positive scalar rescaling across the length-$t$ vector (Assumption B.1), Theorem C.5 gives the HyperMLP readout

$$o_t = \frac{1}{\rho_t(h_t)} \sum_{i=1}^{t} \alpha_\theta(x_t; u_i)_+ \beta_\theta(x_t; v_i), \qquad \alpha_\theta(x_t; u_i) = (h_t)_i, \tag{58}$$

where $(u_i, v_i)$ are the slots instantiated from the mixed contexts $U = R_\theta^{(1)\top}(x_t)X_{t:1}$ and $V = R_\theta^{(2)\top}(x_t)X_{t:1}$, and $\beta_\theta(x_t; v_i) = v_i L_\theta^{(2)\top}(x_t)$. Equation (58) makes the coupling explicit: the activated set is determined by the sign pattern $\mathbf{1}\{\alpha_\theta(x_t; u_i) > 0\}$, but the same scalar $\alpha_\theta(x_t; u_i)$ also sets the magnitude through $\alpha_+$.

**HyperGLU: factorizing selection and strength modulation.** HyperGLU introduces two first-layer score vectors,

$$h_t^{\text{gate}} := x_t W_{\text{MLP,gate}}^{(1)}(X_{t:1}) \in \mathbb{R}^{1 \times t}, \qquad h_t^{\text{scale}} := x_t W_{\text{MLP,scale}}^{(1)}(X_{t:1}) \in \mathbb{R}^{1 \times t}, \tag{59}$$

and defines the hidden activation by

$$a_t := \text{Softplus}(h_t^{\text{scale}}) \odot \text{ReLU}(\text{L2Norm}_t(h_t^{\text{gate}})), \qquad o_t := a_t \, W_{\text{MLP}}^{(2)}(X_{t:1}). \tag{60}$$

Because $\text{L2Norm}_t$ is a positive scalar rescaling, it does not change signs (Lemma B.2). Therefore the activated set is determined only by the sign pattern of $h_t^{\text{gate}}$, while $\text{Softplus}(h_t^{\text{scale}}) > 0$ provides an additional, independent modulation of the strengths of the active slots.

Using the same sign-invariance assumption, (60) expands to

$$o_t = \frac{1}{\rho_t(h_t^{\text{gate}})} \sum_{i=1}^{t} \text{Softplus}(h_{t,i}^{\text{scale}}) \, (h_{t,i}^{\text{gate}})_+ \, \beta_\theta(x_t; v_i), \tag{61}$$

which shows that HyperGLU keeps the same slot-selection mechanism (through $(h_{t,i}^{\text{gate}})_+$) and adds a separate positive multiplier for each selected slot.

**Implementation via splitting the first-layer core rank.** In our implementation, $h_t^{\text{gate}}$ and $h_t^{\text{scale}}$ are produced by splitting the HyperMLP first-layer feature core along the rank dimension (two $d_{qk}/2$ cores, equivalently doubling the first-layer head count), while using the same second-layer instantiation $W_{\text{MLP}}^{(2)}(X_{t:1})$. This preserves the factorized dynamic-weight structure and adds a targeted increase in flexibility where the routing and coefficient formation occur.

**Connection to stability of the activated set.** Since the activated set depends only on the sign pattern of $h_t^{\text{gate}}$, any margin condition of the form $\min_i |(h_t^{\text{gate}})_i| \geq \delta$ implies stability of the selected slots under sufficiently small perturbations (Lemma D.11). In such regimes, the scale branch can still adjust slot strengths without requiring changes to the selection boundaries, which provides a concrete mechanism by which HyperGLU can improve optimization and performance.

**Takeaway.** HyperGLU addresses a specific coupling in ReLU-style dynamic blocks: the same quantity both selects slots and sets their coefficients. By introducing a separate scale branch, HyperGLU keeps ReLU-based selection while enabling independent, smooth control of the strength of selected slots, mirroring the practical benefits of GLU-style modulation in standard Transformer MLPs.

# G. Computational Overhead of HyperMLP vs. (ReLU/Softmax) Attention

This section summarizes the idealized computational overhead introduced by HyperMLP/HyperGLU compared to standard quadratic attention heads (either softmax-normalized or ReLU-normalized), under the intended regime $r_s \ll d_{qk}, d_{vo}$ and assuming we never materialize dense $t \times t$ sequence-mixing matrices.

**Setup (one head).** At step $t$, let the lag-ordered prefix be $X_{t:1} = [x_t; \dots; x_1] \in \mathbb{R}^{t \times d}$. Let $d_{qk}$ and $d_{vo}$ denote the per-head feature ranks (typically $d_{qk} \approx d_{vo} \approx d/n_{\text{head}}$), and let $r_s$ be the rank of sequence-space mixing. Write

$$z_t := x_t \, L^{(1)}(x_t) \, X_{t:1}^\top \in \mathbb{R}^{1 \times t}, \tag{62}$$

so that vanilla attention corresponds to reading out from $z_t$ (with $\sigma = \text{softmax}$ or ReLU-based normalization) without explicit sequence-space mixing.

## G.1. Baseline: quadratic attention (softmax or ReLU-normalized)

A standard quadratic attention head (softmax or ReLU-normalized) can be written in the dynamic-MLP form

$$o_t^{\text{attn}} = \sigma(z_t) \, X_{t:1} \, L^{(2)\top}(x_t), \qquad \sigma = \text{softmax or ReLU}(\text{L2Norm}_t(\cdot)). \tag{63}$$

The dominant work is forming the length-$t$ score vector $z_t$ and aggregating values:

$$\text{Time}_{\text{attn}}(t) = O(t \, d_{qk}) + O(t \, d_{vo}) + O(t), \tag{64}$$

where the $O(t)$ term accounts for softmax (exp + normalization) or for ReLU(L2Norm). Over a length-$T$ sequence (teacher forcing), the arithmetic cost is

$$\text{Time}_{\text{train}}^{\text{attn}}(T) = O(T^2 \, d_{qk}) + O(T^2 \, d_{vo}), \tag{65}$$

and modern implementations (e.g. FlashAttention-style kernels) can avoid storing the full $T \times T$ score matrix.

## G.2. HyperMLP: additional sequence-space mixing

HyperMLP inserts two sequence-space operators (one before and one after $\sigma$):

$$h_t \;:=\; z_t\, R_t^{(1)}(x_t), \qquad w_t \;:=\; \sigma(h_t)\, R_t^{(2)\top}(x_t), \qquad o_t \;:=\; w_t\, X_{t:1}\, L^{(2)\top}(x_t). \tag{66}$$

Each $R_t^{(j)}(x_t) \in \mathbb{R}^{t \times t}$ uses the diagonal-plus-low-rank (DPLR) parameterization

$$R_t^{(j)}(x_t) \;=\; D_{1:t}^{(j)} \;+\; A_{1:t}^{(j)}\,\mathrm{Diag}(s_t^{(j)})\, B_{1:t}^{(j)\top}, \qquad s_t^{(j)} := \phi(x_t W_S^{(j)}) \in \mathbb{R}^{r_s}, \tag{67}$$

where $D_{1:t}^{(j)}$ is diagonal and $A_{1:t}^{(j)}, B_{1:t}^{(j)} \in \mathbb{R}^{t \times r_s}$ are the lag-prefix slices of learnable parameters.

### G.2.1. A BASIC DPLR MULTIPLICATION IDENTITY

**Lemma G.1** (Vector–DPLR multiplication). *Let $R = D + A\,\mathrm{Diag}(s)\,B^\top \in \mathbb{R}^{t \times t}$ with $D$ diagonal, $A, B \in \mathbb{R}^{t \times r_s}$ and $s \in \mathbb{R}^{r_s}$. Then for any $y \in \mathbb{R}^{1 \times t}$,*

$$yR = yD + \big((yA) \odot s^\top\big) B^\top, \qquad yR^\top = yD + \big((yB) \odot s^\top\big) A^\top. \tag{68}$$

*Therefore $yR$ (or $yR^\top$) can be computed in $O(t r_s)$ time using $O(r_s)$ additional temporary memory (beyond storing $y$).*

*Proof.* Associativity gives $yASB^\top = (yA)\mathrm{Diag}(s)B^\top = ((yA) \odot s^\top)B^\top$, and similarly for $R^\top$. The cost is dominated by the two matrix–vector products $yA$ and $(\cdot)B^\top$. $\quad\square$

## G.3. Autoregressive inference overhead

**Proposition G.2** (DPLR temporal mixing keeps attention-style scaling complexity). *Per head, HyperMLP adds two sequence-slot mixes $y \mapsto yR_t^{(1)}(x_t)$ and $y \mapsto yR_t^{(2)\top}(x_t)$ with $R_t^{(j)}(x_t) = D^{(j)} + A^{(j)}S^{(j)}(x_t)B^{(j)\top}$ (rank $r_s$). Without materializing $t \times t$ matrices:*

$$\mathrm{Time}_{\mathrm{HyperMLP}}(t) = O\big(t(d_{qk} + d_{vo})\big) + O(t r_s), \tag{69}$$

$$\mathrm{ExtraState} = O(r_s) \ \text{(per head, per step)}. \tag{70}$$

*Over length-$T$ teacher forcing:*

$$\mathrm{Time}_{\mathrm{train}}^{\mathrm{HyperMLP}}(T) = O\big(T^2(d_{qk} + d_{vo})\big) + O(T^2 r_s), \tag{71}$$

*so under $r_s \ll d_{qk}, d_{vo}$ the overhead is lower-order and the asymptotic scaling matches quadratic attention.*

*Proof sketch.* Apply the vector-DPLR identity (Lemma G.1) twice per step to obtain $O(t r_s)$ time and $O(r_s)$ temporary memory, and sum $\sum_{t=1}^{T} O(t r_s) = O(T^2 r_s)$. See Propositions G.3–G.4 and Table 3. As an implementation remark, the low-rank sequence mixing induced by $R$ admits an alternative convolution/FFT-based realization in $O(T \log T)$ time for long kernels (Remark G.5), without changing the asymptotic conclusions here. $\quad\square$

**Proposition G.3** (Inference time and overhead (one head)). *Assume a standard KV cache stores the lag-ordered projected keys/values (or equivalent sufficient statistics) so that forming $z_t$ in (62) costs $O(t\, d_{qk})$ and the value aggregation in (66) costs $O(t\, d_{vo})$. Then HyperMLP adds two DPLR multiplications of length $t$ vectors, hence by Lemma G.1,*

$$\mathrm{Time}_{\mathrm{HyperMLP}}(t) = \mathrm{Time}_{\mathrm{attn}}(t) + O(t r_s), \qquad \text{(per head, per step)}. \tag{72}$$

*Moreover, beyond the standard KV cache, the only step-dependent quantities needed for the DPLR mixes are the gate vectors $s_t^{(1)}, s_t^{(2)} \in \mathbb{R}^{r_s}$ (and, for HyperGLU, a constant-factor number of such vectors), i.e.*

$$\mathrm{ExtraState}_{\mathrm{HyperMLP}} = O(r_s) \quad \text{per head.} \tag{73}$$

*Proof sketch.* Compute $h_t = z_t R_t^{(1)}(x_t)$ and $w_t = \sigma(h_t)R_t^{(2)\top}(x_t)$ using (68). All other operations are the same order as in (63). $\quad\square$

We omit $t$-independent terms such as computing $s_t^{(j)}$ and feature-side gates, which are comparable to standard projection costs.

|  | **Inference (step $t$)** | **Training (length $T$)** |
|---|---|---|
| Softmax/ReLU attention | $O\big(t(d_{qk} + d_{vo})\big)$ | $O\big(T^2(d_{qk} + d_{vo})\big)$ |
| HyperMLP (one head) | $O\big(t(d_{qk} + d_{vo} + r_s)\big)$ | $O\big(T^2(d_{qk} + d_{vo} + r_s)\big)$ |
| **Overhead** | $O(tr_s)$ time, $O(r_s)$ extra state | $O(T^2 r_s)$ time |

*Table 3.* Idealized asymptotic overhead of HyperMLP per head under the DPLR sequence mixing $R = D + A\operatorname{Diag}(s)B^\top$ and without materializing dense $t \times t$ matrices. Softmax vs. ReLU normalization changes only $O(t)$ activation costs.

**Interpretation.** Since standard attention already costs $O(t(d_{qk}+d_{vo}))$ per step, the additional $O(tr_s)$ term is a lower-order overhead when $r_s \ll d_{qk}, d_{vo}$, and HyperMLP preserves the same asymptotic inference scaling.

### G.4. Full-sequence training overhead

**Proposition G.4** (Training-time overhead (teacher forcing, one head)). *Let the sequence length be $T$. Compared to standard quadratic attention (softmax or ReLU-normalized), HyperMLP introduces two DPLR mixes at each step $t$. Using Lemma G.1, the additional arithmetic is*

$$\sum_{t=1}^{T} O(t\,r_s) \;=\; O(T^2 r_s), \tag{74}$$

*so the overall training-time complexity remains quadratic:*

$$\mathrm{Time}_{\mathrm{train}}^{\mathrm{HyperMLP}}(T) = O(T^2\,d_{qk}) \;+\; O(T^2\,d_{vo}) \;+\; O(T^2\,r_s). \tag{75}$$

*In particular, under $r_s \ll d_{qk}, d_{vo}$, HyperMLP preserves the same $O(T^2)$ scaling as quadratic attention up to a modest $r_s$-dependent factor.*

*Proof.* At step $t$, the extra work is two length-$t$ vector–DPLR multiplications, each $O(tr_s)$ by Lemma G.1. Summing over $t = 1, \ldots, T$ yields (74). $\qquad\square$

### G.5. Lag layout and an optional convolution/FFT implementation view

*Remark* G.5 (Prefix–lag contractions are causal convolutions). The lag layout makes the learned factors $A^{(j)}$ and $B^{(j)}$ naturally interpretable as banks of lag kernels. Fix any sequence $(g_t)_{t=1}^{T}$ with $g_t \in \mathbb{R}^d$ and define $G_{t:1} := [g_t; \ldots; g_1] \in \mathbb{R}^{t \times d}$. For each column $k \in [r_s]$,

$$\big(A_{1:t}^\top G_{t:1}\big)[k,:] = \sum_{\ell=1}^{t} A_{\ell,k}\, g_{t-\ell+1}, \tag{76}$$

which is exactly a causal convolution of $(g_t)$ with kernel $\big(A_{\ell,k}\big)_{\ell \geq 1}$ (up to the usual convolution/correlation convention). Therefore the entire family $\{A_{1:t}^\top G_{t:1}\}_{t=1}^{T}$ (and similarly for $B$) can be computed by 1D convolution kernels, or via FFT-based convolution in $O(T \log T \cdot r_s \cdot d)$ arithmetic in the long-kernel regime. This view provides an FFT-friendly implementation of filter-bank contractions induced by $A, B$ when vectorizing over time. Importantly, HyperMLP is designed as an expressive superset of quadratic attention, so the overall training complexity is still dominated by the quadratic attention-style interactions and remains $O(T^2(\cdot))$ as in (75).

## H. Implications of the Attention-as-MLP View

This section collects several mathematically checkable implications of the Global (parameter-defined) Memory Space $\to$ Context-instantiated Memory Pool $\to$ Current-step Activated Memory decomposition under the activations $\mathrm{ReLU}(\mathrm{L2Norm}(\cdot))$ and a GLU variant with $\mathrm{L2Norm}$ stabilization. Throughout, we focus on a single head and suppress parameters when unambiguous.

**Common setup.** We use Appendix B throughout. When we discuss the "no sequence mixing" regime, we specialize to $R_t^{(1)}(x) \equiv R_t^{(2)}(x) \equiv I_t$, so that $u_i = v_i = X[i,:]$.

## H.1. Prompt-controlled instantiation and two-stage selection (no sequence mixing)

**Prompt inclusion as explicit pool instantiation.** Let two documents be represented by context matrices $D_1 \in \mathbb{R}^{t_1 \times d}$ and $D_2 \in \mathbb{R}^{t_2 \times d}$ (in lag order within each block), and write row concatenation as $D_2 \| D_1 \in \mathbb{R}^{(t_2+t_1) \times d}$. Consider the two prompts

$$X^{(A)} := D_2 \| D_1, \qquad X^{(B)} := D_1.$$

In the no-mixing regime, adding $D_2$ changes the pool measure by adding atoms corresponding exactly to the rows of $D_2$.

**Proposition H.1** (Prompt inclusion is additive pool instantiation (no mixing)). *Under* (B), *the pool measures satisfy*

$$\mu_{X^{(A)}}^{\text{pool}} = \mu_{X^{(B)}}^{\text{pool}} + \sum_{i=1}^{t_2} \delta_{(D_2[i,:],\, D_2[i,:])},$$

*and the activated measure satisfies*

$$\mu_{X^{(A)},x}^{\text{act}} = \mu_{X^{(B)},x}^{\text{act}} + \sum_{i=1}^{t_2} \mathbf{1}\{\alpha(x; D_2[i,:]) > 0\}\, \delta_{(D_2[i,:],\, D_2[i,:])}.$$

*Consequently, the output difference decomposes as*

$$o(x; X^{(A)}) - o(x; X^{(B)}) = \underbrace{\left(\frac{1}{\rho(h_A)} - \frac{1}{\rho(h_B)}\right) \sum_{i=1}^{t_1} \alpha(x; D_1[i,:])_+ \, \beta(x; D_1[i,:])}_{\text{pure normalization rescaling}} + \underbrace{\frac{1}{\rho(h_A)} \sum_{i=1}^{t_2} \alpha(x; D_2[i,:])_+ \, \beta(x; D_2[i,:])}_{\text{new pool atoms (then gated)}},$$

*where $h_A := h(x; X^{(A)})$ and $h_B := h(x; X^{(B)})$.*

*Proof.* With no sequence mixing, $(u_i, v_i)$ are exactly the rows of the chosen context matrix; concatenation is disjoint union of row multisets, hence the pool measure adds. The activated measure follows by multiplying each added atom by the hard gate $g_x(u, v) = \mathbf{1}\{\alpha(x; u) > 0\}$. The output decomposition follows by expanding (E.1) and separating the contributions from $D_1$ and $D_2$. $\square$

**Takeaway.** In the no-mixing regime, the user prompt explicitly controls which atoms are instantiated into the Context-instantiated Memory Pool (a manual selection $\Omega \to \mu^{\text{pool}}$), while the model's ReLU gate performs an automatic selection $\mu^{\text{pool}} \to \mu^{\text{act}}$.

*Remark* H.2 ("Infinite document truncation" as a contextualization of pool instantiation). The statement "knowledge not in the prompt is truncated" should be read for the context-read channel: atoms that are not instantiated into $\mu^{\text{pool}}$ cannot contribute through (E.1). One may formalize this as follows. Let $\widetilde{X} \in \mathbb{R}^{T \times d}$ be any longer history and let $X = P_{t \to T}^{\top} \widetilde{X}$ denote the lag-prefix truncation (most recent $t$ rows). In architectures whose sequence-space operators are extension-consistent (cf. Theorem F.1), tokens in the discarded suffix induce only dummy zero slots and contribute exactly zero to the readout. This realizes the heuristic "far past $\equiv$ truncated" within the attention-style memory mechanism, without making claims about knowledge stored elsewhere in the network parameters.

## H.2. Sequence mixing turns token-wise atoms into context-wide slots

**Setup.** Now allow sequence-space mixing as in HyperMLP:

$$h(x; X) := x\, L^{(1)}(x)\, X^{\top} R^{(1)}(x) \in \mathbb{R}^{1 \times t}, \qquad o(x; X) := \sigma(h(x; X))\, R^{(2)\top}(x)\, X\, L^{(2)\top}(x), \tag{77}$$

with the same $\sigma(z) = \text{ReLU}(\text{L2Norm}_t(z))$ as in (B). Define mixed contexts

$$U := R^{(1)\top}(x)X, \qquad V := R^{(2)\top}(x)X,$$

and denote their rows by $u_i := U[i,:]$, $v_i := V[i,:]$.

**Proposition H.3** (Slots are global (context-wide) linear combinations). *For each slot index $i \in [t]$, letting $r_i^{(1)}(x) := R^{(1)}(x)[:, i] \in \mathbb{R}^t$ and $r_i^{(2)}(x) := R^{(2)}(x)[:, i] \in \mathbb{R}^t$ denote the $i$-th columns, we have*

$$u_i = r_i^{(1)}(x)^\top X = \sum_{j=1}^t r_{i,j}^{(1)}(x) \, X[j, :], \qquad v_i = r_i^{(2)}(x)^\top X = \sum_{j=1}^t r_{i,j}^{(2)}(x) \, X[j, :]. \tag{78}$$

*Hence each instantiated atom $(u_i, v_i) \in \Omega$ is constructed from the entire context via an input-conditioned sequence-basis $\{r_i^{(1)}(x), r_i^{(2)}(x)\}_{i=1}^t$, rather than being tied to a single context token.*

*Proof.* Since $U = R^{(1)\top}(x)X$, the $i$-th row satisfies $U[i, :] = e_i^\top R^{(1)\top}(x)X = R^{(1)}(x)[:, i]^\top X$; similarly for $V$. $\square$

**Pool construction as a learned instantiation map.** Define the instantiation map

$$\Gamma(x; X) : [t] \to \Omega, \qquad \Gamma(x; X)(i) := (u_i, v_i),$$

and the counting measure $\nu_t = \sum_{i=1}^t \delta_i$ on $[t]$. Then, as in the general measure formulation,

$$\mu_{X,x}^{\text{pool}} = (\Gamma(x; X))_\# \nu_t, \qquad \mu_{X,x}^{\text{act}} = g_x \, \mu_{X,x}^{\text{pool}}, \quad g_x(u, v) = \mathbf{1}\{\alpha(x; u) > 0\}. \tag{79}$$

**Takeaway.** With sequence mixing, pool instantiation is no longer "one token $\mapsto$ one atom"; instead, the model learns how to construct $t$ candidate slots from the entire context via an input-conditioned sequence basis. The two-stage selection logic $\Omega \to \mu_{X,x}^{\text{pool}} \to \mu_{X,x}^{\text{act}}$ remains intact.

### H.3. Longer contexts and learnable registers enlarge the candidate pool

**Monotone expressivity under extension consistency.** A minimal formal sense in which "more context cannot reduce expressivity" is: any computation realizable at length $t$ can be realized at length $T > t$ by an appropriate extension of the sequence-mixing operators that ignores the extra $T - t$ rows. This is exactly what Theorem F.1 guarantees in lag layout under extension-consistency of $R^{(1)}, R^{(2)}$ and padding-invariant normalization.

**Corollary H.4** (Monotone realizable family in context length (lag layout)). *Assume the hypotheses of Theorem F.1. Then for any $1 \le t < T$, any current input $x$, and any length-$T$ lag-ordered history $\widetilde{X}^\leftarrow$, letting $X^\leftarrow = P_{t \to T}^\top \widetilde{X}^\leftarrow$ (most recent $t$ rows), we have*

$$o_T(x; \widetilde{X}^\leftarrow) = o_t(x; X^\leftarrow).$$

*In particular, any input–output map realizable at length $t$ is realizable at length $T$ (by choosing an extension-consistent $R_T$), so the realizable function family is monotone in context length.*

**Learnable registers as additional pool atoms.** Let $R \in \mathbb{R}^{k \times d}$ be a set of $k$ learnable register tokens (rows), and consider the augmented context

$$X_{\text{aug}} := X \| R \in \mathbb{R}^{(t+k) \times d}.$$

Even if $R$ contains no semantic content initially, it introduces additional candidate slots (and hence additional hidden width).

**Proposition H.5** (Registers enlarge the instantiated pool (no sequence mixing)). *In the no-mixing regime (B), the pool measure for $X_{\text{aug}}$ decomposes as*

$$\mu_{X_{\text{aug}}}^{\text{pool}} = \mu_X^{\text{pool}} + \sum_{i=1}^k \delta_{(R[i,:], \, R[i,:])}.$$

*Thus registers behave exactly like additional learned atoms that are always present in the candidate pool and can be conditionally activated by the ReLU gate.*

*Proof.* Immediate from the definition $\mu_X^{\text{pool}} = \sum_i \delta_{(X[i,:], X[i,:])}$ when $R^{(1)} = R^{(2)} = I$. $\square$

*Remark* H.6 (Registers under sequence mixing). Under sequence mixing, $U = R^{(1)\top}(x)X_{\text{aug}}$ and $V = R^{(2)\top}(x)X_{\text{aug}}$, so each instantiated slot is a learned mixture of both data tokens and register rows. From (78), registers expand the span of feasible $(u_i, v_i)$ and provide additional learnable degrees of freedom in the constructed slot basis.

**Takeaway.** Longer contexts increase the hidden width $t$ and hence the size of the candidate pool. Learnable registers can therefore increase capacity even without semantic content, by providing additional trainable pool atoms/slots that can be activated as needed.

## H.4. Sequence mixing learns a sequence basis and changes the gating geometry

This implication formalizes two related facts: (i) in the static regime, $R^{(1)}$ is literally a change of basis in the hidden (sequence) dimension, and (ii) in the dynamic regime, input-dependent $R^{(1)}(x)$ warps the ReLU partition boundaries.

**Proposition H.7** (Static sequence mixing mixes hyperplane normals). *Fix a context $X$ and assume $L^{(1)}(x) \equiv L^{(1)}$ and $R^{(1)}(x) \equiv R^{(1)}$ are constant in $x$. Define $B_0(X) := L^{(1)} X^\top \in \mathbb{R}^{d \times t}$ and $B(X) := B_0(X) R^{(1)} \in \mathbb{R}^{d \times t}$. Then $h(x; X) = x B(X)$ and the ReLU gating boundaries are hyperplanes*

$$H_j = \{x : x\, b_j = 0\}, \qquad b_j := B(X)[:, j] = \sum_{i=1}^t R_{i,j}^{(1)}\, B_0(X)[:, i].$$

*In particular, any off-diagonal mixing in $R^{(1)}$ replaces each normal $b_j$ by a linear combination of the base normals $B_0(X)[:, i]$, changing the hyperplane arrangement and hence the partition geometry.*

**Proposition H.8** (Dynamic sequence mixing yields warped (curved) boundaries). *Fix $X$ and consider the fully dynamic case*

$$h(x; X) = x\, L^{(1)}(x)\, X^\top\, R^{(1)}(x).$$

*Assume $L^{(1)}(\cdot)$ and $R^{(1)}(\cdot)$ are $C^1$. Then the boundary set $\mathcal{B}_X = \bigcup_{j=1}^t \{x : h_j(x; X) = 0\}$ is (generically) a union of $C^1$ hypersurfaces, and on each connected component of $\mathbb{R}^d \setminus \mathcal{B}_X$ the sign pattern $\mathbf{1}\{h(x; X) > 0\}$ is constant, so $o(\cdot; X)$ is smooth on that component. Moreover, writing $z(x; X) := x L^{(1)}(x) X^\top$ so that $h = z R^{(1)}(x)$, the differential satisfies*

$$dh = (dx)\, L^{(1)}(x)\, X^\top\, R^{(1)}(x) +\, x\, (dL^{(1)}(x))\, X^\top\, R^{(1)}(x) +\, x\, L^{(1)}(x)\, X^\top\, (dR^{(1)}(x)),$$

*so input-dependent sequence mixing ($dR^{(1)}(x)$) is an explicit deformation term that bends/moves gating boundaries.*

*Remark* H.9 (Low-rank dynamic R gives controlled warping). In HyperMLP, $R^{(1)}(x) = D + A\, S(x)\, B^\top$ with diagonal $S(x)$. Then $dR^{(1)}(x) = A\, (dS(x))\, B^\top$ has rank at most $r_s$, so the deformation term $x L^{(1)} X^\top (dR^{(1)}(x))$ acts through a low-dimensional mechanism, suggesting a favorable expressivity–stability tradeoff: boundaries can warp without introducing arbitrary high-curvature geometry.

**Takeaway.** Learning $R^{(1)}$ is literally learning the basis and geometry of the ReLU gating space along the sequence dimension: static $R^{(1)}$ mixes hyperplane normals (basis change), while dynamic $R^{(1)}(x)$ warps the partition boundaries.

## H.5. Purely linearized attention lacks routing/selection

We formalize the intuition that removing the gating nonlinearity collapses the multi-stage pruning mechanism. Concretely, we consider a "linearized" variant that keeps only a sign-preserving normalization and removes ReLU/GLU gating.

**Proposition H.10** (Ungated normalization yields a full-pool signed readout). *Consider the same dynamic block as (77), but replace the activation by the ungated normalization*

$$\sigma_{\mathrm{lin}}(z) := \mathrm{L2Norm}_t(z) = \frac{z}{\rho_t(z)} \qquad (\textit{no ReLU/GLU gating}),$$

*where $\rho_t(z) > 0$ is a scalar (e.g. $\rho_t(z) = \sqrt{\|z\|_2^2 + \varepsilon}$). Then the output admits the measure form*

$$o_{\mathrm{lin}}(x; X) = \frac{1}{\rho_t(h(x; X))} \int_\Omega \alpha(x; u)\, \beta(x; v)\, d\mu_{X,x}^{\mathrm{pool}}(u, v), \tag{80}$$

*i.e., the readout is taken from the entire Context-instantiated Memory Pool with (generally) signed coefficients $\alpha(x; u)$, and there is no restriction to an activated subpool.*

*Equivalently, the "pool $\to$ activated" stage becomes trivial: if we define a trivial gate $g_{\mathrm{full}} \equiv 1$ and the corresponding "active" measure*

$$\mu_{X,x}^{\mathrm{full}} := g_{\mathrm{full}}\, \mu_{X,x}^{\mathrm{pool}} = \mu_{X,x}^{\mathrm{pool}},$$

*then*

$$\operatorname{supp}\big(\mu_{X,x}^{\text{full}}\big) = \operatorname{supp}\big(\mu_{X,x}^{\text{pool}}\big) \qquad \textit{for all } (X,x).$$

*In particular, the discrete, input-conditioned sub-network selection induced by a ReLU mask is absent in this variant.*

*Proof.* With $\sigma_{\text{lin}}(h) = h/\rho_t(h)$ we have

$$o_{\text{lin}}(x; X) = \frac{1}{\rho_t(h(x; X))} \sum_{i=1}^{t} h_i(x; X)\, \beta(x; v_i).$$

By construction of the pool slots, $h_i(x; X) = \alpha(x; u_i)$, hence

$$o_{\text{lin}}(x; X) = \frac{1}{\rho_t(h(x; X))} \sum_{i=1}^{t} \alpha(x; u_i)\, \beta(x; v_i) = \frac{1}{\rho_t(h(x; X))} \int_{\Omega} \alpha(x; u)\, \beta(x; v)\, d\mu_{X,x}^{\text{pool}}(u, v),$$

which is (80). Since there is no sign-dependent masking, there is no restriction step $\mu^{\text{pool}} \to \mu^{\text{act}}$; equivalently one may view the "active" measure as the trivially gated $\mu^{\text{full}} = \mu^{\text{pool}}$. $\qquad\square$

*Remark* H.11 (Active-set partition collapses (but the map need not be linear)). Under ReLU/GLU gating, the sign pattern $\mathbf{1}\{h(x; X) > 0\}$ induces a nontrivial partition of input space into regions with different active sets (polyhedral in the static case, warped in the dynamic case), yielding combinatorial, input-conditioned routing over pool slots. Under $\sigma_{\text{lin}}$, all coordinates contribute for all inputs (no coordinates are pruned by a mask), so the partition induced by changes in the active set becomes trivial: there is a single active set equal to the full pool. Note that $x \mapsto o_{\text{lin}}(x; X)$ may still be nonlinear due to the input dependence of $L^{(j)}(x), R^{(j)}(x)$ and the normalization scalar $\rho_t(h(x; X))$; the point here is the loss of combinatorial gating.

**Takeaway.** Replacing ReLU/GLU gating by ungated normalization removes the pool $\to$ activated subpool restriction and collapses routing to a single signed readout over the full context-instantiated pool, reducing the model's ability to perform discrete, input-conditioned sub-network selection over instantiated slots.

### H.6. LoRA: spend adaptation rank on the readout side (V/O)

**Dynamic-MLP viewpoint.** In our notation, a single attention head at token $t$ can be written as a residual depth-two map

$$o_t = x_t + \sigma\big(x_t W^{(1)}(X, x_t)\big)\, W^{(2)}(X, x_t), \tag{81}$$

where the first layer $W^{(1)}$ (implemented by the $QK$ pathway) determines routing/selection, and the second layer $W^{(2)}$ (implemented by the $VO$ pathway) determines the action/readout applied to the residual stream.

**Why low-rank adapters are naturally effective on $V/O$.** LoRA injects trainable low-rank updates into a frozen weight matrix, $W \leftarrow W + \Delta W$, with $\operatorname{rank}(\Delta W) \leq r$ (Hu et al., 2021). If we allocate LoRA capacity to the readout side, i.e., $W^{(2)} \leftarrow W^{(2)} + \Delta W^{(2)}$, then the induced change in the head output is

$$\Delta o_t = \sigma\big(x_t W^{(1)}(X, x_t)\big)\, \Delta W^{(2)}(X, x_t). \tag{82}$$

For any fixed $(X, x_t)$, this implies $\Delta o_t$ always lies in the row space of $\Delta W^{(2)}$, whose dimension is at most $r$. Equivalently, under a rank budget $r$, adapting $V/O$ directly controls how many new update directions the head can add to the residual stream (a direct corollary of Theorem F.3 applied to the update term).

**Why $Q/K$ adapters can be less parameter-efficient under the same rank budget.** If we instead place LoRA on the routing side (e.g., $Q/K$, which changes $W^{(1)}$), the primary effect is to perturb the coefficient map $\sigma(\cdot)$ in Equation (81). Our Theorem F.5 formalizes that low-rank constraints on the first layer induce a quotient structure: only components of $x_t$ that fall inside the row space of the (low-rank) perturbation can influence routing. Intuitively, with a small rank budget, it can be harder to reshape routing boundaries across diverse contexts than it is to adjust the readout/action vectors.

**Empirical signals consistent with the readout-side priority.** Multiple empirical probes suggest that $V/O$ are information-dense and more involved in adaptation: (i) low-rank fine-tuning analysis reports that $W_v$ and the output dense projection change more than $W_q/W_k$ during adaptation (Radiya-Dixit & Wang, 2020); (ii) rank-profiling for LLM compression finds the effective rank distribution is unbalanced, with $V$ having substantially higher effective rank than $Q/K$ (Mi et al., 2025). At the same time, extreme quantization results highlight that if the routing path is overly constrained, accuracy can degrade sharply, indicating that $Q/K$ can become a bottleneck in aggressive regimes (Bai et al., 2021). Taken together, these observations align with the dynamic-MLP view: under a limited adaptation budget, prioritizing $V/O$ often yields a strong return, while $Q/K$ should not be compressed or adapted too aggressively.

### H.7. Gated attention: gates are most expressive on the readout side (between $V$ and $O$)

We write a one-head attention-style block (no explicit sequence mixing) in our dynamic-MLP notation as

$$h_t := x_t L^{(1)} X^\top \in \mathbb{R}^{1 \times t}, \qquad o_t := \sigma(h_t) X L^{(2)\top} \in \mathbb{R}^{1 \times d}, \tag{83}$$

where $X := X_{t:1} \in \mathbb{R}^{t \times d}$ is the lag-ordered prefix and, for standard attention factors, $L^{(1)} = W_q W_k^\top$ and $L^{(2)\top} = W_v W_o^\top$ (per head).

A gate placed "between $V$ and $O$" corresponds to making the readout-side feature mixing operator input-conditioned:

$$L^{(2)\top} \rightsquigarrow L^{(2)\top}(x_t) := W_v G(x_t) W_o^\top, \qquad G(x_t) = \mathrm{Diag}(g(x_t)) \in \mathbb{R}^{d_{vo} \times d_{vo}}, \tag{84}$$

so the head becomes

$$o_t^{\mathrm{gate}} = \sigma(h_t) X L^{(2)\top}(x_t). \tag{85}$$

In the measure formulation (Appendix Theorem C.5), this gate changes only the readout map $\beta(x_t; v) = v L^{(2)\top}(x_t)$ while leaving the addressing map $\alpha(x_t; u) = x_t L^{(1)} u^\top$ (and thus the ReLU active-set geometry induced by $h_t$) unchanged:

$$o_t^{\mathrm{gate}} = \frac{1}{\rho_t(h_t)} \sum_{i=1}^{t} \alpha(x_t; u_i)_+ \underbrace{\left( v_i L^{(2)\top}(x_t) \right)}_{\beta(x_t; v_i) \text{ is gated}}, \tag{86}$$

with $(u_i, v_i)$ instantiated from the context (and from $R^{(1)}, R^{(2)}$ if present). Thus a $V/O$-side gate is an input-conditioned modulation of the second-layer/readout weights in the dynamic-MLP view: it changes what selected slots do in the residual stream without requiring changes to the routing boundaries (the margin-based gate-stability Lemma D.11).

This interpretation matches empirical findings in gated-attention designs: the most effective placement is to gate on the $V \to O$ side, and making the gate depend on the current token $x_t$ is particularly beneficial (Qiu et al., 2025). Finally, note that our HyperMLP parameterization already subsumes this mechanism: the dynamic diagonal core $M^{(2)}(x_t)$ inside $L^{(2)\top}(x_t) = W_v M^{(2)}(x_t) W_o^\top$ is exactly a $V/O$-side gate, while HyperMLP further adds learnable sequence-space mixing via $R^{(2)}(x_t)$.

## I. Additional Function Classes of HyperMLP and Task-Level Implications

This appendix section summarizes (i) which function classes HyperMLP can represent beyond token-wise ReLU attention, (ii) concrete examples of such functions, and (iii) what these extra classes suggest about performance on six canonical context-based tasks.

### I.1. From token-wise polyhedral routing to context-wide warped routing

We fix a context length $t$ and use lag order $X := X_{t:1} \in \mathbb{R}^{t \times d}$. Throughout, $\mathrm{L2Norm}_t$ denotes an affine-free positive scalar rescaling across the length-$t$ vector, i.e., $\mathrm{L2Norm}_t(z) = z/\rho_t(z)$ for some $\rho_t(z) > 0$, so it preserves sign patterns.

**Token-wise ReLU attention (polyhedral class).** A minimal ReLU-attention-style head with no sequence mixing can be written as

$$o_{\mathrm{att}}(x; X) := \sigma_t\!\left( x L X^\top \right) X \widetilde{L}^\top, \qquad \sigma_t(z) := \mathrm{ReLU}(\mathrm{L2Norm}_t(z)), \tag{87}$$

where $L, \widetilde{L} \in \mathbb{R}^{d \times d}$ are constant in $x$. For fixed $X$, the pre-activation is $h_{\text{att}}(x; X) = xB(X)$ with $B(X) := LX^\top \in \mathbb{R}^{d \times t}$, hence the ReLU boundary set is a finite hyperplane arrangement

$$\mathcal{B}_X^{\text{att}} := \bigcup_{j=1}^{t} \{x : \ (h_{\text{att}})_j(x; X) = 0\} = \bigcup_{j=1}^{t} \{x : \ x\, b_j = 0\}. \tag{88}$$

Equivalently, token-wise ReLU attention induces a polyhedral (CPWL) partition of the $x$-space for each fixed $X$.

**HyperMLP (warped spline class).** A one-head HyperMLP block computes

$$h(x; X) := x\, L^{(1)}(x)\, X^\top R^{(1)}(x) \in \mathbb{R}^{1 \times t}, \qquad o_{\text{hyp}}(x; X) := \sigma_t\big(h(x; X)\big)\, R^{(2)\top}(x)\, X\, L^{(2)\top}(x), \tag{89}$$

where $L^{(1)}(\cdot), L^{(2)}(\cdot)$ are feature-space mixings and $R^{(1)}(\cdot), R^{(2)}(\cdot)$ are sequence-space mixings (diagonal-plus-low-rank in our parameterization). For fixed $X$, the boundaries are the level sets $\{x : \ h_j(x; X) = 0\}$, which are generically curved and define a warped orthant partition (Appendix Theorem C.13).

**Two sources of extra function classes.** The strict generalization can be decomposed into two structural mechanisms:

1. **(S) Slot construction / learned sequence basis.** HyperMLP instantiates pool slots as context-wide linear combinations

$$u_i(x; X) = r_i^{(1)}(x)^\top X, \qquad v_i(x; X) = r_i^{(2)}(x)^\top X, \tag{90}$$

   instead of tying each slot to a single token $X[i, :]$. This enables direct computation with latent (span-/entity-level) memory atoms such as $c^\top X$.

2. **(G) Warped routing / curved gating geometry.** Input-dependent mixing (especially $R^{(1)}(x)$) turns polyhedral gating boundaries into smooth, deformable level sets. Thus HyperMLP can implement acceptance/routing regions that are naturally curved in representation space.

**Proposition I.1** (Strict expressivity gap (summary)). *For fixed $X$, token-wise ReLU attention induces polyhedral (hyperplane-arrangement) gating geometry in $x$ as in (88). HyperMLP contains this class and can additionally realize warped (piecewise-smooth) spline maps whose boundary sets contain curved hypersurfaces not representable by any finite union of hyperplanes.*

**Proposition I.2** (Paying for temporal mixing by shrinking QK yields a strict budgeted expressivity gain). *Fix $(d, t)$ and $\sigma_t(z) = \text{ReLU}(\text{L2Norm}_t(z))$, where $\text{L2Norm}_t(z) = z/\rho_t(z)$ with $\rho_t(z) > 0$ (sign patterns are preserved).*

**Baseline (token-wise ReLU attention, no sequence mixing).** *For one head, consider*

$$o_{\text{att}}(x; X) = \sigma_t\big(x\, W_q W_k^\top X^\top\big)\, X\, W_v W_o^\top, \qquad W_q, W_k \in \mathbb{R}^{d \times d_{qk}}, \ \ W_v, W_o \in \mathbb{R}^{d \times d_{vo}}. \tag{91}$$

*We count per-head parameters as*

$$P_{\text{att}}(d_{qk}, d_{vo}) := 2d\, d_{qk} + 2d\, d_{vo}. \tag{92}$$

**HyperMLP (minimal form: same feature factors, plus temporal mixing).** *Consider the restriction of HyperMLP in which feature mixing is still low-rank but static (we set the diagonal cores $M^{(1)}(x) \equiv I$ and $M^{(2)}(x) \equiv I$ for the budget argument), while sequence mixing is learned:*

$$o_{\text{hyp}}(x; X) = \sigma_t\big(x\, \widetilde{W}_q \widetilde{W}_k^\top X^\top R^{(1)}(x)\big)\, R^{(2)\top}(x)\, X\, \widetilde{W}_v \widetilde{W}_o^\top, \tag{93}$$

*with $\widetilde{W}_q, \widetilde{W}_k \in \mathbb{R}^{d \times \widetilde{d}_{qk}}$ and $\widetilde{W}_v, \widetilde{W}_o \in \mathbb{R}^{d \times d_{vo}}$ (we keep $d_{vo}$ unchanged). Each $R^{(j)}(x) \in \mathbb{R}^{t \times t}$ is diagonal-plus-low-rank:*

$$R^{(j)}(x) = D^{(j)} + A^{(j)} S^{(j)}(x) B^{(j)\top}, \qquad D^{(j)} = I + \text{Diag}(p^{(j)}), \quad S^{(j)}(x) = \text{Diag}(\phi(x W_S^{(j)})), \tag{94}$$

*where $A^{(j)}, B^{(j)} \in \mathbb{R}^{t \times r_s}$, $p^{(j)} \in \mathbb{R}^t$, and $W_S^{(j)} \in \mathbb{R}^{d \times r_s}$. A crude per-head upper bound on the added temporal parameters (for both $j \in \{1, 2\}$) is*

$$P_{\text{temp}}(t, r_s) := 4t\, r_s + 2t + 2d\, r_s. \tag{95}$$

*Assume we shrink QK so that the saved parameters pay for temporal mixing:*

$$2d\,(d_{qk} - \widetilde{d}_{qk}) \;\geq\; P_{\text{temp}}(t, r_s). \tag{96}$$

*Then:*

1. **(Matched-budget feasibility).** *There exists a HyperMLP head* (93) *with ranks* $(\widetilde{d}_{qk}, d_{vo})$ *and temporal rank* $r_s$ *such that*

$$P_{\text{att}}(d_{qk}, d_{vo}) \;\geq\; P_{\text{att}}(\widetilde{d}_{qk}, d_{vo}) + P_{\text{temp}}(t, r_s),$$

   *i.e., it fits in the same per-head parameter budget as the baseline* (91) *while preserving* $d_{vo}$.

2. **(Strictly richer function class at matched budget: curved routing).** *If* $\widetilde{d}_{qk} \geq 2$ *and* $r_s \geq 1$, *then for some* $X$ *there exists a choice of HyperMLP parameters satisfying* (96) *such that the map* $x \mapsto o_{\text{hyp}}(x; X)$ *cannot be realized by any token-wise ReLU-attention head with static (input-independent) feature projections, i.e., any head for which, for fixed* $X$, *the score/pre-activation vector is affine in* $x$. *In particular, it cannot be realized by any token-wise head of the form* (91) *with fixed (input-independent) projection matrices* $W_q, W_k, W_v, W_o$ *(for any choice of ranks).*

*Moreover, shrinking* $d_{vo}$ *instead of* $d_{qk}$ *imposes a hard output-subspace restriction: for any head (either* (91) *or* (93)*) the output lies in the fixed subspace* $\text{col}(W_o)$ *(or* $\text{col}(\widetilde{W}_o)$*), so in a residual block* $f(x; X) = x + o(x; X)$ *every direction* $u \perp \text{col}(W_o)$ *is invariant (*$\langle f(x; X) - x, u \rangle \equiv 0$*). Hence, under a fixed budget, it is more expressivity-preserving to keep* $d_{vo}$ *large and pay for temporal mixing by shrinking* $d_{qk}$.

*Proof.* **Budget.** The baseline uses $2dd_{qk}$ QK parameters; reducing to $\widetilde{d}_{qk}$ saves $2d(d_{qk} - \widetilde{d}_{qk})$. Under (96), this covers the temporal parameters (95), yielding the first claim.

**Strict separation.** We make explicit the baseline class used in the separation argument. Call a head token-wise ReLU-attention with static feature projections if, for every fixed context $X$, its score vector is affine in $x$:

$$h_{\text{att}}(x; X) = x\, B(X), \qquad B(X) \in \mathbb{R}^{d \times t} \text{ independent of } x,$$

and the output is of the form $o_{\text{att}}(x; X) = \sigma_t(h_{\text{att}}(x; X))\, C(X)$ with $C(X)$ independent of $x$. This includes the standard token-wise head (91) (and any variant that only inserts fixed linear transforms inside the bilinear form), where $B(X) = W_q W_k^\top X^\top$ and $C(X) = X W_v W_o^\top$. For this baseline class, each coordinate of $h_{\text{att}}(x; X)$ is linear in $x$. Since $\text{L2Norm}_t$ is a positive scalar rescaling, the gate pattern can change only when some coordinate is 0; thus the nondifferentiability set of $x \mapsto o_{\text{att}}(x; X)$ is contained in a finite union of affine hyperplanes $\bigcup_{j=1}^t \{x :\ (xB(X))_j = 0\}$ (Appendix Theorem C.10).

We now exhibit a HyperMLP instance with a curved nondifferentiability set. Take $t = 2$ and any $d \geq 2$. Let $X = [e_1^\top; e_2^\top] \in \mathbb{R}^{2 \times d}$. Choose $\widetilde{d}_{qk} = 2$ with $\widetilde{W}_q = \widetilde{W}_k = [e_1, e_2] \in \mathbb{R}^{d \times 2}$, so $x\widetilde{W}_q \widetilde{W}_k^\top X^\top = (x_1, x_2)$. Set $R^{(2)}(x) \equiv I_2$. Let $r_s = 1$ and define

$$R^{(1)}(x) = I_2 + e_1\, \phi(cx_1)\, e_2^\top = \begin{bmatrix} 1 & \phi(cx_1) \\ 0 & 1 \end{bmatrix},$$

which is of the form (94) with $A = e_1$, $B = e_2$, and $W_S^{(1)} = ce_1$. Then

$$h(x; X) = (x_1, x_2)\, R^{(1)}(x) = (x_1,\ x_2 + \phi(cx_1)\, x_1).$$

Finally choose $\widetilde{W}_v = \widetilde{W}_o = e_2 \in \mathbb{R}^{d \times 1}$ so that $X\widetilde{W}_v \widetilde{W}_o^\top$ has only the second row nonzero and equals $e_2^\top$. Therefore the head output is

$$o_{\text{hyp}}(x; X) = \frac{1}{\rho_2(h(x; X))} \text{ReLU}\big(x_2 + \phi(cx_1)x_1\big)\, e_2^\top.$$

Because $\partial_{x_2}\big(x_2 + \phi(cx_1)x_1\big) \equiv 1$, this function is nondifferentiable along the entire smooth curve $\{x :\ x_2 + \phi(cx_1)x_1 = 0\}$, which is not contained in any finite union of affine hyperplanes. Hence it cannot equal any head in the above token-wise static-projection (polyhedral) class: in particular, it cannot equal any token-wise head of the form (91) with fixed projections, whose nondifferentiability set is always contained in a finite hyperplane union.

**Why not shrink VO.** For either (91) or (93), the output has the form $o(x; X) = z(x; X)\, W_o^\top$ (or $z\, \widetilde{W}_o^\top$), so $o(x; X) \in \text{col}(W_o)$ for all $(x, X)$. Thus in $f(x; X) = x + o(x; X)$ every $u \perp \text{col}(W_o)$ is invariant, and reducing $d_{vo}$ shrinks this update subspace. $\qquad\square$

| Task | Expected gain | Mechanism | Structural reason |
|---|---|---|---|
| Compression | high | (S) | needs low-rank summaries $c^\top X$ |
| Noisy in-context recall | high | (S)+margin | suppress spurious active slots; improve SNR |
| Fuzzy in-context recall | high | (G) | curved acceptance/routing regions in $x$ |
| Selective copying | med–high | (S) | learn sequence operators (shift/offset) in $R^{(2)}$ |
| Clean in-context recall | low | – | token-wise basis already aligned |
| Memorizing training data | low | – | mostly parametric memory (not context readout) |

*Table 4.* Qualitative predictions under matched parameter budgets, based on the expressivity mechanisms (S) and (G).

## I.2. Examples of additional functions realized by HyperMLP

We list three concrete families that lie outside the polyhedral/token-wise ReLU-attention class (for fixed $X$) but are realizable by HyperMLP via (S) and/or (G).

**Example 1: sigmoid-warped halfspaces (curved boundary in $\mathbb{R}^2$).** Let $d = t = 2$, $X = I_2 = [e_1^\top; e_2^\top]$, $L^{(1)}(x) \equiv I$, and $R^{(2)}(x) \equiv I$, $L^{(2)}(x) \equiv I$. Choose a rank-1 dynamic sequence mixing

$$R^{(1)}(x) = I_2 + e_1\, \phi(cx_1)\, e_2^\top, \qquad \phi = \text{Sigmoid},\ c > 0,$$

so $h(x; X) = xR^{(1)}(x) = (x_1,\ x_2 + \phi(cx_1)x_1)$. Then the function

$$f(x) := \frac{1}{\rho_t(h(x;X))}\, \text{ReLU}\big(x_2 + \phi(cx_1)x_1\big)$$

has a curved boundary $\{x:\ x_2 = -\phi(cx_1)x_1\}$ and therefore cannot arise from any finite hyperplane arrangement.

**Example 2: moving-normal halfspaces (query-dependent address direction).** Fix $d \geq 2$ and $t = 2$ with context rows $X[1,:] = a^\top$ and $X[2,:] = b^\top$ (linearly independent). Construct a slot

$$u_2(x; X) = b + \phi(w^\top x)\, a$$

using $R^{(1)}(x) = I_2 + e_1\, \phi(w^\top x)\, e_2^\top$ (so that $u_2$ is a context-wide mixture). Then $\alpha(x; u_2) = xb^\top + \phi(w^\top x)\, xa^\top$, and the gating boundary $\{x:\ \alpha(x; u_2) = 0\}$ generically has nonconstant normal direction (hence is not polyhedral).

**Example 3: multi-knot warped splines via $r_s > 1$.** With $R^{(1)}(x) = D + A\, S(x)\, B^\top$ and $S(x) = \text{Diag}(\phi(xW_S))$, one can realize pre-activations containing sums of sigmoid-modulated linear forms, e.g.

$$g(x; X) = x\, b^\top + \sum_{k=1}^{r_s} \phi(w_k^\top x)\, x\, a_k^\top,$$

yielding warped boundaries $\{x:\ g(x; X) = 0\}$ with multiple bends/inflections as $r_s$ grows. This induces a spline geometry qualitatively different from a hyperplane arrangement.

## I.3. Implications for canonical context tasks

We now connect the extra function classes to six tasks frequently used to probe "in-context memory". The guiding principle is an approximation argument: for a task distribution $\mathcal{D}$ over $(x, X, y)$, a strict function-class superset can achieve strictly smaller population risk whenever the Bayes predictor is closer to the larger class. Here, HyperMLP's advantage is most pronounced when the target requires either latent slot construction (S) or warped routing (G).

### I.3.1. NOISY IN-CONTEXT RECALL: SPURIOUS ACTIVATION SCALES WITH CONTEXT LENGTH

A minimal noisy-recall model is: among $t$ slots, only a small subset $S$ contains useful signal and the rest are noise. In token-wise ReLU attention, each noise token is an independent slot and can be (spuriously) activated.

**Proposition I.3** (Token-wise spurious activation under symmetric noise)**.** *Consider token-wise ReLU attention* (87) *and fix* $(x, X)$*. Let* $N \subset [t]$ *be indices of "noise" tokens such that* $\alpha(x; X[i, :])$ *is symmetric about* 0 *for each* $i \in N$ *(e.g.* $x^\top X[i, :] \sim \mathcal{N}(0, \sigma^2)$ *conditionally on* $x$*). Then the expected number of activated noise slots satisfies*

$$\mathbb{E}\Big[\#\{i \in N : \alpha(x; X[i, :]) > 0\}\Big] = \frac{|N|}{2}.$$

*Equivalently, the activated-pool TV mass contributed by noise grows linearly in* $|N|$*:*

$$\mathbb{E}\big[\|\mu_{X,x}^{\mathrm{act}}\|_{\mathrm{TV}} \text{ from noise}\big] = |N|/2.$$

*Proof.* For symmetric $\alpha$, $\mathbb{P}(\alpha > 0) = 1/2$ and linearity of expectation gives the claim. The TV identity follows from the atomic-measure interpretation (Appendix Remark E.7). $\square$

**Why HyperMLP helps.**   HyperMLP can reduce spurious activations by changing the pool before gating: sequence mixing constructs context-wide slots (S) so that many noisy tokens can be aggregated into a few "dummy" slots with small variance, while signal is concentrated into a small number of high-margin slots. Concretely, averaging $m$ independent noise tokens reduces the address variance by $1/m$, increasing the gate margin. Under a margin, the activated set is stable (Appendix Lemma D.11), so noisy recall benefits directly from (S) via improved SNR and fewer spurious activated atoms.

I.3.2. COMPRESSION: TARGETS THAT DEPEND ON LOW-RANK CONTEXT SUMMARIES

Many compression-style objectives can be idealized as depending on a low-dimensional statistic $s(X) \in \mathbb{R}^k$ with $k \ll t$, e.g. $s(X) = C^\top X$ for some $C \in \mathbb{R}^{t \times k}$.

**Why HyperMLP helps.**   HyperMLP can implement such summaries directly through $R^{(2)\top}(x)X$: choosing $R^{(2)}$ so that a small set of rows of $V = R^{(2)\top}(x)X$ equals $C^\top X$ yields $k$ "summary slots" that can then be selectively gated/read out. Token-wise ReLU attention, by contrast, instantiates one slot per token; it can represent low-rank summaries only indirectly (by distributing the computation across many heads/layers). Under matched parameter budgets, the ability to instantiate and route through a small number of summary atoms corresponds to a smaller approximation error for compression objectives.

I.3.3. FUZZY IN-CONTEXT RECALL: CURVED ACCEPTANCE REGIONS AND WARPED ROUTING

Fuzzy recall often means the model should retrieve when the query lies in a neighborhood of a key or a cluster, rather than requiring exact matching. A stylized form is thresholded similarity:

$$\text{retrieve key } k \text{ if } g(x, k) \geq 0,$$

where the boundary $\{x : g(x, k) = 0\}$ is typically curved (e.g. spheres/ellipsoids for distance thresholds).

**Why HyperMLP helps.**   For fixed $X$, token-wise ReLU attention yields polyhedral routing boundaries (88). Approximating curved boundaries by a hyperplane arrangement generally requires many pieces (many gates/regions), whereas HyperMLP can realize curved boundaries directly via input-dependent mixing (G) (Appendix Theorem C.13 and Examples 1–3). Thus fuzzy recall is a canonical setting where "warped orthant partitions" are a better inductive match than polyhedral ones.

I.3.4. SELECTIVE COPYING: SEQUENCE OPERATORS INSIDE THE READOUT PATH

Selective copying requires producing (parts of) the context, but with structured selection rules. A particularly revealing subcase is copy-with-offset: select a position $i^\star$ but output the token at $i^\star + \Delta$ (in lag coordinates, a fixed shift).

**Why HyperMLP helps.**   HyperMLP can implement shift/offset behavior directly by choosing $R^{(2)\top}$ to be a (possibly learned) shift operator $J_\Delta$:

$$V = R^{(2)\top}(x)X \equiv J_\Delta X \quad \Rightarrow \quad \beta(x; v_i) \text{ reads from } X[i + \Delta, :].$$

The gate can still be driven by $u_i$ (possibly different from $v_i$), so one head can implement "match here, copy there". This is awkward for token-wise attention because it ties each gating coordinate and readout coordinate to the same position basis; emulating an offset typically requires additional heads/layers or explicit positional feature engineering. Hence selective copying benefits when the selection rule involves nontrivial sequence transforms.

### I.3.5. CLEAN IN-CONTEXT RECALL: TOKEN-WISE ROUTING ALREADY MATCHES THE TASK

In clean/exact in-context recall, the correct memory item is typically a single token/span that should dominate the gate (e.g. a unique match with large positive margin). This aligns well with the token-wise basis of ReLU attention: a single coordinate can activate and directly read out the corresponding value. Therefore HyperMLP's extra capacity (S,G) is not required for representation and may yield only modest gains in this regime, absent additional confounders (e.g. span aggregation, noise, or fuzzy matching).

### I.3.6. MEMORIZING TRAINING DATA: PRIMARILY PARAMETRIC, NOT CONTEXT-READOUT LIMITED

Memorizing training data is often dominated by parametric memory (information stored in $\theta$) rather than the context-read channel $\Omega \to \mu^{\text{pool}} \to \mu^{\text{act}}$. While HyperMLP changes the hypothesis class, the expressivity gap analyzed here is most directly about how context is instantiated and routed. Thus, without additional assumptions on optimization or inductive bias, the present analysis does not predict a strong and universal advantage for parametric memorization itself; any differences are more likely to arise indirectly (e.g. through regularization, optimization dynamics, or interactions with depth).

**Takeaway.** The math suggests the largest gains should appear when the task requires either (i) constructing a small number of meaningful context-wide slots from many tokens (compression, noisy recall), or (ii) learning curved acceptance/routing regions in query space (fuzzy recall). Selective copying can also benefit when it requires explicit sequence operators (shifts/offsets) in the readout path. In contrast, clean recall is already well matched by token-wise routing, and parametric memorization is not primarily constrained by the context-read mechanism studied here.

## J. Efficient Implementation

HyperMLP is designed to be an expressive *superset* of quadratic attention while preserving efficient autoregressive inference and avoiding any explicit materialization of dense $t \times t$ sequence-mixing matrices. Our implementation follows three principles: (i) **lag-ordered, length-agnostic parameter tensors** so that extending context corresponds to taking longer prefix-slices without reindexing; (ii) **offset-aligned (skewed) layouts** to make causal prefixes contiguous in memory when processing row-blocks; and (iii) **DPLR mixing via two low-rank contractions** with fused epilogues to minimize memory traffic.

### J.1. Length-agnostic DPLR parameters in lag order

Recall the DPLR (diagonal-plus-low-rank) parameterization used for temporal mixing:

$$R_t^{(j)}(x) = D_{1:t}^{(j)} + A_{1:t}^{(j)} \operatorname{Diag}(s^{(j)}(x)) B_{1:t}^{(j)\top}, \qquad s^{(j)}(x) \in \mathbb{R}^{r_s}.$$

To support variable context lengths without reparameterizing across $t$, we store *maximum-length* tensors $A^{(j)}, B^{(j)} \in \mathbb{R}^{L \times r_s}$ and diagonal parameters $d^{(j)} \in \mathbb{R}^L$ (for all heads, in per-head form), and at runtime we slice correspondingly to the current length $T$:

$$s = L - T, \quad A_{T:1} \leftarrow A[s{:}], \quad B_{T:1} \leftarrow B[s{:}], \quad d_{T:1} \leftarrow d[s{:}].$$

This "slicing in lag order" implements the canonical prefix-extension rule and directly matches the extension-consistency construction (Appendix Lemma C.6). As a result, long-context training and evaluation do not require any special handling beyond slicing.

### J.2. Offset-aligned (skew) layout for row-blocked causal computation

In full-sequence training (teacher forcing), we process the score-like matrices in *row blocks* of size $M$ (similar in spirit to FlashAttention-style tiling) to control peak memory. Let $X_{\text{left}} \in \mathbb{R}^{B \times H \times M \times T}$ denote a block of $M$ query rows against $T$ key columns in the *left layout*, where the causal mask corresponds to a lower-triangular region in the (row, col) plane. Many operations we need (temporal DPLR right-mixing, normalization across the length-$T$ axis, and elementwise gating) are most efficient when each row's *valid causal prefix* is contiguous.

We therefore use an *offset-aligned (skew) layout* transform: for a block starting at global row index row_start (0-based),

define $r = \texttt{row\_start} + m$ and

$$X_{\mathrm{off}}[m, \tau] = \begin{cases} X_{\mathrm{left}}[m, c], & c = \tau + r - (T-1) \geq 0, \\ 0, & \text{otherwise.} \end{cases} \tag{97}$$

In this offset layout, the diagonal element $c = r$ is aligned to $\tau = T - 1$ for *all* rows, and each row's causal prefix becomes a right-aligned contiguous segment (length $r + 1$), with left padding filled by zeros. This makes causal masking a simple predicate on $(m, \tau)$ that can be applied (or re-applied) cheaply after mixing.

### J.3. DPLR right-mixing without materializing $R$

A key efficiency point is that we never form $R_t^{(j)}(x) \in \mathbb{R}^{t \times t}$ explicitly. Instead, we use the standard vector–DPLR identity (Appendix Lemma G.1): for any row-batched tensor $Y \in \mathbb{R}^{B \times H \times M \times T}$ in offset layout and $R = D + A \operatorname{Diag}(s) B^\top$,

$$YR = Y \odot d + \big((YA) \odot s^\top\big) B^\top,$$

where $A \in \mathbb{R}^{T \times r_s}$, $B^\top \in \mathbb{R}^{r_s \times T}$, $d \in \mathbb{R}^T$, and $s \in \mathbb{R}^{r_s}$ (typically broadcastable over $(B, H, M)$). This reduces mixing to two low-rank contractions along the sequence axis:

$$H := YA \in \mathbb{R}^{B \times H \times M \times r_s}, \qquad Y_{\mathrm{lr}} := (H \odot s^\top)B^\top \in \mathbb{R}^{B \times H \times M \times T},$$

followed by an elementwise diagonal term $Y \odot d$ (and, when desired, a shortcut/residual addition). The arithmetic cost per block is $O(BHMT\,r_s)$ and the working-set overhead is $O(BHM\,r_s)$, matching the analysis in Appendix G.

### J.4. Kernel fusion: packing, epilogues, and masked residual mixing

The dominant practical bottleneck for DPLR mixing is *memory bandwidth*, not FLOPs. To reduce memory traffic, we fuse "cheap" post-ops into a single epilogue kernel. Concretely, after computing the low-rank expansion $Y_{\mathrm{lr}}$, we apply: (i) an optional shortcut ($Y_{\mathrm{lr}} \leftarrow Y_{\mathrm{lr}} + Y$), (ii) the diagonal term ($Y_{\mathrm{lr}} \leftarrow Y_{\mathrm{lr}} + Y \odot d$), and (iii) an optional causal mask in offset layout (zero out the left-padded region), all in one Triton kernel (`lr_epilogue_fwd_kernel`). This avoids extra reads/writes of intermediate tensors and keeps the hot path bandwidth-efficient.

We wrap the overall operation in a custom autograd Function (`FusedLowRankResidualMixFn`) so the backward can reuse the same structure (and, for the layout transform, the backward is simply the inverse transform). When Triton is unavailable, or when the input dtype is not supported, we fall back to a reference PyTorch implementation, ensuring portability without changing model semantics.

**Autoregressive inference.** At inference step $t$, HyperMLP uses the same caching philosophy as attention: the only step-dependent quantities needed for the DPLR mixes are the gate vectors $s_t^{(1)}, s_t^{(2)} \in \mathbb{R}^{r_s}$, so the extra per-head state is $O(r_s)$, and the per-step additional compute is $O(t\,r_s)$ (Appendix Proposition G.3). Thus HyperMLP preserves the same asymptotic inference scaling as quadratic attention, up to a low-order $r_s$ factor.

**Summary.** Overall, the implementation realizes the intended design goals: (i) no dense $t \times t$ sequence-mixing matrices are instantiated; (ii) DPLR mixing is implemented via two low-rank contractions plus fused epilogues; and (iii) offset-aligned layouts make causal prefixes contiguous and masking cheap when processing row blocks.

## K. Additional Experimental Results

**Computational Cost.** As shown in Table 5, we evaluate the training and inference speed of HyperGLU on an NVIDIA L40S GPU. To minimize confounding factors, we adopt a GPT-small configuration with 12 attention heads and a hidden dimension of 768. All experiments are conducted on randomly generated data with a context length of 1K and a batch size of 8. Specifically, we replace the attention module in a GPT-2–style Transformer (hidden size 768, 12 heads, 12 layers, with the number of heads of HyperGLU set to 2) with different attention variants and evaluate all models at a sequence length of 1024. We compare against RWKV7 (Peng et al., 2025), HGRN2 (Qin et al., 2024), a naive PyTorch implementation of softmax attention, FlashAttention (Dao et al., 2022), Gated Linear Attention (Yang et al., 2024b), Mamba (Gu & Dao, 2023), Gated Slot Attention (Zhang et al., 2024), and PaTH attention (Yang et al., 2025).

*Table 5.* Overall efficiency comparison (SeqLen=1024, Dim=768, Batch=8). FLOPs are reported in T ($10^{12}$) and FLOPs per token in M ($10^6$), where lower is better. Latency is measured in seconds (lower is better), throughput in K tokens/s (higher is better), and peak memory in GB (lower is better). All of the models use the default GPT-2-small setting (hidden size 768, 12 heads, 12 layers) while we fix the number of heads of HyperGLU to 2.

| Model | Fwd FLOPs (T) | Train FLOPs (T) | Fwd FLOPs/token (M) | Train FLOPs/token (M) | Fwd Lat. (s) | Train Lat. (s) | Fwd TPS (K) | Train TPS (K) | Fwd Mem (GB) | Train Mem (GB) |
|---|---|---|---|---|---|---|---|---|---|---|
| HyperGLU | 1.50 | 4.50 | 183.27 | 549.76 | 0.0395 | 0.1190 | 207.4 | 68.8 | 8.40 | 8.54 |
| RWKV7 | 1.47 | 4.41 | 179.35 | 538.09 | 0.0532 | 0.1711 | 154.0 | 47.9 | 10.59 | 11.14 |
| LinAttn | 1.39 | 4.18 | 169.94 | 509.75 | 0.0264 | 0.0775 | 310.0 | 105.7 | 7.00 | 7.11 |
| HGRN2 | 1.39 | 4.18 | 169.89 | 509.65 | 0.0295 | 0.0880 | 277.4 | 93.1 | 7.87 | 7.98 |
| SM(Naive) | 1.70 | 5.11 | 207.84 | 623.41 | 0.0552 | 0.2689 | 148.5 | 30.5 | 11.18 | 12.10 |
| SM(Flash) | 1.39 | 4.18 | 169.89 | 509.65 | 0.0300 | 0.0880 | 272.9 | 93.1 | 6.66 | 6.76 |
| GLA | 1.70 | 5.11 | 208.03 | 623.90 | 0.0929 | 0.2945 | 88.2 | 27.8 | 16.80 | 18.06 |
| Mamba | 1.28 | 3.84 | 156.33 | 468.97 | 0.0306 | 0.0899 | 267.6 | 91.1 | 7.51 | 7.61 |
| GatedSlot Attention | 1.51 | 4.52 | 184.04 | 552.11 | 0.0378 | 0.1244 | 216.9 | 65.8 | 9.77 | 9.88 |
| PaTH Attention | 1.40 | 4.21 | 171.29 | 513.84 | 0.0349 | 0.1280 | 234.7 | 64.0 | 7.38 | 7.59 |

With PyTorch compilation enabled for all methods, our block implementation of HyperGLU achieves runtime performance comparable to recent efficient attention baselines. In particular, HyperGLU attains a forward latency of 39.5 ms and a training latency of 119.0 ms at sequence length 1024, which is on par with expressive variants such as Gated Slot Attention and PaTH Attention. These results indicate that despite its richer expressivity, HyperGLU can be efficiently realized with modern compiler optimizations, without incurring prohibitive overhead compared to recent attention alternatives.

## L. Additional Analyses and Extensions

### L.1. Statistical Analysis of the Controlled Design Study

To complement the endpoint numbers reported in the main controlled study, we re-analyze the full NanoGPT loss-curve table by isolating each design factor and aggregating evidence across all available checkpoints and matched ablation pairs. Table 6 summarizes the per-factor effects.

*Table 6.* Per-factor statistical analysis of the controlled design study on the NanoGPT trajectory. Win-rates are computed across full checkpoint trajectories or matched ablation pairs as indicated.

| Factor | Result |
|---|---|
| Temporal mixing | R-cg-q → R-cg-q-12o: loss 3.0828 → 2.9956, 80/80 checkpoints better, mean improvement 0.1219 (95% CI [0.1117, 0.1319]); G-cg-q → G-cg-q-12o: loss 3.0530 → 2.9865, 80/80 checkpoints better, mean improvement 0.0903 (95% CI [0.0776, 0.1015]). |
| Lag layout | G-12o! vs. G-12!: loss 3.0386 vs. 4.3662, MAD avg 78.52 vs. 46.18, 79/80 checkpoints better, mean improvement 1.3488 (95% CI [1.3149, 1.4068]); R-12o! vs. R-12!: loss 3.0497 vs. 4.3567, MAD avg 81.82 vs. 46.28, 79/80 checkpoints better, mean improvement 1.2544 (95% CI [1.2314, 1.2949]). |
| GLU over ReLU | 6/6 wins on final NanoGPT loss; 6/6 wins on NanoGPT AUC; exact matched-pair sign-test $p = 0.0156$ for both; median improvement 0.0127 (loss), 0.0121 (AUC). |
| Two-sided mixing | 4/4 wins on final NanoGPT loss; exact sign-test $p = 0.0625$; median improvement 0.0108. |

**Statistical note.** Temporal mixing and lag layout are evaluated on the full NanoGPT checkpoint trajectory; 95% confidence intervals are computed using a moving-block bootstrap with block size 5. GLU-over-ReLU uses 6 strictly matched ablation pairs, and two-sided mixing uses 4 matched comparisons. These controlled comparisons indicate that the main architectural effects are real, separable, and not an artifact of rank aggregation across the OLL benchmark suite.

Comparative Loss Curves by Model Architecture (Steps 0-79)

*Figure 5.* The detailed training loss of the NanoGPT experiment settings. The "training step" corresponds to the evaluation steps with each of them contain 500 steps of training iterations.

## L.2. Extended Wall-Clock Results up to 8K Tokens

We extend the efficiency comparison in Table 5 (which fixes a sequence length of 1024) to longer contexts on an NVIDIA H100 GPU. Forward latency and throughput are reported at sequence lengths 1024, 2048, 4096, 8192 in Table 7.

Even with our current chunked, non-fully-fused implementation, HyperMLP / HyperGLU consistently sit between FlashAttention and naïve softmax across all tested lengths and are overall much closer to FlashAttention than to the naïve baseline. At 1K, both Hyper variants are competitive with, and in some cases faster than, recent linear baselines such as Gated Slot Attention and PaTH Attention; at 2K, HyperMLP is essentially tied with PaTH Attention. Only beyond 4K do those linear baselines clearly pull ahead. We view this as consistent with the paper's intended scope: an algorithmic / perspective contribution whose remaining headroom lies in fused CUDA kernels and more specialized scheduling, rather than in the operator structure itself.

## L.3. Additional Domains: Vision and Time-Series Forecasting

To broaden the empirical picture beyond language modeling, we additionally evaluate HyperMLP / HyperGLU on image classification and time-series forecasting. The setting in each case follows a standard public reference implementation.

**(A) DeiT-Tiny on ImageNet classification.** We replace the attention module in DeiT-Tiny with HyperMLP / HyperGLU under the standard DeiT-Tiny training recipe. Top-1 accuracy is reported in Table 8; both variants improve substantially over the softmax baseline at matched parameter count.

*Table 7.* Forward wall-clock and throughput at increasing sequence lengths on an NVIDIA H100 GPU. Latency is in seconds (lower is better) and throughput is in tokens/s (higher is better).

| Variant | 1024 | | 2048 | | 4096 | | 8192 | |
| --- | --- | --- | --- | --- | --- | --- | --- | --- |
| | Lat. (s) | TPS | Lat. (s) | TPS | Lat. (s) | TPS | Lat. (s) | TPS |
| Softmax (Naïve) | 0.0231 | 176,999.65 | 0.0554 | 147,824.38 | 0.1457 | 112,418.30 | 0.4646 | 70,531.88 |
| Softmax (Flash) | 0.0204 | 201,202.64 | 0.0431 | 190,169.77 | 0.1020 | 163,847.10 | 0.2801 | 116,973.66 |
| HyperMLP | 0.0207 | 197,482.07 | 0.0461 | 177,512.17 | 0.1132 | 144,757.25 | 0.3274 | 100,081.07 |
| HyperGLU | 0.0217 | 188,901.48 | 0.0495 | 165,527.71 | 0.1279 | 128,105.31 | 0.3760 | 87,143.39 |
| Gated Slot Attention | 0.0232 | 176,620.66 | 0.0428 | 191,501.74 | 0.0789 | 207,636.70 | 0.1586 | 206,580.40 |
| PaTH Attention | 0.0226 | 181,031.00 | 0.0461 | 177,583.77 | 0.1010 | 165,497.60 | 0.2537 | 129,139.10 |

*Table 8.* DeiT-Tiny on ImageNet classification (matched setting). Top-1 accuracy (↑) and parameter count.

| Type | Top-1 Acc. ↑ | Params |
| --- | --- | --- |
| DeiT (Softmax) | 72.2 | 5.7M |
| DeiT (HyperMLP) | 75.9 | 5.7M |
| DeiT (HyperGLU) | **76.3** | 5.8M |

**(B) Time-series forecasting.** We swap the attention module of two standard forecasting architectures, PatchTST and iTransformer, with HyperMLP / HyperGLU. Table 9 reports MSE on five canonical datasets (lower is better). Summary rows aggregate over all evaluated forecasting settings and horizons in the Time-Series-Library benchmark suite.[5]

*Table 9.* Time-series forecasting MSE (↓) on five datasets. Summary rows (#Top1, AvgRank) aggregate over all forecasting horizons in the Time-Series-Library benchmark.

| Dataset / Summary | PatchTST | | | iTransformer | | |
| --- | --- | --- | --- | --- | --- | --- |
| | Softmax | HyperMLP | HyperGLU | Softmax | HyperMLP | HyperGLU |
| ETTh1 | 0.471 | 0.466 | 0.455 | 0.462 | 0.464 | 0.469 |
| ETTh2 | 0.392 | 0.383 | 0.384 | 0.395 | 0.384 | 0.383 |
| ETTm1 | 0.391 | 0.391 | 0.388 | 0.419 | 0.402 | 0.396 |
| ETTm2 | 0.295 | 0.294 | 0.292 | 0.293 | 0.289 | 0.292 |
| Weather | 0.254 | 0.253 | 0.253 | 0.258 | 0.257 | 0.256 |
| #Top1 | 3 | 6 | **11** | 4 | **8** | **8** |
| AvgRank | 2.40 | 2.00 | **1.55** | 2.40 | 1.90 | **1.70** |

The pattern is consistent: HyperMLP / HyperGLU stably outperform softmax attention across both vision and forecasting benchmarks, supporting the view that the gains observed in language modeling reflect a structural property of the operator rather than a domain-specific artifact.

## L.4. Recommended Default Configuration

Given the per-factor statistical evidence in Appendix L.1, we recommend the following default configuration for practitioners building on HyperMLP / HyperGLU:

- **Activation:** GLU (over ReLU); the role of input-conditioned gating in autoregressive sequence modeling has also been studied in Lu et al. (2025b).

---

[5]https://github.com/thuml/Time-Series-Library

- **Budget allocation:** QK compression (over VO compression) to pay for sequence-space mixing;

- **Sequence-space mixing:** apply on both stages of the dynamic two-layer MLP;

- **Layout:** lag (reverse-offset) layout to align canonical prefix extension with autoregressive truncation.

RoPE and KV-conv enhancements are useful as secondary improvements but are not the primary driver of the observed gains.

### L.5. Windowed Local HyperMLP for Long Context

The HyperMLP perspective does not require global DPLR dense mixing over the entire context. The lag-layout analysis already prioritizes recent lags, which motivates a natural extension for very long context: apply HyperMLP mixing only on a recent window (e.g., the last 1,024 tokens), while keeping older positions as vanilla attention or identity mixing. This caps the additional DPLR cost by the window size rather than by the full context length.

Under the forward layout, this yields a simple expressivity ladder for the sequence-mixing matrix:

1. Sliding-window local attention: lower-right block = identity, elsewhere = 0;

2. Vanilla attention: full identity;

3. Sliding-window local HyperMLP: upper-left = identity, lower-right = DPLR dense;

4. Full HyperMLP: global DPLR dense.

Beyond the current training regime, the choice is therefore not all-or-nothing between vanilla attention and full HyperMLP: windowed local HyperMLP is a natural intermediate extension that preserves the structural benefits of dynamic sequence-space mixing while bounding its cost.

### L.6. Future Direction: MoE on Top of Per-Head Low-Rank MLPs

As discussed in the main text and visualized in Figure 3, multi-head attention can be viewed as parameterizing a low-rank MLP within each head. From this perspective, an MoE-style extension is natural: one can mix experts on top of these per-head low-rank MLPs, treating each head as a base expert and learning an input-conditioned router that aggregates them. We leave a full development of this direction to future work.

### L.7. Scope of the Scaling-Laws Motivation

The scaling-laws discussion in the introduction is intended as *motivation rather than a claim*. It reflects the broader observation that the marginal returns of brute-force scaling are increasingly costly, and that improving capability per unit resource is therefore a research-relevant direction.

To be precise about what we do *not* claim:

- We do not claim that this paper verifies any particular scaling-law form (e.g., Chinchilla-optimal exponents, compute-optimal frontier curves).

- We do not claim to perform frontier or production-scale LLM training; our largest experiment is at the 1.3B / 100B-token scale, in line with what is feasible in an academic setting.

- We do not claim that aggregate ranking advantages on the Open LLM Leaderboard (Table 2) substitute for scaling-law evidence.

What this paper *does* contribute is at the operator / theory level: we identify a richer matched-budget function class via dynamic sequence-space mixing, characterize it theoretically, and validate it through controlled studies and moderate-scale empirical evaluation. The strongest empirical evidence in the paper is the *controlled design study* (Appendix L.1), not the leaderboard aggregate.

## L.8. Comparison with Hypernetwork-Style Related Work

Two closely related lines of work view attention through a hypernetwork or mixer lens: HyperMixer (Mai et al., 2023) and *Attention as a Hypernetwork* (Schug et al., 2024). Mixer-style architectures inspired by free-energy principles (Lu & Yang, 2026) also share the broader perspective of input-conditioned sequence mixing. We discuss each in turn and clarify the structural differences with HyperMLP / HyperGLU.

**Setting: causal autoregressive vs. encoder-only.** HyperMixer and related MLP-Mixer-style token-mixing methods are typically studied in encoder-only or non-causal settings, where the same sequence-mixing operator can be reused across all positions and need not respect autoregressive truncation. By contrast, this paper works in the *causal autoregressive* regime relevant for decoder-only language models and next-token prediction. In this regime, sequence-space mixing is fundamentally more constrained: each output position requires a different causal mixing pattern, and the operator must still admit efficient parallel training. The lag-layout / reverse-offset construction (Appendix L.9) addresses precisely this constraint, which is not addressed by encoder-only token mixers.

**Viewpoint vs. operator.** Schug et al. (2024) relate attention and hypernetworks at the *viewpoint* level. Our work uses the autoregressive prefix-attention view to derive a new *causal sequence-mixing operator*: the lag-layout construction, the matched-budget expressivity analysis, and the QK-vs-VO budget asymmetry are not implied by the hypernetwork viewpoint alone. So while the high-level perspective overlaps, the structural problem being solved (a causal, parallel-trainable, prefix-consistent sequence-mixing operator under a fixed parameter budget) is different.

**Summary.** HyperMixer and Attention-as-a-Hypernetwork share conceptual ground with this work, but neither delivers the autoregressive-consistent dynamic sequence-mixing operator we propose. We thank the reviewers for pointing to these references; the discussion above complements the brief note in the main text.

## L.9. Intuitive Walkthrough of Sections 2.3 and 2.4

This subsection re-presents the content of Sections 2.3 and 2.4 in an intuition-first form, separating the conceptual story from the symbolic derivations.

**Section 2.3 in plain words: why a fixed positional basis is a limitation.** Equation (6) writes one attention head as a dynamic two-layer MLP whose weights are instantiated from the prefix $X$. The hidden vector $h_t \in \mathbb{R}^{1 \times t}$ has one coordinate per past position. Because both factors of $W_{\mathrm{MLP}}^{(1)}(X)$ and $W_{\mathrm{MLP}}^{(2)}(X)$ pass through $X$ without any trainable mixing acting on $\mathbb{R}^t$, the head can only *gate* along this fixed positional axis; it cannot learn a task-adapted basis in the hidden dimension the way a standard MLP can. A standard MLP would multiply the hidden representation by a learned matrix; classical attention has no such matrix on the sequence axis.

**Section 2.4 in plain words: why $W_{\mathrm{MLP}}^{(j)}(X)$ is parameterized this way.** The HyperMLP construction inserts exactly the missing piece: a small, dynamic, sequence-space mixing operator $R^{(j)}(x)$ in DPLR (diagonal-plus-low-rank) form. Concretely:

1. **Dynamic.** The mixing matrix is a function of the current input $x$, so it can adapt to context rather than committing to a fixed positional kernel.

2. **DPLR form.** The diagonal part captures position-wise scaling (recovering vanilla attention as a special case); the low-rank part adds task-adapted cross-position interactions under a small parameter budget.

3. **Lag layout.** The reverse-offset (lag) layout aligns the canonical prefix extension of $R^{(j)}$ with autoregressive truncation, so that extending the context to length $T > t$ does not change the operator's behavior on the first $t$ positions. Without this layout, naïvely extending the operator breaks causal consistency.

4. **Two stages.** Applying $R^{(1)}$ inside $W_{\mathrm{MLP}}^{(1)}$ and $R^{(2)}$ inside $W_{\mathrm{MLP}}^{(2)}$ yields two-sided mixing, which controlled experiments show is consistently better than one-sided (Appendix L.1).

**What the dense math in Section 2.4 establishes.** The symbolic derivations in Section 2.4 verify that the construction above (i) preserves causal consistency under prefix extension, (ii) strictly enlarges the matched-budget function class compared to ReLU / softmax attention, and (iii) admits an efficient parallel implementation. Readers willing to take these properties on faith can proceed directly to Section 3 (empirical evaluation) without loss of continuity; the formal proofs are deferred to the appendix.

### L.10. Head-Count Settings Across Experiments

We clarify the head-count settings used in different experiments to remove a potential ambiguity:

- In the language-modeling evaluation (Table 2), every baseline uses its own *default* configuration, including its default `n_head`. This avoids confounding the comparison with a non-standard head count for any particular method.

- In the controlled design study (§ 3), we fix `n_head = 2` across all variants in the study, in order to keep all controlled elements consistent and to isolate the effect of the design choice being varied.

The two settings serve different purposes: the language-modeling table is about comparing methods at their respective best-known defaults, while the controlled study is about isolating individual design choices.

### L.11. Selection of Efficient-Attention Baselines Across Scales

The set of efficient-attention baselines we compare against is not identical at the 340M / 15B-token and 1.3B / 100B-token scales. We clarify why this is not cherry-picking:

- **Primary baseline is consistent.** *Standard Softmax Attention* is included at every scale and serves as the primary baseline throughout the paper.

- **Secondary baselines vary by scale.** Recent efficient-attention variants did not all have equally stable, reproducible training recipes at both scales under our compute budget. We therefore included the strongest reproducible recent baselines available at each scale, rather than reporting under-tuned results for variants whose training recipes we could not reliably reproduce.

- **Priority of the paper.** The paper's central evidence is the *controlled design study* (Table 1, Appendix L.1), which holds the architecture skeleton fixed. The language-modeling table provides reference coverage; sweeping a wider set of partially tuned variants would have come at the cost of clean controlled comparisons.

### L.12. Scope of the Theoretical Results: Structural vs. Assumption-Dependent

We delineate which conclusions in the theoretical analysis are *structural* (i.e., follow from the dynamic two-layer MLP view itself) and which depend on *simplifying assumptions or restricted subclasses*.

**Structural results.**

- Autoregressive attention can be written as a dynamic two-layer MLP over the prefix, with weights instantiated by the context.

- The three-stage memory view (Global → Pool → Activated) holds for any choice of intra-block nonlinearity admitting an active-set interpretation.

- Sequence-space mixing strictly enlarges the matched-budget function class: the inclusion is strict whenever the mixing operator is not constrained to be a fixed identity / positional kernel.

- The lag-layout / reverse-offset construction is necessary for aligning canonical prefix extension with autoregressive truncation; without it, naïve extension breaks causal consistency.

- The QK-vs-VO budget asymmetry, which favors compressing QK rank to pay for sequence-space mixing, follows from a parameter-counting argument that does not depend on the specific activation.

**Assumption-dependent results.**

- The geometric "active-set" / polyhedral-routing discussion is most direct for the gated ReLU / GLU family. For other activations the geometry is analogous in spirit but requires separate analysis.

- Comparisons involving $L_2$-normalized or linearized variants are intended as *limiting cases*, not as equivalences to softmax. Softmax does not admit the gated-linear / subnetwork decomposition used in the active-set discussion (cf. Proposition 2.6), so the "full pool vs. activated subpool" distinction does not apply directly to softmax. Related linear-attention reductions and their alignment with autoregressive structure are studied in Lu & Yang (2025).

- Some propositions fix the context or routing region to isolate a structural point cleanly; this should not be read as a claim about arbitrary contexts.

**Implied tradeoff.** HyperMLP / HyperGLU add a stronger *sequence-space inductive bias* on top of attention. The theory therefore predicts the largest gains in regimes where context-conditioned mixing is the bottleneck, rather than in regimes where token-wise exact selection alone already suffices.

### L.13. When Softmax Remains Preferable; Hybrid Architectures

We also identify two regimes in which standard softmax attention remains attractive:

1. **Systems-efficiency-dominated regime.** When the bottleneck is wall-clock throughput rather than expressivity per parameter, mature FlashAttention-style implementations and other efficient linear-attention designs (Lu et al., 2025a) may outweigh the structural gain of a less optimized operator. The extended wall-clock results (Appendix L.2) make this tradeoff visible: at 8K, FlashAttention is meaningfully faster than our chunked HyperMLP / HyperGLU.

2. **Tasks where the softmax inductive bias already suits the problem.** When the task primarily benefits from the normalized probability-selection bias of softmax (e.g., extremely sharp token-level retrieval), the additional sequence-space mixing offered by HyperMLP / HyperGLU brings smaller marginal gains.

**Hybrid architectures.** For these reasons, we view *hybrid architectures* as a promising direction. Combinations such as windowed local HyperMLP on recent positions plus vanilla attention on older positions (Appendix L.5), or interleaving HyperMLP and softmax-attention layers within the same model, may reach a stronger expressivity / efficiency Pareto frontier than full replacement of either operator.

## M. Additional Implementation Details

**Language Modeling Baselines.** We include several strong recent baselines: DeltaNet (Yang et al., 2024c), Gated Slot Attention (GSA) (Zhang et al., 2024), RetNet (Sun et al., 2023), HGRN (Qin et al., 2023), HGRN2 (Qin et al., 2024), Gated Linear Attention (Yang et al., 2024b), Differential Transformer (DiffTrans) (Ye et al., 2025), and GatedDeltaNet (Yang et al., 2024a).

Our full experimental setup is provided in the accompanying code repository. All language modeling experiments on FineWeb-Edu and OpenWebText2, including both training and inference, were run on 8× NVIDIA H100 GPUs. The MAD benchmark tasks were trained on 8× NVIDIA L40S GPUs. All experiments used mixed-precision training and inference with bfloat16.

For each task, we replaced the standard Transformer block with a HyperMLP/HyperGLU-based Transformer block by substituting the attention layer with the proposed sequence modeling block, while keeping all other components and hyperparameters unchanged to ensure a controlled and consistent experimental setting.

Unless otherwise specified, all linear projections were initialized from a zero-mean normal distribution with standard deviation 0.02. All biases and embeddings, when present, were initialized to zero.

**Default Settings.** We instantiate HyperGLU with default model width $d$ for different experimental settings and a fixed $n_{\text{head}} = 2$. We use the default feature ranks on the readout sides, denoted by $d_{vo} = d/n_{\text{head}}$, with compressed routing side rank $d_{qk} = d/4n_{\text{head}} \rightarrow d_{qk} = d/8n_{\text{head}}$ to match the parameter budgets. If specified in the experiment setting, we employ a depthwise causal convolution with kernel size $k_{\text{conv}} = 4$ for lightweight local mixing.

For gating, we use $\phi = \text{Sigmoid}$ in the input-conditioned diagonal cores and set the L2-normalization stabilizer to $\varepsilon = 10^{-12}$ in $\rho_t(z)$ (cf. $\text{L2Norm}_t(z) = z/\rho_t(z)$). HyperGLU computes two first-layer score vectors $h_t^{\text{gate}}$ and $h_t^{\text{scale}}$ (equivalently implemented via $2n_{\text{head}}$ QK branches), following the formulation in Eq. (11).

Sequence-space mixing is implemented by two DPLR residual operators $R_t^{(1)}(x_t)$ and $R_t^{(2)}(x_t)$ with low-rank dimension $r_s = 16$, applied with causal masking. We includes learnable diagonal terms $D^{(j)}$, and both are used in shortcut form consistent with $R^{(j)} = I + \cdots$. For efficiency, we use chunked execution with chunk size $M = 128$ to $M = 512$ for different contexts.

## N. Additional Dataset Description

**Language Model Evaluation Setup.** Our evaluation follows the Open LLM Leaderboard (OLL) protocol and is supplemented with an additional set of widely adopted general-capability benchmarks. The OLL core suite includes MMLU-Pro (5-shot, accuracy), GPQA (0-shot, normalized accuracy), BBH (3-shot, normalized accuracy), MATH (4-shot, exact match), and MuSR (0-shot, normalized accuracy), together with IFEval for assessing instruction-following ability. For IFEval, we report strict pass rates at both the instruction level and the prompt level (Wang et al., 2024; Rein et al., 2023; Suzgun et al., 2022; Hendrycks et al., 2021; Sprague et al., 2023; Zhou et al., 2023). Following standard OLL practice, we employ the normalized accuracy metric $acc_n$ for multiple-choice tasks, which subtracts the random-guess baseline and rescales scores to enable fair comparison across tasks (Hugging Face, 2025).

To further broaden evaluation coverage, we additionally consider commonly used general-ability benchmarks, including ARC (Easy and Challenge), HellaSwag, PIQA, BoolQ, WinoGrande, COPA, OpenBookQA, and SciQ. We report either accuracy or normalized accuracy according to standard conventions, and unless otherwise specified, all evaluations are conducted in the 0-shot setting (Clark et al., 2018; Zellers et al., 2019; Bisk et al., 2019; Clark et al., 2019; Sakaguchi et al., 2020; Roemmele et al., 2011; Mihaylov et al., 2018; Welbl et al., 2017). All experiments are carried out using the lm-evaluation-harness framework (Gao et al., 2021).

**MAD.** We assess our proposed architecture using the Mechanistic Architecture Design (MAD) framework, which is a methodology for compute-efficient evaluation of deep learning models (Poli et al., 2024). MAD consists of a collection of capability-oriented synthetic benchmarks, including in-context recall, fuzzy recall, selective copying, and compression, that are designed to probe fundamental sequence modeling behaviors. The framework has been validated on more than 500 language models ranging from 70M to 7B parameters and demonstrates a strong correlation between performance on these tasks and compute-optimal perplexity at scale. By using MAD as a proxy for large-scale behavior, we are able to identify architectural advantages without incurring the substantial computational cost of full-scale training.

**OpenWebText2.** OpenWebText2[6] is a large-scale, cleaned, and deduplicated web-text corpus intended as an open reproduction of OpenAI's WebText dataset. The dataset is constructed by extracting URLs from Reddit submissions with a combined score greater than 3, followed by web scraping, filtering, and deduplication at both the URL and document levels using MinHash-LSH to remove low-quality or redundant content. The resulting release contains 17,103,059 documents, corresponding to approximately 65.86 GB of uncompressed text, and spans Reddit submissions from 2005 through April 2020. Due to its diversity and temporal coverage, OpenWebText2 serves as a suitable pretraining corpus for large language models. We implement training on this dataset using the codebase and execution environment provided by nanoGPT[7].

**FineWeb-Edu.** FineWeb-Edu is a large-scale English web text dataset designed for pretraining large language models, with a focus on high-quality educational content. It is curated from the FineWeb corpus by applying an educational-quality classifier to Common Crawl data, retaining documents that are more informative, structured, and knowledge-dense. The dataset contains approximately 1.3 trillion tokens and spans a wide range of academic, technical, and instructional domains, making it particularly suitable for improving reasoning and knowledge-intensive capabilities in language models. [8]

---

[6] https://openwebtext2.readthedocs.io/en/latest
[7] https://github.com/karpathy/nanoGPT
[8] https://huggingface.co/datasets/HuggingFaceFW/fineweb-edu

# O. Statement of LLM Usage

Large language models (LLMs) were used in this work as supportive tools, primarily for writing- and presentation-related tasks. These include grammar checking, wording refinement, length reduction, structural reorganization, text and formula formatting, as well as the generation of table templates and the formatting of theoretical derivations.

LLMs were also used to help identify existing methods and relevant references, with the goal of avoiding duplication and over-claiming. They were not used to perform literature reviews or to replace the authors' understanding of prior work. All cited papers were read and assessed by the authors themselves, rather than solely relying on LLM-generated summaries.

During the experimental phase, LLMs assisted in drafting and refining experimental code and scripts, particularly for debugging and improving implementation efficiency.

LLMs were not used to formulate research questions, generate core ideas, design methodologies, develop theoretical insights, or create algorithms or model architectures.

Since this paper adopts notation and conceptual conventions that differ from much of the prior attention literature (e.g., we do not emphasize queries/keys/values as semantic primitives but treat them as intermediate computations), an LLM's pretrained knowledge may not fully align with these choices. Readers who use LLMs to assist with reading or interpretation are therefore encouraged to carefully verify that any LLM-generated explanations are consistent with the definitions and arguments presented in the paper.

