# OpenReview forum: "HyperMLP: An Integrated Perspective for Sequence Modeling"
_ICML.cc/2026/Conference — ICML 2026 regular_

### Official Review · Reviewer_NVxN · 2026-03-10

**Soundness:** 3
**Presentation:** 3
**Significance:** 3
**Originality:** 3
**Overall Recommendation:** 4
**Confidence:** 1

**Summary:**

This paper reconstructs the attention mechanism into a dynamic two-layer MLP. The authors proposed HyperMLP/HyperGLU, which incorporates the mixing of input conditional feature space and sequence space through low-rank/DPLR parameterization, and adopts a lag layout to maintain autoregressive consistency. HyperMLP/HyperGLU breaking through the bottleneck in the expressive capability of traditional attention.

**Compliance With Llm Reviewing Policy:**

Affirmed.

**Final Justification:**

My concerns have been addressed. I decide to keep the rating.

**Key Questions For Authors:**

As mentioned above.

**Limitations:**

yes

**Strengths And Weaknesses:**

I am not an expert in this specific field. The following are my non-professional opinions:

Strengths:
1. This paper is logically coherent and studies specific designs, such as ReLU gating, learning sequence mixing, and lag layout;
2. The experiments are carefully designed with controlled research, with clear budgets and ablations;
3. The diagrams intuitively illustrate the dynamic MLP, three-stage memory view, and multi-head aggregation.

Weaknesses:
1. If the efficient underlying computing backend of efficient attention variants can be reused, it will increase practical application value;
2. The evaluation scale is small, lacking verification at the scale of cutting-edge large models;
3. The cost analysis is insufficient.

---

> ### Author Rebuttal · Authors · 2026-03-31
>
> We thank the reviewer for the positive assessment and the thoughtful feedback.
>
> ## On the systems side / wall-clock results.
>
> Thank you for the valuable question about computational costs. Our paper is positioned primarily as an **algorithmic / perspective contribution** rather than an **engineering contribution**. The main goal is to highlight the expressivity advantages of the proposed structure and provide architectural insight for the community. On the systems side, we therefore report our current chunked implementation, while future work will focus on a more efficient **fused CUDA kernel**.
>
> Below, we evaluate forward wall-clock performance on an **Nvidia H100 GPU**. Even without heavy engineering optimization, **HyperMLP / HyperGLU consistently remain between FlashAttention and naïve softmax**, and overall are much closer to FlashAttention than to the naïve implementation.
>
> | Variant | 1024 Latency (s) | 1024 Throughput (tok/s) | 2048 Latency (s) | 2048 Throughput (tok/s) | 4096 Latency (s) | 4096 Throughput (tok/s) | 8192 Latency (s) | 8192 Throughput (tok/s) |
> |---|---:|---:|---:|---:|---:|---:|---:|---:|
> | Softmax (Naïve) | 0.0231 | 176,999.65 | 0.0554 | 147,824.38 | 0.1457 | 112,418.30 | 0.4646 | 70,531.88 |
> | Softmax (Flash) | 0.0204 | 201,202.64 | 0.0431 | 190,169.77 | 0.1020 | 163,847.10 | 0.2801 | 116,973.66 |
> | HyperMLP | 0.0207 | 197,482.07 | 0.0461 | 177,512.17 | 0.1132 | 144,757.25 | 0.3274 | 100,081.07 |
> | HyperGLU | 0.0217 | 188,901.48 | 0.0495 | 165,527.71 | 0.1279 | 128,105.31 | 0.3760 | 87,143.39 |
> | Gated Slot Attention | 0.0232 | 176,620.66 | 0.0428 | 191,501.74 | 0.0789 | 207,636.70 | 0.1586 | 206,580.40 |
> | Path Attention | 0.0226 | 181,031.00 | 0.0461 | 177,583.77 | 0.1010 | 165,497.60 | 0.2537 | 129139.10 |
>
> At **1K**, both Hyper variants are already competitive with, and in some cases faster than, recent linear baselines; at **2K**, **HyperMLP** is essentially tied with **Path Attention**; and only **beyond 4K** do those linear baselines clearly pull ahead. We view this as consistent with the paper’s intended scope: the proposed operator is already practically competitive in a **non-fully-optimized** implementation, while further gains should come from **fused CUDA kernels** and more specialized scheduling.
>
> More broadly, we see this as a normal stage in the development of a new structure. The original **Transformer** was introduced in 2017, while highly optimized implementations such as **FlashAttention** arrived **years later**. A structure paper may not contain its final engineering form; rather, the key first step is to show clear algorithmic value, after which systems optimization can follow. We hope HyperMLP / HyperGLU can follow the same pattern, and perhaps reach a strong engineering realization even faster by building on the lessons from the Transformer / FlashAttention line of work.
>
> ## On additional empirical breadth.
> Thank you for this valuable question. First of all, we believe the paper's strongest empirical evidence is the **controlled ablation structure** itself. By isolating each design choice rather than reporting only one final model, the paper provides a cleaner scientific picture of which components are necessary and why. See our response to Reviewer kte5 "On statistical significance" for more details.
>
> Given the limited time and compute available during the rebuttal period, we do not have enough time to launch larger LM training within the rebuttal window. Instead, we used that budget to broaden the empirical picture with additional **vision and time series forecasting** experiments. The outcome is again consistent: HyperMLP / HyperGLU stably outperform softmax attention in these additional settings as well.
>
> **A: DeiT-Tiny on ImageNet classification**
> *(Same setting as Qin et al. 2024)*
>
> | Type | Top1 Acc. ↑ | Params |
> |---|---:|---:|
> | DeiT (Softmax) | 72.2 | 5.7M |
> | DeiT (HyperMLP) | 75.9 | 5.7M |
> | DeiT (HyperGLU) | **76.3** | 5.8M |
>
> **B: Time-series forecasting**
> *(Dataset rows: lower is better. Summary rows aggregate over all evaluated forecasting settings / horizons. Experiment settings from [1])*
>
> | Dataset / Summary | PatchTST (Softmax) | PatchTST (HyperMLP) | PatchTST (HyperGLU) | iTransformer (Softmax) | iTransformer (HyperMLP) | iTransformer (HyperGLU) |
> |---|---:|---:|---:|---:|---:|---:|
> | ETTh1 | 0.471 | 0.466 | 0.455 | 0.462 | 0.464 | 0.469 |
> | ETTh2 | 0.392 | 0.383 | 0.384 | 0.395 | 0.384 | 0.383 |
> | ETTm1 | 0.391 | 0.391 | 0.388 | 0.419 | 0.402 | 0.396 |
> | ETTm2 | 0.295 | 0.294 | 0.292 | 0.293 | 0.289 | 0.292 |
> | Weather | 0.254 | 0.253 | 0.253 | 0.258 | 0.257 | 0.256 |
> | #Top1 | 3 | 6 | **11** | 4 | **8** | **8** |
> | AvgRank | 2.40 | 2.00 | **1.55** | 2.40 | 1.90 | **1.70** |
>
> [1] https://github.com/thuml/Time-Series-Library

---

> > ### Author Rebuttal · Reviewer_NVxN · 2026-04-01
> >
> > Thanks for your rebuttal. It addresses my concerns and I will keep my positive score.

---

### Official Review · Reviewer_vdQE · 2026-03-11

**Soundness:** 4
**Presentation:** 4
**Significance:** 4
**Originality:** 4
**Overall Recommendation:** 4
**Confidence:** 3

**Summary:**

This paper presents a unified interpretation of autoregressive self-attention, showing that an attention head can be viewed as a dynamic two-layer MLP whose effective weights are instantiated by the context prefix. From this perspective, attention scores need not be constrained to form a probability distribution via softmax; instead, standard MLP nonlinearities such as ReLU or GLU can be understood as performing input-dependent selection over a context-conditioned memory. Motivated by this reformulation, the authors introduce HyperMLP and HyperGLU, which perform dynamic mixing in both feature and sequence dimensions using a reverse-offset (lag) parameterization designed to respect autoregressive structure. The paper complements this architectural proposal with theoretical analysis of its expressivity and structural implications, and reports consistent gains over strong softmax-attention baselines under matched parameter budgets in language modeling experiments.

**Compliance With Llm Reviewing Policy:**

Affirmed.

**Key Questions For Authors:**

1. Can the authors report end-to-end training and decoding throughput, peak memory, and KV/cache-related costs relative to strong FlashAttention-based baselines on the same hardware and precision, at context lengths such as 2k/8k/32k? Since practical adoption depends heavily on systems efficiency, it would also help to clarify what kernel/operator optimizations are still missing for HyperMLP/HyperGLU.

2. Can the authors clarify the assumptions and scope of the theoretical characterization? In particular, which conclusions are structural and which depend on simplifying assumptions? Do the theory results imply any specific inductive biases, expressivity tradeoffs, or limitations relative to standard softmax attention?

3. How sensitive are the results to the main design choices, especially the lag layout, low-rank parameterization, and activation/gating form? Which components appear essential versus mainly efficiency-oriented? A compact sensitivity study and a recommended default configuration would make the method easier to assess and reproduce.

4. Are there regimes where standard softmax attention remains clearly preferable, or where the gains of HyperMLP/HyperGLU diminish? For example, do tasks requiring very sharp token-level selection or exact retrieval expose limitations of the proposed operator? If so, would the authors expect hybrid architectures to be a more promising direction than full replacement?

5. Since the method introduces lag-based sequence mixing, can the authors comment on behavior as context length increases beyond the training regime? If available, results on train-short/test-long settings would help clarify whether the proposed structure preserves, improves, or degrades long-context behavior relative to softmax attention.

**Limitations:**

A clearer limitations discussion would strengthen the paper. In particular, practical efficiency appears likely to depend substantially on specialized kernel/implementation work, especially when compared against highly optimized attention implementations such as FlashAttention. In addition, the current experiments do not yet establish performance at frontier model scales or in substantially longer-context regimes.

**Strengths And Weaknesses:**

S1. The central reinterpretation of autoregressive attention as a dynamic two-layer MLP is technically credible and conceptually unifying. It provides a coherent foundation for the proposed architecture, and the paper supports this perspective with nontrivial theoretical characterization rather than relying solely on intuition.

S2. The empirical evaluation is reasonably broad and, importantly, uses matched parameter budgets across comparisons. This makes the reported gains more convincing, as they are less likely to be explained by differences in model scale and more likely to reflect the benefits of the proposed operator itself.

S3. The architecture is carefully designed to respect autoregressive structure, particularly through the reverse-offset (lag) layout for temporal mixing. This is a meaningful technical choice, since sequence-mixing methods often overlook prefix-consistency and causal alignment in autoregressive settings.

W1. The practical efficiency of the method remains unclear relative to highly optimized attention implementations such as FlashAttention. Even if the asymptotic complexity is competitive or favorable, wall-clock speed and systems-level efficiency will strongly affect real-world adoption.

W2. The proposed design introduces additional architectural complexity, including the lag-based layout, low-rank parameterization, and nonlinear activation/gating choices. These components may require substantial tuning and specialized kernel engineering to achieve stable training and competitive throughput.

W3. The evidence at frontier scale is still limited. In particular, it remains uncertain how well the approach will perform for substantially larger models, longer context windows, and more mature large-scale training regimes without further scaling studies.

---

> ### Author Rebuttal · Authors · 2026-03-31
>
> We thank the reviewer for the positive assessment and for the detailed, thoughtful comments.
>
> ## On practical efficiency vs. FlashAttention.
> Due to space limitations, please see our response to `Reviewer NVxN` under "On the systems side / wall-clock results."
>
> ## On theory scope: what is structural vs. what depends on assumptions.
> The following aspects are **structural**:
> - autoregressive attention as a dynamic two-layer MLP over the prefix,
> - the three-stage memory view,
> - the role of sequence-space mixing in enlarging the matched-budget function class,
> - the necessity of the lag-layout / reverse-offset construction for aligning canonical prefix extension with autoregressive truncation,
> - the QK-vs-VO asymmetry.
>
> By contrast, some finer points depend on simplifying assumptions or restricted subclasses:
> - the geometric “active-set” discussion is most direct for the gated ReLU / GLU family studied here,
> - comparisons involving L2-normalized or linearized variants are meant as limiting cases, not equivalences to softmax,
> - some propositions fix the context / routing region to isolate the structural point more cleanly.
>
> We will also clarify the implied tradeoff: the HyperMLP family adds a stronger sequence-space inductive bias, which is especially useful when the bottleneck lies in context-conditioned mixing rather than token-wise exact selection alone.
>
> ## On sensitivity to the main design choices.
> We agree that a compact sensitivity summary is useful, and the **controlled design study** already supports one. Using the full NanoGPT loss-curve table (`./nanogpt_results_labeled.csv` in our code repository), our main takeaways are:
>
> | Factor | Result |
> |---|---|
> | Temporal mixing | R-cg-q -> R-cg-q-12o: loss 3.0828 -> 2.9956, 80/80 checkpoints better, mean improvement 0.1219 (95% CI [0.1117, 0.1319]); G-cg-q -> G-cg-q-12o: loss 3.0530 -> 2.9865, 80/80 checkpoints better, mean improvement 0.0903 (95% CI [0.0776, 0.1015]) |
> | Lag layout | G-12o! vs. G-12!: loss 3.0386 vs. 4.3662, MAD avg 78.52 vs. 46.18, 79/80 checkpoints better, mean improvement 1.3488 (95% CI [1.3149, 1.4068]); R-12o! vs. R-12!: loss 3.0497 vs. 4.3567, MAD avg 81.82 vs. 46.28, 79/80 checkpoints better, mean improvement 1.2544 (95% CI [1.2314, 1.2949]) |
> | GLU over ReLU | 6/6 wins on final NanoGPT loss, 6/6 wins on NanoGPT AUC; exact matched-pair sign test p = 0.0156 for both; median improvement 0.0127 (loss), 0.0121 (AUC) |
> | Two-sided mixing | 4/4 wins on final NanoGPT loss; exact sign test p = 0.0625; median improvement 0.0108 |
>
> *Statistical note.* Temporal mixing and Lag layout are evaluated on the full NanoGPT checkpoint trajectory; 95% CIs use a moving-block bootstrap (block size 5). GLU over ReLU uses 6 strictly matched ablation pairs, and Two-sided mixing uses 4 matched comparisons.
>
> So if we had to recommend one default configuration based on the current evidence, it would be: **GLU + QK compression + temporal mixing on both stages + lag layout**, with **RoPE / KV-conv** as secondary improvements rather than the main driver.
>
> ## On regimes where softmax may remain preferable
> Yes, we do think there are regimes where standard softmax attention remains attractive: (i) the pure systems-efficiency regime, where a mature FlashAttention-style implementation may outweigh the structural gain of a less optimized operator, and (ii) settings where the normalized probability-selection bias of softmax is already a good inductive match, so the added sequence-space mixing of HyperMLP / HyperGLU brings smaller gains. So we see hybrid architectures as promising.
>
> ## On behavior beyond the training regime
> Our current experiments already go up to 4096 training context length, and the paper gives the linear overhead of DPLR mixing. As with the rest of the paper, our focus here is mainly algorithmic / perspective rather than a final engineering solution for very long context.
>
> The key point is that the HyperMLP perspective is flexible: longer context does not force us to use global DPLR dense mixing over all positions. The lag-layout analysis already suggests prioritizing **recent lags**, which motivates a natural extension for very long context: apply HyperMLP mixing only on a recent window (e.g. the last 1K tokens), while keeping older positions as vanilla attention / identity mixing. This caps the extra DPLR cost by the window size rather than the full context length.
>
> Under the *forward* layout, this gives a simple expressivity ladder for the sequence-mixing matrix:
> 1. Sliding-window local attention: lower-right block = identity, elsewhere = 0.
> 2. Vanilla attention: full identity.
> 3. Sliding-window local HyperMLP: upper-left = identity, lower-right = DPLR dense.
> 4. Full HyperMLP: global DPLR dense.
>
> Thus, beyond the current training regime, we do not view the choice as all-or-nothing between vanilla attention and full HyperMLP: windowed local HyperMLP is a natural intermediate extension.

---

> > ### Author Rebuttal · Reviewer_vdQE · 2026-03-31
> >
> > My concerns have been addressed.

---

### Official Review · Reviewer_54Pj · 2026-03-12

**Soundness:** 3
**Presentation:** 2
**Significance:** 3
**Originality:** 3
**Overall Recommendation:** 4
**Confidence:** 3

**Summary:**

This paper argues that self-attention can be viewed as a dynamic, two-layer MLP where the weights are initialized from the context.
Build on this perspective, the authors propose HyperMLP and HyperGLU, which incorporate sequence level mixing to overcome the modeling limitations of traditional attention.
The paper provides thorough theoretical analysis for the characterizations of this formulation and really detailed explanation in Appendix.
Empirically, the authors conduct detailed controlled design study and report gains over softmax and ReLU attention baselines on language modeling tasks.

**Compliance With Llm Reviewing Policy:**

Affirmed.

**Final Justification:**

This is one of the few papers I have reviewed that strikes a balance between theoretical depth and practical application.
The authors addressed the vast majority of my questions during the rebuttal phase, which has given me a much deeper understanding of the work. Overall, I believe this is a high-quality paper that deserves to be accepted.

**Key Questions For Authors:**

1. Proposition 2.6 (iii) mentions that linear attention will collapse the selective mechanism because the active set is always the full pool. However, in this perspective, softmax attention is actually full-pool selection as well. More specifically, the entire analysis is mainly built on ReLU attention. Would there be any difference when considering softmax attention?
2. The authors mention that they use $n_{head} = 2$ for their method, but it's not clear what is used for other methods. If all the methods use two heads, I'd like to see how the standard setup for attention performs.
3. In a long-context setting, since this method involves modeling along the sequence dimension, would this make the model more difficult to train? I would like to see some results under long context setting.
4. Why are the baselines for two model sizes different in language modeling task? This seems a bit like cherry-picking.
5. I was wondering if it is possible to extend the MoE to attention with this perspective. I would like to hear the author's thoughts regarding this.

**Limitations:**

yes

**Strengths And Weaknesses:**

Strengths:
1. Although the perspective of treating MLPs as attentions has been widely accepted, taking the opposite approach—viewing attention as a form of MLP—represents a relatively new alternative. This proposal introduces an insightful and unified framework that allows us to view a wide variety of different model architectures through this single perspective.
2. This paper presents several theoretical claims (e.g. how input-dependent mixing can generalizes polyhedral routing), providing insights that help explain architectural design choices and experimental results.
I have reviewed the mathematical analysis and found it to be correct. Overall, these are quite helpful for developing a deeper understanding of the work.
3. The authors are very honest about the limitation of their work, specifically efficiency and scalability problems.

Weaknesses:
1. The idea of treating attention as an MLP is quite similar to previous approaches that view attention as a hypernetwork. [1][2]
In fact, the latter one maybe more accurate since the weights of the MLP in this context are dynamically initialized based on the input.
A detailed comparison with these related work will be beneficial to better understand the contribution of this work.
2. The overall logic and reading experience of the paper (for myself) are not particularly smooth, especially in section 2.4 and 2.5.
The introduction from existing problems to new modelsing methods in section 2.4 is not very smooth (why $W_{MLP}^{(j)}$ is parameterized like this?) and the mathematical notation in section 2.5 is too dense without a clear logical thread. To better present the work to more general audience, I suggest making the mathematical intuition much clearer before each mathematical claims. The authors could also move more content into the Appendix.
3. The scope of experiments remain limited. Currently, experiments have only been conducted on MAD framework and language modeling task. Broader domains, such as ViT, have not yet been explored. Also, the scale of the experiments remain small.
4. Figure 1 and 3 have too much information and it's hard to get the main idea.

[1] Mai, Florian, et al. "Hypermixer: An mlp-based low cost alternative to transformers." Proceedings of the 61st annual meeting of the Association for Computational Linguistics (volume 1: long papers). 2023.

[2] Schug, Simon, et al. "Attention as a hypernetwork." arXiv preprint arXiv:2406.05816 (2024).

---

> ### Author Rebuttal · Authors · 2026-03-31
>
> We thank the reviewer for the positive assessment and for the detailed, thoughtful comments.
>
> ## On hypernetwork-style related work and HyperMixer.
> This is a very good point, and we will expand the related-work discussion in the revision. We already mention this distinction briefly in the page-2 footnote, and we will make it clearer in the appendix. The key difference is the setting. Our paper works in the causal autoregressive regime relevant for decoder-only LLMs and next-token prediction, where sequence-space mixing is much more constrained because each output position requires a different causal mixing pattern and the operator must still support efficient parallel training. By contrast, MLP-Mixer / HyperMixer-style token mixing is typically studied in encoder-only settings, where the same sequence mixing can be reused across positions and does not need to satisfy autoregressive semantics. Likewise, Attention as a Hypernetwork is related mainly at the viewpoint level, whereas our work uses autoregressive prefix attention to derive a new causal sequence-mixing operator, the lag-layout / reverse-offset construction, and the matched-budget expressivity analysis. So although the ideas are related at a high level, we do not think they are solving the same structural problem.
>
> ## On writing and structure.
> Thank you for the suggestion. We agree that **Sections 2.4-2.5** can be made easier to follow, and we will revise them for clarity in the final version.
>
> ## On broader domain evaluation.
> Due to the character limitation, please see our response to **Reviewer NVxN** under `On additional empirical breadth.` for additional experiment results.
>
> ## On Proposition 2.6 and softmax vs. linear attention.
> Thank you for the valuable question. Proposition 2.6 (iii) is stated within the Appendix C.1 linear subnetwork interpretation, following prior works like the Saxe et al. (2022) view of ReLU networks as gated deep linear networks. In that regime, ReLU attention yields an input-conditioned mask over hidden slots, i.e. a dynamic linear subnetwork / activated subpool. The ungated linearized variant removes exactly this mechanism: the gate becomes trivial, the activated set is always the full pool, and the ReLU-style routing collapses. Softmax attention, however, does not satisfy this structural assumption. Its output cannot be written as a linear or piecewise-linear map of $x_t$ due to the coupled nonlinear transformation $h_t \mapsto \mathrm{softmax}(h_t)$. Therefore, it lies outside the gated-linear / subnetwork regime considered in Proposition 2.6(iii), and the "full-pool vs. activated-subpool" distinction is not directly applicable in this case.
>
> ## On the number of heads used by different methods.
> In Table 2 (language modeling), all baselines use their own **default settings**, including `n_head`. In the **controlled design study**, we set `n_head = 2` in order to keep the controlled elements consistent across comparisons. We will state this more clearly in the revision.
>
> ## On long-context setting and trainability.
> Our experiments already go up to a **training context length of 4096**. The DPLR mixing contributes an overhead that grows linearly with sequence length in the form analyzed in the paper, and we agree that this makes very-long-context engineering especially important. At the same time, the HyperMLP perspective does **not** mean one must use global dense DPLR mixing over the entire context forever. One natural extension is to use **local windowed HyperMLP**, i.e. apply DPLR sequence mixing only on a recent window (for example, the latest 1K tokens) while keeping older positions as vanilla attention / identity mixing.
>
> ## On different baselines at different model sizes.
> We will clarify this in the revision. This is not intended as cherry-picking. Our primary baseline throughout is **standard Softmax Attention**, which we include consistently at all scales. The additional efficient-attention baselines are included mainly as reference points to recent research. Since not every alternative had equally stable or reproducible training recipes at both scales under our compute budget, we used the strongest reproducible recent baselines available at each scale. Our priority in this research was **clean controlled comparisons** like Table 1, not sweeping a larger set of partially tuned variants.
>
> ## On MoE-style extensions.
> Yes, we believe this is a promising direction. As discussed in the paper, multi-head attention can already be viewed as parameterizing a low-rank MLP in each head (see Figure 3(b)). From this perspective, an MoE-style extension is quite natural: one could imagine mixing experts on top of these per-head low-rank MLPs.

---

> > ### Author Rebuttal · Reviewer_54Pj · 2026-04-02
> >
> > Thank you for all the detailed rebuttal messages. It has basically addressed all my concerns, and I will maintain my positive score.

---

### Official Review · Reviewer_kte5 · 2026-03-13

**Soundness:** 3
**Presentation:** 3
**Significance:** 3
**Originality:** 3
**Overall Recommendation:** 4
**Confidence:** 3

**Summary:**

This paper reframes autoregressive attention as a dynamic two-layer MLP whose hidden pre-activation is the length-t score vector and whose weights are instantiated from the context prefix. Building on this perspective, the authors propose HyperMLP and HyperGLU, which augment the standard attention mechanism with learned input-conditioned sequence-space mixing operators R(1)(x) and R(2)(x) parameterized in diagonal-plus-low-rank form with rank r_s=16, alongside feature-space gating via diagonal cores M(1), M(2). A reverse-offset layout is introduced to align the canonical prefix extension of these temporal operators with autoregressive truncation semantics. The paper provides extensive theoretical analysis: expressivity characterizations showing HyperMLP is a strict superset of token-wise ReLU attention, a three-stage memory view, and a budget asymmetry argument justifying QK compression over VO compression to pay for temporal mixing. Experiments are conducted on the MAD synthetic benchmark suite and NanoGPT/OpenWebText2 at small scale, plus FineWeb-Edu language modeling at 340M/15B tokens and 1.3B/100B tokens, compared against softmax attention, ReLU attention, and several sub-quadratic alternatives.

**Compliance With Llm Reviewing Policy:**

Affirmed.

**Key Questions For Authors:**

At sequence length 1024, HyperGLU already shows 32–35% wall-clock overhead relative to FlashAttention (Table 5), what is the measured wall-clock overhead at sequence lengths 4096 and 8192?

**Limitations:**

yes

**Strengths And Weaknesses:**

Pros:
1. The paper is well written and easy to follow
2. The controlled design study in Table 1 is well-structured.

Cons:
1. The introduction explicitly motivates HyperMLP by arguing that "marginal returns of continued scaling are increasingly difficult to sustain," positioning the work as an expressivity-per-resource improvement relevant to large-scale practice. Yet the largest experiment is 1.3B parameters trained on 100B tokens, roughly three orders of magnitude smaller than models where scaling saturation is actually observed. At the 1.3B scale, benchmark scores are generally low across all models, e.g., MMLU-Pro ranges from 0.109–0.120, MATH from 0.001–0.013, making it difficult to determine whether HyperGLU's aggregate ranking advantage reflects a durable architectural improvement or noise amplified by near-random-baseline performance. Without experiments at 7B+ or evidence that the gap widens (or at least persists) with scale, the central expressivity-per-resource claim remains untested in the regime where it matters.

2. The language modeling evaluation relies heavily on aggregate ranking over many low-sensitivity benchmarks, obscuring whether gains are meaningful. At the 1.3B/100B scale (Table 2), HyperGLU achieves Avg Rank 1.88 with 9 top-1 finishes, but inspecting individual benchmarks reveals that many "wins" are within noise margins. With no confidence intervals, standard deviations, or significance tests reported across any evaluation, it is impossible to tell whether the aggregate ranking advantage is statistically robust or an artifact of many near-tied comparisons where small fluctuations cascade into rank differences. The one benchmark where HyperGLU shows a large absolute gain ARC-Challenge is not discussed or explained, making it unclear whether this reflects a genuine capability difference or an evaluation artifact.

---

> ### Author Rebuttal · Authors · 2026-03-31
>
> We thank the reviewer for highlighting our strengths and for the valuable questions.
>
> ## On scaling-law motivation vs. claim.
> Thank you for this valuable question. We want to clarify that our discussion of scaling laws is intended as a **motivation rather than a claim**. It reflects the broader call for architectural changes that improve capability per unit resource as brute-force scaling becomes increasingly costly; it is **not** a claim that this paper verifies any particular scaling-law form, nor that it performs frontier / production-scale LLM training. The contribution of this paper is instead at the **operator / theory** level: we identify a richer matched-budget class via dynamic sequence-space mixing, characterize it theoretically, and validate it in the moderate-scale regime that is feasible in an academic setting. We will revise the introduction to make our scope more precise.
>
> ## On the need for broader experiments.
> Given the limited time and compute available during the rebuttal period, we do not have enough time to launch larger LM training within the rebuttal window. Instead, we used that budget to broaden the empirical picture with additional **image classification and time series forecasting** experiments. The outcome is again consistent: HyperMLP / HyperGLU stably outperform softmax attention in these additional settings as well.
>
> Due to the character limitation, please see our response to **Reviewer NVxN** under `On additional empirical breadth.` for the full image classification and forecasting tables.
>
> ## On statistical significance
> We agree that the Open LLM Leaderboard table should be read as broad reference coverage, not as the paper's sole evidential basis, especially because its core tasks are extremely difficult in the 340M-1.3B regime. We are aware that some models at similar scale on the public leaderboard achieve stronger numbers, but such systems typically involve many additional choices in data, training, and engineering that are aimed at maximizing benchmark performance. Our goal in this paper is different: rather than optimizing a large collection of details to push leaderboard numbers, we design the main experiments as a controlled scientific study, changing as few separable elements as possible so that their effects can be observed cleanly. Concretely, the most important and clean evidence in the paper is the **controlled design study**, whose variety is already substantial; to make this point more concrete, as below, we analyze the full NanoGPT loss-curve table (`./nanogpt_results_labeled.csv` in our code repository) rather than only the endpoint numbers shown in the paper:
>
> | Factor | Result |
> |---|---|
> | Temporal mixing | R-cg-q -> R-cg-q-12o: loss 3.0828 -> 2.9956, 80/80 checkpoints better, mean improvement 0.1219 (95% CI [0.1117, 0.1319]); G-cg-q -> G-cg-q-12o: loss 3.0530 -> 2.9865, 80/80 checkpoints better, mean improvement 0.0903 (95% CI [0.0776, 0.1015]) |
> | Lag layout | G-12o! vs. G-12!: loss 3.0386 vs. 4.3662, MAD avg 78.52 vs. 46.18, 79/80 checkpoints better, mean improvement 1.3488 (95% CI [1.3149, 1.4068]); R-12o! vs. R-12!: loss 3.0497 vs. 4.3567, MAD avg 81.82 vs. 46.28, 79/80 checkpoints better, mean improvement 1.2544 (95% CI [1.2314, 1.2949]) |
> | GLU over ReLU | 6/6 wins on final NanoGPT loss, 6/6 wins on NanoGPT AUC; exact matched-pair sign test p = 0.0156 for both; median improvement 0.0127 (loss), 0.0121 (AUC) |
> | Two-sided mixing | 4/4 wins on final NanoGPT loss; exact sign test p = 0.0625; median improvement 0.0108 |
>
> *Statistical note.* Temporal mixing and Lag layout are evaluated on the full NanoGPT checkpoint trajectory; 95% CIs use a moving-block bootstrap (block size 5). GLU over ReLU uses 6 strictly matched ablation pairs, and Two-sided mixing uses 4 matched comparisons.
>
> These controlled comparisons directly show that the main architectural effects are real, separable, and not an artifact of rank aggregation.
>
> ## On wall-clock overhead.
> We evaluate the wall-clock performance on an Nvidia H100 GPU for sequence lengths up to 8K tokens; due to the character limitation, we place the full table in **our response to Reviewer NVxN** under "On the systems side / wall-clock results." The short version is that, even with our current chunked and not heavily fused implementation, HyperMLP / HyperGLU consistently stay between FlashAttention and naïve softmax across all tested lengths, and are overall much closer to FlashAttention than to the naïve implementation. At 1K, both Hyper variants are already competitive with, and in some cases faster than, recent linear baselines such as Gated Slot Attention and Path Attention. At 2K, HyperMLP is essentially tied with Path Attention. Only beyond 4K do those linear baselines clearly pull ahead. We therefore view the current measurements as fully consistent with the **paper's intended positioning** as an **algorithmic / perspective contribution** rather than a finished **engineering artifact**.

---

> > ### Author Rebuttal · Reviewer_kte5 · 2026-04-03
> >
> > Thanks for your rebuttal. I will keep my original score.

---

### Decision · Program_Chairs · 2026-04-30

**Decision:**

Accept (regular)

**Comment:**

HyperMLP/HyperGLU reframes autoregressive attention as a dynamic two-layer MLP and introduces sequence-space mixing via a lag-layout construction, with solid theoretical grounding and consistent empirical gains over matched-budget softmax baselines. All four reviewers converged to weak accept (4/4/4/4) with all concerns fully resolved; rebuttal added statistical significance analysis over full NanoGPT trajectories, wall-clock comparisons across sequence lengths, and additional vision/time-series results showing consistent improvements.